# Molecular Insights into New Particle Formation in Barcelona, Spain

**James Brean[1], David C.S. Beddows[1], Zongbo Shi[1],**
**Brice Temime-Roussel[2], Nicolas Marchand[2], Xavier Querol[3],**
**Andrés Alastuey[3], María Cruz Minguillón[3], and**
**Roy M. Harrison[1a∗]**

**[1]Division of Environmental Health and Risk Management**
**School of Geography, Earth and Environmental Sciences**
**University of Birmingham, Edgbaston, Birmingham B15 2TT**
**United Kingdom**

**[2]Aix Marseille Univ, CNRS, LCE**
**Marseille, 13003, France**

**[3]Institute of Environmental Assessment and**
**Water Research (IDAEA-CSIC), Barcelona, 08034 Spain**

**[a]Also at: Department of Environmental Sciences / Center of**
**Excellence in Environmental Studies, King Abdulaziz University, PO**
**Box 80203, Jeddah, 21589, Saudi Arabia**

---

∗ To whom correspondence should be addressed (Email: r.m.harrison@bham.ac.uk)

**ABSTRACT**

Atmospheric aerosols contribute some of the greatest uncertainties to estimates of global radiative forcing, and have significant effects on human health. New particle formation (NPF) is the process by which new aerosols of sub-2 nm diameter form from gas-phase precursors and contributes significantly to particle numbers in the atmosphere, accounting for approximately 50% of cloud condensation nuclei globally. Here, we study summertime NPF in urban Barcelona in NE Spain utilising particle counting instruments down to 1.9 nm and a Nitrate CI-APi-ToF. The rate of formation of new particles is seen to increase linearly with sulphuric acid concentration, although particle formation rates fall short of chamber studies of $H_2SO_4$-DMA-$H_2O$, while exceeding those of $H_2SO_4$-BioOxOrg-$H_2O$ nucleation, although a role of highly oxygenated molecules (HOMs) cannot be ruled out. The sulphuric acid dimer:monomer ratio is significantly lower than that seen in experiments involving sulphuric acid and DMA in chambers, indicating that stabilization of sulphuric acid clusters by bases is weaker in this dataset than in chambers, either due to rapid evaporation due to high summertime temperatures, or limited pools of stabilising amines. Such a mechanism cannot be verified in this data, as no higher-order $H_2SO_4$-amine clusters, nor $H_2SO_4$-HOM clusters were measured. The high concentrations of HOMs arise from isoprene, alkylbenzene, monoterpene and PAH oxidation, with alkylbenzenes providing greater concentrations of HOMs due to significant local sources. The concentration of these HOMs shows a dependence on temperature. The organic compounds measured primarily fall into the SVOC volatility class arising from alkylbenzene and isoprene oxidation. LVOC largely arise from oxidation of alkylbenzenes, PAHs and monoterpenes, whereas ELVOC arise from primarily PAH and monoterpene oxidation. New particle formation without growth past 10 nm is also observed, and on these days oxygenated organic concentrations are lower than on days with growth by a factor of 1.6, and thus high concentrations of low volatility oxygenated organics which primarily derive from traffic-emitted VOCs appear to be a necessary condition for the growth of newly formed particles in Barcelona. These results are consistent with prior observations of new particle formation from sulphuric acid-

amine reactions in both chambers and the real atmosphere, and are likely representative of the urban
background of many European Mediterranean cities.  A role for HOMs in the nucleation process
cannot be confirmed or ruled out, and there is strong circumstantial evidence for the participation of
HOMs across multiple volatility classes in particle growth.


## 1.    INTRODUCTION

Atmospheric aerosols, defined as liquid or solid droplets suspended in a gas, affect the climate both directly by scattering and absorbing radiation, and indirectly by acting as cloud condensation nuclei (CCN) (Penner et al., 2011), providing great uncertainties in estimates of global radiative forcing (IPCC, 2014). Further, fine ambient aerosols (defined as those with diameter below 2.5 μm) are the fifth greatest global mortality risk factor, resulting in 103.1 million disability-adjusted life year loss in 2015 (Cohen et al., 2017). The number concentration of the ultrafine fraction of these (aerosols with diameter below 0.1 μm, referred to as ultrafine particles or UFP) pose potentially significant health risks also, due to their high concentration and surface area. The more diffuse, gas-like behaviour of UFP allows them to penetrate into the deep lung and enter the bloodstream (Miller et al., 2017). Ultrafine particles occur in the urban environment either as primary emissions (e.g., from car exhaust (Harrison et al., 2018)) or secondarily as the product of new particle formation (NPF) (Brines et al., 2015; Guo et al., 2014; Kulmala et al., 2017; Lee et al,. 2019)

NPF is the formation of aerosol particles from gas-phase precursors. NPF can be considered a two-step process involving initial formation of a cluster of gas molecules at the critical diameter at around 1.5 nm - the diameter at which a free-energy barrier must be overcome to allow the spontaneous phase transition from gas to liquid or solid (Zhang et al., 2012), and the subsequent growth of this droplet to a larger aerosol particle. The first step of this process is dependent upon the stability and abundance of the clustering molecules. Sulphuric acid, water, and dimethylamine (DMA), for example, efficiently form particles as the strong hydrogen bonding between the acid base pair produces near negligible evaporation, much lower than the evaporation rate seen for the more weakly bound sulphuric acid-ammonia-water system. Nucleation of sulphuric acid, DMA and water proceeds at, or near to the kinetic limit in a chamber at 278 K when DMA mixing ratios are sufficient (Almeida et al., 2013; Kürten et al., 2014). Once past this 1.5 nm diameter, condensation and coagulation will drive particle growth. Both the abundance of condensable gases and their

vapour pressures limit condensational growth. Vapour pressures are especially important for the
initial growth stages, as the Kelvin effect barrier impairs condensation of more volatile species, with
this condition of low vapour pressures becoming less significant as the diameter of the particle
increases (Tröstl et al., 2016). Once sufficiently large (>50 nm), the loss processes of coagulation
and evaporation of these particles become inefficient, resulting in a significant atmospheric lifetime.
It is from these these diameters onwards the climate forcing effects of these particles become most
pronounced.

NPF processes happen globally, across a diverse range of environments from pristine polar regions,
to polluted urban megacities (Kerminen et al., 2018), and represent a significant source of CCN,
with 10-60% of NPF events shown to produce CCN and enhancement factors to CCN count ranging
from 0.5 – 11 (Lee et al., 2019 and references within). Strong NPF events are observed across a
range of urban environments, despite high condensation sinks $>10^{-2}$ s$^{-1}$ (Bousiotis et al., 2019; Yu et
al., 2016), and can act as a precursor to strong haze events (Guo et al., 2014). The occurrence of
urban NPF has only been partially explained by growing understanding from recent in-depth studies
(Yao et al., 2018). Recent advances in instrumentation allow for the measurement of particles down
to the critical diameter with instruments such as the particle size magnifier (PSM), and (Neutral) Air
Ion Spectrometer (NAIS/AIS) (Lee et al., 2019), and mass spectral techniques for measuring the
abundance and composition of neutral (Jokinen et al., 2012) and charged (Junninen et al., 2010)
clusters. Elucidated mechanisms with these techniques involve sulphuric acid and ammonia in
remote environments (Jokinen et al., 2018; Yan, 2018), monoterpene derived highly oxygenated
molecules (HOM) in remote environments (Rose et al., 2018), iodic acid in coastal environments
(Sipilä et al., 2016), and sulphuric acid and DMA in polluted urban environments (Yao et al., 2018).

Urban Barcelona sees frequent, strong summer-time NPF events occurring on 28% of days. These
events are associated with high insolation and elevated ozone (~60 µg m$^{-3}$) when considering the
whole year (Brines et al., 2014, 2015).Ground-level observations report NPF events starting
typically at midday, and either occurring in urban Barcelona and the surrounding regional
background simultaneously, or isolated to eitherurban Barcelona or just the regional background
(Dall'Osto et al., 2013). Vertical profiles over urban Barcelona reveal that NPF occurs at higher
altitudes, and starts earlier in the day, as at a given altitude these events are not suppressed by early
traffic peaks contributing to particle load (Minguillón et al., 2015). Here, we examine gas phase
mass spectral evidence and particle formation rates at the critical diameter from sulphuric acid in
Barcelona, with possible contribution from strong bases and highly oxygenated organic molecules
(HOMs), as well as factors influencing subsequent particle growth.

**2.     METHODS**
**2.1     Sampling Site**
The Palau Reial site in Barcelona (41°23'15" N, 2°6'53.64" E) is representative of the urban
background of Barcelona, located at the Institute of Environmental Assessment and Water Research
(IDAEA-CSIC) in the north-west of the city. Sampling was performed from a container 20 m from
a low traffic road, and 200 m from the nearest main road (Avinguda Diagonal). Data were taken
from 2018/06/28 through 2018/07/18.

**2.2     Chemical Ionisation Atmospheric Pressure Interface Time of Flight Mass**
**Spectrometry**
The Aerodyne Nitrate Chemical Ionisation Atmospheric Pressure interface Time of Flight Mass
Spectrometer (CI-APi-ToF) was used to make measurements of neutral oxygenated organic
compounds, organic and inorganic acids, bases, and their molecular clusters at high time resolution
with high resolving power. The ionization system charges molecules by adduct formation, such as
in the case of organic compounds with two or more hydrogen bond donor groups (Hyttinen et al.,
2015), or proton transfer in the case of strong acids like sulphuric acid (Jokinen et al., 2012).
Hydroxyl or hydroperoxyl functionalities are both common hydrogen bond donating groups, with
hydroperoxyl being the more efficient hydrogen bond donor (Møller et al., 2017). This instrument
has been explained in great detail elsewhere (Jokinen et al., 2012; Junninen et al., 2010), but briefly,
the front end consists of a chemical ionisation system where a 10 L min$^{-1}$ sample flow is drawn in
through the 1 m length 1" OD stainless steel tubing opening. A secondary flow was run parallel and
concentric to this sample flow, rendering the reaction chamber effectively wall-less. A 3 cm$^3$ min$^{-1}$
flow of a carrier gas ($N_2$) is passed over a reservoir of liquid $HNO_3$, entraining vapour which is
subsequently ionised to $NO_3^-$ via an X-ray source. Ions are then guided into the sample flow. The
nitrate ions will then charge molecules either by clustering or proton transfer. The mixed flows
travelling at 10 L min$^{-1}$ enter the critical orifice at the front end of the instrument at 0.8 L min$^{-1}$ and
are guided through a series of differentially pumped chambers before reaching the ToF analyser. All
data analysis was carried out in the Tofware package in Igor Pro 7 (Tofwerk AG, Switzerland).
Signals except for those of amines and ammonia are divided by the sum of reagent ion signals and
multiplied by a calibration coefficient to produce a concentration. A calibration coefficient of $3\times10^9$
cm$^{-3}$ was established based upon comparison with a sulphuric acid proxy (Mikkonen et al., 2011)
and is in line with a prior calibration with our instrument (Brean et al., 2020). Uniform sensitivity
between $H_2SO_4$ and all other species measured by CI-APi-ToF bar amines and ammonia was
assumed in this work. This introduces some uncertainties, as it relies upon both collision rates and
charging efficiencies to be the same within the ionisation source for all species. Amine and
ammonia signals are normalised to the nitrate trimer signal (Simon et al., 2016). Prior reports of
ammonia and amines as measured by CI-APi-ToF employed corona discharge systems, which
utilise higher concentrations of nitric acid, thus we report normalised signals. We present
correlations of each of these bases clustered with the nitrate dimer plotted against measurements
with the nitrate trimer, as well as their intercorrelations and example peak fits across Figure S1. $C_2$
amines, $C_4$ amines and ammonia were the only molecules of this kind found in our mass spectra.
Systematic uncertainties of +100% / -50 % arising from this method are assumed.
Due to the high resolving power of the CI-APi-ToF system (mass resolving power of 3000, and
mass accuracy of 20 ppm at 201 m/Q), multiple peaks can be fit at the same unit mass and their
molecular formulae assigned. Beyond 500 m/Q, peak fitting and assignment of compositions
becomes problematic as signal decreases, mass accuracy decreases, and the total number of possible
chemical compositions increases, so peaks above the $C_{20}$ region have not been assigned (Cubison
and Jimenez, 2015), however, signals past this region tended to be extremely low. All ions
identified are listed in Table S1. As proton transfer mostly happens with acids, and nearly all HOM
molecules will be charged by adduct formation it is possible to infer the uncharged formula;
therefore, all HOMs from here onwards will be listed as their uncharged form. The CI-APi-ToF
inlet was placed approximately 1.5 m a.g.l. CI-APi-ToF data is only available between the dates
2018/07/06 and 2018/07/17.

**2.2    Particle Size and Number Measurements**
Two Scanning Mobility Particle Sizer (SMPS) instruments measured particle size distributions at 5
minute time resolution, one Long Column SMPS (TSI 3080 EC, 3082 Long DMA, 3772 CPC, TSI,
USA) and one NanoSMPS (3082 EC, 3082 Nano DMA, 3776 CPC, TSI, USA) measuring the
ranges 10.9 – 478.3 nm and 4.5 – 65.3 nm respectively. A Particle Size Magnifier (A10, Airmodus,
FN) linked to a CPC (3775, TSI, USA) measured the sub-3 nm size fraction. The PSM was run in
stepping mode, operating at four different saturator flows to vary the lower size cut of particles that
it will grow (defined as the point of 50% efficiency, $D_{50}$). The instrument provided $D_{50}$ from 1.4 to
2.4 nm. The instrument switched between saturator flows each 2.5 minutes, giving a sub-2.4 nm
size distribution every 10 minutes. Aerosol sampling inlets were placed approximately 2 m a.g.l.

**2.3    Other Measurements**
Mixing ratios of non-methane VOC with proton affinity greater than $H_3O^+$ were made using the
proton transfer reaction time of flight mass spectrometer (PTR-ToF-MS 8000, Ionicon Analytik
GmbH, Austria). A detailed description of the instrument is provided by Graus et al., (2010) The
sampling set up, operating conditions, and quantification procedures are similar to those described
in Minguillón et al. (2016). Continual monitoring of composition and mass of submicron aerosol
>75 nm was carried out with an Aerosol Chemical Speciation Monitor (ACSM, Aerodyne, USA)
(Ng et al., 2011). Ozone, NO, and $NO_2$ were measured by conventional ultraviolet and
chemiluminiscence air quality instrumentation. Meteorological data were supplied by the Faculty of
Physics of University of Barcelona, from a nearby (200 m from the measurement site)
meteorological station located at the roof of an 8 floor building.

## 2.4  Condensation Sink and Particle Growth Rate

The condensation sink (CS) represents the rate at which a vapour phase molecule will collide with
pre-existing particle surface, and was calculated from the size distribution data as follows (Kulmala
et al., 2012):

$$CS = 2\pi D \sum_{d_p} \beta_{m,d_p} d_p N_{d_p} , \tag{1}$$

where D is the diffusion coefficient of the diffusing vapour (assumed sulphuric acid), $\beta_m$ is a
transition regime correction (Kulmala et al., 2001), $d_p$ is particle diameter, and $N_{dp}$ is the number of
particles at diameter $d_p$. The formation rate of new particles at size $d_p$ is calculated as follows:

$$J_{d_p} = \frac{dN_{d_p}}{dt} + CoagS_{d_p} . N_{d_p} + \frac{GR}{\Delta d_p} . N_{d_p} \tag{2}$$

where the first term on the right-hand side comprises the rate at which particles enter the size $d_p$,
and the latter two terms represent losses from this size by coagulation and growth respectively. $J_5$
was calculated using the data in the range of 5 – 10 nm, and $J_{1.9}$ was calculated using the
measurements in the range of 1.9 – 4.5 nm. We also calculated $J_{1.9}$ from our NanoSMPS data,
employing the equations of Lehtinen et al. (2007). $J_{1.9}$ from both methods showed reasonable
agreement ($R^2 = 0.34$). Agreement between $J_5$ and $J_{1.9}$ for each method was similar ($R^2 = 0.37$ and
$R^2 = 0.38$ for $J_{1.9}$ calculated from PSM data and from Lehtinen et al. (2019) respectively). $J_{1.9}$ is
greater than $J_5$ as predicted from equation (2) by around a factor of 20. See Kulmala et al. (2001) for
more information on calculation of coagulation sinks and formation rates. Growth rates in the range
of 4.5 – 20 nm were calculated according to the lognormal distribution function method (Kulmala et
al., 2012), whereas those in the range of 1.9 – 4.5 nm were calculated from PSM data using a time-
delay method between PSM and NanoSMPS data. Systematic uncertainties on our calculated $J_{1.9}$
values include 25% method uncertainty (Yli-Juuti et al., 2017), with a further 25% arising from
uncertainties in PSM cutoff ($\pm 0.5$ nm), as well as a 10% uncertainty in counting errors. A 50%
error arising from calculated coagulation sink is also applied (Kürtenet al., 2016). The above
calculations rely on the assumption of homogeneous air masses, and while air mass advection, as
well as primary particle emissions can cause errors in estimations of temporal changes in particle
count and diameter, the appearance and persistence of a new mode of particles across a period of
several hours is typically indicative of a regional process.

Growth rates from irreversible condensation of various vapours were calculated as according to
Nieminen et al. (2010). At our measured relative humidity, sulphuric acid favours binding to 2 $H_2O$
molecules (Kurtén et al., 2007). As amine concentrations are likely limited, we presume no mass
from amines in the condensing species. $H_2SO_4$ was assigned a density of 1.8 g cm$^{-3}$. For simplicity,
the properties of MSA regarding density and hydration were presumed the same as $H_2SO_4$, and
$HIO_3$ was presumed to have the same hydration as $H_2SO_4$, with a density of 4.98 g cm$^{-3}$. The
density of condensing organic vapours was assumed 1.5 g cm$^{-3}$, and concentration-weighted mean
mass (~276 g mol$^{-1}$ for LVOC) and atomic weighted diffusion volumes of organic compounds were
used to calculate GRs.

## 2.4 DBE and 2D-VBS

The double bond equivalent (DBE) describes the degree of unsaturation of an organic molecule and is defined simply as:

$$DBE = N_C - \frac{N_H}{2} - \frac{N_N}{2} + 1 \qquad (3)$$

The saturation vapour pressure at 300 K is defined by the 2D-volatility basis set (2D-VBS) as follows, if all nitrogen functionality is assumed to take the form -ONO$_2$ (Bianchi 2019; Donahue 2011; Schervish and Donahue, 2020):

$$Log_{10}(C^*)(300\,K) = (N_{C0} - N_C)b_C - N_O b_O - 2\frac{N_O N_C}{N_C + N_O}b_{CO} - N_n b_N \qquad (4)$$

where $N_C$, $N_H$, and $N_N$, are the number of carbon, hydrogen, and nitrogen atoms respectively. $N_O$ is the number of oxygen atoms minus $3N_N$ to account for -ONO$_2$ groups, $N_{C0}$ is 25 (the carbon number of an alkane with a saturation mass concentration of 1 µg m$^{-3}$), $b_C$, $b_O$, $b_{CO}$, and $b_N$ are 0.475, 0.2, 0.9 and 2.5 respectively, and represent interaction and nonideality terms. The final term of equation (4) accounts for -ONO$_2$ groups, each reducing the saturation vapour pressure by 2.5 orders of magnitude. $C^*$ values are calculated at 300 K and not corrected for temperature, as 300 K is within 1 K of the campaign average temperature.

## 3. RESULTS AND DISCUSSION

### 3.1 General Conditions of NPF Events

Summer NPF events in the regional background around Barcelona are associated with high insolation, relatively low ozone concentration (high compared with the rest of the year), and lower particulate matter load (Brines et al., 2014; Carnerero et al., 2019). Figure 1 shows an example of a day with no NPF in panel (a), referred to as "non-event" here, where two traffic-associated peaks in

particle number are seen during rush hours. Midday traffic peaks are also seen on certain days, but
these are easily distinguished from nucleation processes due to the lack of a significant <10 nm
mode. Figure 1(b) shows a nucleation day with growth to larger sizes >10 nm, termed "full-event",
showing the growth through the course of the day. These fulfil all the criteria of Dal Maso et al.
(2005). 4 events of this type were observed with CI-APi-ToF data coverage. Figure 1(c) shows a
day with nucleation occurring, but no growth past 10 nm. These days are referred to as "burst-
event" days. Here, NPF is seen to occur, but particles fail to grow past the nucleation mode. 2 such
events were seen in this data with CI-Api-ToF data coverage, and both are accompanied by a
distinct mode appearing beforehand in the range of ~20 – 40 nm. Condensation sinks were not
significantly higher than on full event days, so this failure of particles to grow further cannot be
attributed to condensational (or coagulational) losses. $GR_{4.5-20}$ ranged between 2.47 and 7.31 nm h$^{-1}$
(4.69 ± 2.03 nm h$^{-1}$), $GR_{1.9-4.5}$ ranged between 3.12 and 5.20 nm h$^{-1}$ (4.36 ± 1.02 nm h$^{-1}$). The
survival parameter (P) as suggested by Kulmala et al. (2017) is defined as $CS \cdot 10^{-4} / GR$, and for this
data is equal to 41, higher than other European cities.

Figure 2 contains box plots showing condensation sink, temperature and global radiation for all 3
NPF types across the entire day (diurnal profiles plotted in Figure S2). Condensation sinks during
NPF periods of both types (Figures 1(b) & 1(c)) were not significantly lower than in non-event
periods. Condensation sinks were supressed prior to the beginning of an event for full-events,
increasing relative to non-events through the afternoon period. Of the two burst-events, one was
similarly characterised by a suppression to condensation sink, whereas the other showed a sharp rise
in the midday. Global radiation and temperature were higher for full-events, most significantly for
temperature. Figure 3 is as Figure 2 but for sulphuric acid, ammonia and amines, and HOMs as
measured by CI-Api-ToF (HOM criteria are discussed in section 3.3.1). Sulphuric acid is elevated
during both full-event and burst-event periods. In urban Barcelona, sulphuric acid will primarily
arise from oxidation of $SO_2$ by the OH$^{\cdot}$ radical, with anthropogenic emissions such as shipping

emissions from the port areas being significant sources of $SO_2$ (Henschel et al., 2013). Direct traffic emissions have been shown to be a significant primary sulphuric acid source (Olin et al., 2020), but our sulphuric acid data show no traffic peaks. Ammonia and amines show enhancement for full-event periods, but not burst-event periods. Nucleation rates (at typical tropospheric sulphuric acid concentrations) are sensitive to amine concentrations in the range of a few pptv, with enhancements to amine mixing ratios past this point increasing the nucleation rate marginally (Almeida et al., 2013), while typical concentrations of DMA and other alkylamines vary from zero to a few pptv in the boundary layer (Ge et al., 2011a).

Barcelona has been shown to contain ppbv levels of ammonia (Pandolfi et al., 2012), arising from both agriculture to the north (Van Damme et al., 2018), and anthropogenic activities such as waste management and traffic, with waste management being the primary ammonia source. Highest ammonia mixing ratios are found in the densely populated old city centre (Reche et al., 2015). Agriculture, waste management, and traffic are also all significant sources of low molecular weight alkylamines, such as DMA (Ábalos et al., 1999; Cadle and Mulawa, 1980; Hutchinson et al., 1982; Ge et al., 2011a), and are likely the source of amines found in this dataset. Activities such as composting and food industry are especially strong sources of trimethylamine (TMA) (Ge et al., 2011a). Although high emission fluxes of TMA are expected in this environment, they are not present in our spectra. The TMA ion has been reported previously with a similar ionisation setup to that utilised in this study (Kürten et al., 2016). On full-event days, the signal for $C_2$ and $C_4$ amines has a midday elevation concurrent with peaks to solar radiation (Figure S2), and can help explain the high formation rates we see in this dataset (see section 3.2). The relative strength of these signals are shown in Figure S3, with significantly higher signals attributed to ammonia compared to amines, despite a likely lower sensitivity (Simon et al., 2016).

HOM concentrations were greatly enhanced during full-event periods (factor of 1.5 higher
compared to non-NPF mean), but lower during burst-event periods (factor of 1.2 lower compared to
non-NPF mean), implying their necessity for growth. The sources and implications of these HOMs
are discussed in section 3.3. Further, concentrations of iodine and DMS-derived acids such as iodic
acid ($HIO_3$) and methanesulphonic acid (MSA) are low ($7.8 \cdot 10^5$ and $3.3 \cdot 10^5$ cm$^{-3}$ respectively),
indicating a small influence of oceanic emissions on particle nucleation/growth. Extended box plots
as Figures 2 & 3 are presented in Figure S4, and HYSPLIT back trajectories per event in Figure S5.

**3.2      Mechanisms of New Particle Formation**
The correlation between $J_{1.9}$ and concentration of sulphuric acid is plotted in Figure 4. A close
relationship between nucleation rates and sulphuric acid concentrations ($R^2 = 0.49$) are consistent
with observations globally (Lee et al., 2019). This relationship is not dependent upon condensation
sink. These NPF rates have no dependence on other ions as measured by CI-Api-ToF, including
$HIO_3$, MSA, ammonia, amines or HOMs ($R^2$ for all <0.1). This is not to say that all of these
molecules are not involved in the nucleation process, rather that elevations or reductions to their
concentrations during nucleation periods do not have significant impact on nucleation rates. In the
example of alkylamines, their gas phase concentration may decrease due to clustering with elevated
sulphuric acid, as they cluster at around a 1:1 ratio at high amine mixing ratios (Kürten et al., 2014)
(and therefore they will not be detectable as free amines). Further, if amines are present at a few
pptv, their mixing ratios are significantly higher than our ambient measured sulphuric acid
concentrations, and will be sufficient to accelerate nucleation rates (Almeida et al., 2013).
Photochemical losses will also be greater during the periods of highest NPF rate (Ge et al., 2011b).
The strength of the relationship between sulphuric acid and nucleation rate has been quantitatively
reproduced in chamber studies involving the $H_2SO_4$-$H_2O$-DMA, and $H_2SO_4$-$H_2O$-BioOxOrg
system, accurately reproducing tropospheric observations of nucleation rates (Almeida et al., 2013;
Riccobono et al., 2014), although a later revision of the former shows nucleation rates at 278 K
exceeding typical tropospheric observations in the presence of high mixing ratios of DMA (Kürten
et al., 2018). A comparison between our data and results from the CLOUD chamber is presented in
Figure 5; included are the $H_2SO_4$-$H_2O$, $H_2SO_4$-$NH_3$-$H_2O$ (Kirkby et al., 2011), $H_2SO_4$-$H_2O$-DMA
(Kürten  et al., 2018) and $H_2SO_4$-BioOxOrg-$H_2O$ systems (Riccobono et al., 2014) – BioOxOrg
refers to the oxidation products of pinanediol ($C_{10}H_{18}O_2$) and $OH^.$. Data from these chamber
experiments is for 278 K and 38 – 39 % relative humidity. Nucleation rates measured in Barcelona
($J_{1.9}$ 178 ± 190 $cm^{-3}$ $s^{-1}$ at [$H_2SO_4$] 7.1·$10^6$ ± 2.7·$10^6$ $cm^{-3}$) are around an order of magnitude lower
than that seen for the $H_2SO_4$-DMA-$H_2SO_4$ system , but exceed that of the $H_2SO_4$-BioOxOrg-$H_2O$
system by ~1 order of magnitude, and that of the $H_2SO_4$-$NH_3$-$H_2O$ and $H_2SO_4$-$H_2O$ system multiple
orders of magnitude. No dissimilarity is seen between the data points corresponding to full or burst
type nucleation, indicating similar mechanisms of formation, despite lower HOM concentrations on
burst-event days. Conversely, research in remote boreal environments show that the mechanism of
nucleation can modulate dependent upon the $H_2SO_4$:HOM ratio (Yan et al., 2018). Model studies of
sulphuric acid-amine nucleation show a decline in nucleation rate with an increasing temperature
(Almeida et al., 2013; Olenius et al., 2017), as the evaporation rate of sulphuric acid-amine clusters
will increase with temperature (Paasonen et al., 2012). Conversely, evaporation rates of such small
clusters, and resultant nucleation rates tend to increase modestly with increases in relative humidity,
most pronounced at lower amine concentrations (Almeida et al., 2013; Paasonen et al., 2012).
Despite this, high nucleation rates at temperatures nearing 300 K have been reported previously
(Kuang et al., 2008; Kürten et al., 2016), although these tend to show a temperature dependence
(Yu et al., 2016).  No higher-order sulphuric acid clusters, sulphuric acid-base clusters, nor
sulphuric acid-HOM clusters were visible in the mass spectral data, likely due to these being below
the limit of detection of the instrument (Jokinen et al., 2012), so cluster identity cannot be directly
identified. Sulfuric acid trimer stabilisation is dependent upon base abundance (Ortega et al 2012),
and conversely, sensitivity of nitrate CI-Api-ToF to sulfuric acid-base clusters is reduced due to the
high base content of such clusters (Jen et al., 2016).
To further explore the relationship between sulphuric acid clusters and the rate of nucleation, the
sulphuric acid dimer:monomer ratio is plotted in Figure 6.  The sulphuric acid dimer:monomer ratio
is elevated by the presence of gas-phase bases such as DMA, and this elevation is dependent upon
both the abundances and proton affinities of such bases (Olenius et al., 2017). Upon charging,
evaporation of water and bases from sulphuric acid clusters occurs, and thus these are detected as
sulphuric acid dimer (Ortega et al., 2012, 2014). The binding energy of the bisulphate-$H_2SO_4$ ion is
in excess of 40 kcal mol$^{-1}$ (Curtius et al., 2001), and thus minimal declustering of the dimer is
expected within the CI-Api-ToF instrument – however, declustering of higher order sulphuric acid
clusters has been shown to be sensitive to voltage tune (Passananti et al., 2019), and this likely
extends to the dimer also, and as such discrepancies between sets of results due to instrument setup
cannot be ruled out. The ratio of sulphuric acid dimer:monomer is also highly sensitive to
condensation sinks, with a difference in dimer concentration of approximately a factor of 4
expected at $10^7$ cm$^{-3}$ between 0.001 s$^{-1}$ (a clean environment) and 0.03 s$^{-1}$ (condensation sinks
during these NPF events measured in this dataset) (Yao et al., 2018) and thus our low
dimer:monomer ratio can, in part, be explained by elevated condensation sinks. The dashed line
represents the ratio that would be seen due to ion induced clustering (IIC) in the nitrate chemical
ionisation system for a 50 ms reaction time (Zhao et al., 2010). The sulphuric acid dimer:monomer
ratio seen in the CLOUD $H_2SO_4$-DMA-$H_2O$ system is plotted, alongside our own data from
Barcelona. The ratio from our own data is seen to be much lower than that for the system purely
involving DMA as a ternary stabilising species. Similarly, this ratio is lower than for reports of
$H_2SO_4$-DMA-$H_2O$ nucleation in Shanghai (Yao et al., 2018), but is markedly similar to reports in
central rural Germany (Kürten et al., 2016). Similar to central Germany, this ratio increases at lower
sulphuric acid concentrations to a ratio more similar to the $H_2SO_4$-DMA-$H_2O$ system. A possible
explanation for this is that at higher sulphuric acid concentrations, the concentrations of stronger
stabilising bases are insufficient to stabilise all present sulphuric acid, with the higher end of the
sulphuric acid concentrations seen in this data roughly equivalent to 1 pptv sulphuric acid ($3\times10^7$
$cm^{-3}$ = 1.2 pptv sulphuric acid). We also cannot account for clustering due to naturally charged
sulphuric acid in the atmosphere, but ion concentrations in urban environments tend to be small due
to efficient sink processes (Hirsikko et al., 2011). Particle formation plausibly operates by sulphuric
acid-amine nucleation involving the measured $C_2$ and $C_4$ amines in our data, with nucleation rates
hindered relative to those measured in the CLOUD experiments by elevated temperatures, and a
decline to the sulphuric acid dimer:monomer ratio indicates that base concentrations may be
limited. We cannot rule out an involvement of HOMs in particle formation processes, and, as no
higher-order clusters were observed, we cannot establish sulphuric acid-amine nucleation with
certainty.

### 3.3      HOMs and Growth

### 3.3.1      HOM composition and sources

Barcelona, as a densely populated urban agglomerate, is distinct from the remote conditions under
which HOMs have primarily been studied (Bianchi et al., 2016, 2017; Schobesberger et al., 2013;
Yan et al., 2016), and is characterised by elevated temperatures, insolation and $NO_x$ mixing ratios,
as well as a diverse host of potential precursor VOC. The first of these affects HOM yields
significantly, as yields are highly dependent upon temperature (Quéléver et al., 2019; Stolzenburg et
al., 2018). Lower temperatures result in slower H-abstractions, which will result in the likelihood of
an $RO_2$. To undergo a different reaction pathway, such as termination with $HO_2$. To increase (Praske
et al., 2018). This is particularly important in this study if there is a large energy barrier for the first
or second H-abstraction taking place, as this will determine the number of hydrogen bond donating
groups, and therefore whether the $NO_3^-$ CI-Api-ToF is sensitive to a molecule or not. Elevated
insolation will result in enhanced photochemistry, and thus more rapid $RO_2$. Formation rates,
whereas elevated $NO_x$ will produce more HOM with nitrate ester functionality (Garmash et al.,
2020; Rissanen, 2018), which tend towards higher volatilities, and less efficient participation in
particle formation (Ehn et al., 2014; Lehtipalo et al., 2018), and growth (Yan et al., 2020).

Oxygenated volatile organic compounds (OVOC) are defined as species visible in the nitrate CI-APi-ToF that do not classify as HOM. Here, the first of the three criteria provided by Bianchi et al. (2019), that HOM must be formed by peroxy radical autoxidation, cannot be applied to define HOM, as knowledge as to whether a molecule is a result of autoxidation requires sound knowledge of the structure of the precursors, oxidants and peroxy radical terminators present, however, the number of molecules observed with $N_n = 2$ is around an order of magnitude lower than that for $N_n = 1$, where the primary source of multiple nitrogen functionalities would be multiple peroxy radical termination reactions from $NO_x$, and therefore while multiple generations of oxidation have been shown to occur in aromatics (Garmash et al., 2020), it is a small contributor to the concentration of what is classed as HOM here. The second criterion to define HOM are that they must be formed in the gas phase under atmospherically relevant conditions, which we deem appropriately fulfilled as all CI-APi-ToF measurements are of gas phase compounds, and the final criterion is that HOM must contain more than 6 oxygen atoms. To attempt to satisfy these criteria as best possible, the criteria of both containing 6 oxygen atoms and 5 carbon atoms or greater and having an O:C ratio >0.6 is applied.

The diversified range of HOM precursors in Barcelona will be primarily anthropogenic in origin. Averaged PTR-MS mixing ratios of different VOCs are presented in Figure S6. Figure 7(a) shows HOM concentration plotted against temperature, showing a dependence of HOM concentrations on temperature, with a lesser dependence on global radiation. The precursors for these HOMs are presumed to be largely isoprene, alkylbenzenes, monoterpenes, and PAHs. The mean peak intensities assigned to alkylbenzene derived HOMs are approximately a factor of two higher than those assigned to isoprene and monoterpene oxidation across this entire campaign. In this data these VOC mixing ratios are, with the exception of isoprene, not largely temperature dependent, with many of these HOMs forming under negligible or zero insolation, and therefore very low OH⋅ concentrations. These nighttime HOMs will not be derived from the oxidation of aromatics,

however, as rates of oxidation of alkylbenzenes by $O_3$ and $NO_3$. Are negligible (Molteni et al,
2018). These nighttime HOMs will therefore mostly be derived from biogenic emissions which
undergo more rapid nocturnal oxidation, and are likely transported from inland by the land breeze
during night (Millán, 2014; Querol et al., 2017).

Operating under the assumption that $C_5$, $C_6$, $C_7$, $C_8$, and $C_9$ HOMs primarily arise from isoprene,
benzene toluene, $C_2$-alkylbenzene $C_3$-alkylbenzene oxidation respectively (Massoli et al., 2018;
Molteni et al., 2018; Wang et al., 2017), HOM signals plotted against parent VOC concentration
indicate their dependence upon that VOC. Here, a $C_7$ HOM is thought to follow the formula $C_7H_{8-12}O_{5-10}N_{0-2}$.
We have plotted HOM concentrations against VOC concentrations in Figure 7(b). $C_{10}$
HOMs are not included in these analyses as these may primarily arise from $C_{10}H_{12-14}$ alkylbenzene,
or monoterpene oxidation. HOM concentration appears mostly independent of VOC concentration,
with the exception of isoprene, for which emissions are highly temperature dependent, and thus this
is likely a function of the effect of temperature on HOM formation (Figure 7(a)). A lack of
correlation between other VOCs and their HOMs confirms that this relationship between HOMs
and temperature is not a function of enhanced VOC emission fluxes from, for example, evaporation,
except in the instance of isoprene. Fragmented monoterpene oxidation products will also contribute
to the total number of $C_9$ HOMs, and similarly, other VOCs can fragment upon oxidation. However,
these results indicate that HOM concentrations are elevated by temperature, and operate quite
independent of precursor VOC concentration.

DBE as calculated by equation 3 is equal to the number of pi bonds and rings within a molecule.
Benzene, toluene, and similar aromatics have DBE = 4,  naphthalene = 7 and monoterpenes = 3.
DBE can be used as an indicator of sources when considering HOM in bulk. Saturation mass
concentration as calculated by equation 4 can help describe capacity of a molecule to both condense
onto newly formed particles and participate in nucleation. Figure 8 shows concentrations of HOMs
and other oxygenated organic molecules binned to the nearest integer $Log_{10}(C^*)$(300 K), coloured
by DBE. Mean ion signals per carbon number are shown in Figure S7. Most measured molecules
fall into the SVOC class ($0.3 < C^*$(300 K) $< 300$ μg m$^{-3}$) which will mostly exist in equilibrium
between gas and particle phase. Highest SVOC concentrations arise from fingerprint molecules for
isoprene oxidation under high $NO_x$ concentrations ($C_5H_{10}N_2O_8$) (Brean et al., 2020), and oxidation
of small alkylbenzenes ($C_7H_8O_5$, $C_8H_{10}O_5$). LVOC and ELVOC ($3\cdot10^{-5} < C^* < 0.3$ μg m$^{-3}$ and $3\cdot10^{-9}$
$< C^*$(300 K) $< 3\cdot10^{-5}$ μg m$^{-3}$ respectively) have a greater contribution from molecules with higher
DBE, i.e., $C_{10}H_{10}O_8$ arising most likely from PAH oxidation (Molteni et al., 2018), and $C_{10}H_{15}O_7N$,
a common molecule arising from monoterpene oxidation in the presence of $NO_x$. The contribution
of molecules with carbon number $\leq 9$ to these LVOC is modest, and ELVOCs are entirely
comprised of molecules with carbon numbers $\geq 10$, and is dominated by DBEs of 8 and 4,
attributable to PAH and monoterpene oxidation respectively. No molecules classed as ultra-low
volatility organic compounds (ULVOC, $C^*$(300 K) $< 3\cdot10^{-9}$ μg m$^{-3}$) were observed in our data, and
thus any pure HOM nucleation is unlikely.

### 3.3.2 HOMs and NPF

As shown in Figure 3, an elevated HOM concentration appears to be a necessary condition for particle
growth past 10 nm during NPF events. These days are associated with elevated temperatures, solar
radiation, higher ozone, and lower $NO:NO_2$ ratio. $HIO_3$ is also significantly higher on burst-event
days. A recent study in a remote environment reports growth rates matching condensation rates
without accounting for aqueous phase chemistry (Mohr et al., 2019). From 2D-VBS volatility
calculations discussed in the previous section, is it shown that LVOC and ELVOC measured in
Barcelona plausibly arise from the oxidation of aromatics (particularly PAHs in the case of ELVOC)
and monoterpenes. Calculated growth rates according to the method of Nieminen et al. (2010) are
presented in Figure S8 for both $GR_{1.9-5}$ and $GR_{5-20}$. Best agreement for $GR_{5-20}$ is when condensation
of SVOC, LVOC, ELVOC, MSA, $HIO_3$ and $H_2SO_4$ is considered, and best agreement for $GR_{1.9-5}$ is

seen for condensation of all these except SVOC. The uncertainties in this method are large, and

assumptions of irreversible condensation of SVOC onto particles of 5 nm likely lead to

overestimations; however, these results indicate an essential role of the condensation of organic

compounds to produce high growth rates observed in urban environments.

Figure 9 shows three mass-defect plots for a non-event period, full-event period, and burst-event

period. The non-event day included in Figure 8 was characterised by lower solar radiation and

temperatures than average, so lower signals for oxygenated species are seen due to weaker

photochemistry (i.e., $OH^.$ concentration), and slower autoxidation due to slower H-shift reactions

(Frege et al., 2018; Quéléver et al., 2019; Stolzenburg et al., 2018). The full-event day sees

enhancements to smaller OVOCs and HOMs compared to the non-event day, especially around

150-200 m/Q, which contains peaks corresponding to dicarboxylic acids and isoprene oxidation

products. Some of the largest peaks in the mass spectra correspond to formulae seen arising from

the enhanced $OH^.$ oxidation of alkylbenzenes (such as $C_7H_7NO_6$) (Molteni et al., 2018; Wang et al.,

2017). Larger HOMs see a less significant enhancement to smaller alkylbenzene derived HOMs.

The presence of larger, unidentified HOMs >400 m/Q is enhanced during full-events, these peaks

will comprise the largest compounds, most likely of class ELVOC, arising from the oxidation of

large VOCs, or $RO_2^.$-$RO_2^.$ accretion reactions, and thus, we likely underpredict ELVOC

concentrations and resultant impacts on particle growth in Figure S8. These unidentified peaks

>400 m/Q are both more numerous and larger during full-event periods, with a factor of two

difference in total peak area.  The burst-event day has significantly lower concentrations of OVOCs

and HOMs, and to a lesser degree, their nitrogen containing counterparts (N-OVOCs and N-

HOMs), with significantly fewer compounds >400 m/Q. The most significant difference between

full and burst-event days is in the SVOCs, accounting for a factor of two difference in

concentration. The sulphur containing acids all have similar peak areas to the full-event day. These

elevations to condensable OVOCs and HOMs on particle formation days with growth are consistent

with particle composition data as measured by ACSM (Figure S9). Particle composition on full-
event days shows an elevation to organic mass concentration in the late evening and night around
when new particles from NPF will reach sizes detectable by the ACSM (~75 nm, Ng et al., 2011).
Organic mass between 16:00 and 23:00 is 3.5 µg m$^{-3}$ on burst-event days, versus 7.8 µg m$^{-3}$ on full-
event days.

**4.      CONCLUSIONS**

We show new particle formation rates in Barcelona are linearly dependent upon the sulphuric acid
concentrations, and while formation rates far exceed that of $H_2SO_4$-BioOxOrg-$H_2O$ nucleation, they
fall short of those of $H_2SO_4$-DMA-$H_2O$ nucleation at 278 K, as does the sulphuric acid
dimer:monomer ratio, possibly explained by cluster evaporation due to high temperatures in
summertime Barcelona (303 K during events), and limited pools of gas-phase amines. These results
are similar to reports of nucleation rates in rural Germany (Kürten et al., 2016). As no higher-order
clusters were directly measured, we cannot determine nucleation mechanisms with certainty, and an
involvement of HOMs in nucleation is plausible.

High concentrations of OVOCs and HOMs were measured by CI-APi-ToF. Of these, the SVOC
arose from mostly isoprene and alkylbenzene oxidation, whereas LVOC and ELVOC arose from
alkylbenzene, monoterpene and PAH oxidation together, with a dependence of their concentration
on temperature. Concentrations of species associated with coastal and oceanic sources such as MSA
and $HIO_3$ were low. High HOM concentrations are seen to be a necessary condition for new particle
growth past 10 nm, with the most significant difference between days with and without particle
growth being SVOC concentrations (factor of 2 difference), while modelled growth rates from
condensation of these organic compounds, alongside $H_2SO_4$, MSA and $HIO_3$ were shown to match
growth rates within measurement error. Thus, oxidation of traffic derived alkylbenzenes and PAHs,

and to a lesser degree, isoprene and monoterpene emissions is a significant determinant of new

particle growth in this environment.

These results are consistent with extensive chamber and flow tube studies on particle formation

from sulphuric acid, amines and HOMs, and further, nucleation rates relative to sulphuric acid are

similar to many tropospheric observations. Barcelona is representative of many Mediterranean

urban environments, with moderate pollution, influence of shipping emissions, and high insolation,

and the present study reveals the complexity of NPF mechanisms in these environments.

## DATA AVAILABILITY

Data supporting this publication are openly available from the UBIRA eData repository at

https://doi.org/10.25500/edata.bham.00000434

## AUTHOR CONTRIBUTIONS

RMH and XQ conceived the study, JB and DCSB carried out the CI-APi-TOF and related

measurements with assistance from AA and MCM. The VOC measurements were proposed by NM

and collected by BT-R. JB wrote the first draft of the manuscript which was enhanced by

contributions from the co-authors.

## COMPETING INTERESTS

The authors have no conflict of interests.

## ACKNOWLEDGEMENTS

Financial assistance from the Spanish Ministry of Science, Innovation and Universities and

Competitiveness and FEDER funds under the project HOUSE (CGL2016-78594-R), and by the

Generalitat de Catalunya (AGAUR 2017 SGR41) is gratefully acknowledged. MCM acknowledges

the Ramón y Cajal Fellowship awarded by the Spanish Ministry. Financial support of the UK
scientists by the Natural Environment Research Council through the National Centre for Atmospheric
Science is also acknowledged (R8/H12/83/011).

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

 **FIGURE LEGENDS:**

**Figure 1:** Average SMPS contour plots for (a) non-event days, (b) full-event days and (c) burst-event days.

**Figure 2:** Box plots for days of non-event, full-event and burst-event, showing (a) condensation sink, (b) temperature, and (c) global radiation from hourly data. "Full-event" and "burst-event" include data across the entire day.

**Figure 3:** Box plots for days of non-event, full-event and burst-event, showing (a) sulphuric acid, (b) $C_2$ and $C_4$ amines, as clustered with the nitrate dimer and trimer, and (c) summed HOM concentration from $C_5+$ from hourly data. Units for ammonia + amines are normalised counts, as no calibration was performed. Event days include data across the full event day.

**Figure 4:** Formation rate ($J_{1.9}$) plotted against sulphuric acid monomer concentration, coloured by condensation sink. Circles represent burst-events, squares represent full events. Data is for hourly averages across NPF periods, typically within the hours 08:00 – 16:00. Slope of the line = $4.9 \cdot 10^{-5}$ s$^{-1}$. Error bars represent systematic uncertainties on [$H_2SO4$] and $J_{1.9}$

**Figure 5:** Formation rate plotted against sulphuric acid monomer concentration for data collected from Barcelona. Tan circles represent burst-events, purple squares represent full events. as well as that for the $H_2SO_4$-$H_2O$ (blue inverted triangles), $H_2SO_4$-$NH_3$-$H_2O$ (yellow inverted triangles), $H_2SO_4$-DMA-$H_2O$ (pink triangles), and $H_2SO_4$-BioOxOrg-$H_2O$ (brown diamonds) systems from the CLOUD chamber (Kürten et al., 2018 Kirkby et al., 2011; Riccobono et al., 2014). CLOUD chamber experiments were performed at 278 K and 38 – 39 % RH. Data is for hourly averages across NPF periods, typically within the hours 08:00 – 16:00. Error bars represent systematic uncertainties on [$H_2SO_4$] and $J_{1.9}$.

**Figure 6:** Sulphuric acid dimer concentration plotted against monomer concentration, showing burst-event periods (tan circles), full event periods (purple squares), non-event periods (green inverted triangles), and the ratio of sulphuric acid dimer:monomer in the CLOUD chamber for the $H_2SO_4$-$H_2O$-DMA system (pink triangles) (Almeida et al., 2013). Dashed line represents the dimer concentration produced by ion induced clustering in the chemical ionization unit (Zhao et al., 2010). CLOUD chamber experiments were performed at 278 K and 38 – 39 % RH. Data is for hourly averages across NPF periods, typically within the hours 08:00 – 16:00. Error bars represent systematic uncertainties on [$H_2SO_4$] and [($H_2SO_4)_2$].

**Figure 7:** Influencing factors on HOM concentration, showing (a) $C_{5-10}$ HOM concentration plotted against temperature, coloured by global radiation. Ellipsis shows 95% confidence on a multivariate t-distribution. (b) HOM concentration by carbon number potted against parent VOC mixing ratio. These are segregated by carbon number/VOC, i.e, $C_7$ HOMs plotted against toluene, under the assumption that toluene oxidation is the main producer of $C_7$ HOMs. Time for both plots is of hourly time resolution.

**Figure 8:** Concentrations of all oxygenated organic molecules and HOMs binned to integer $Log_{10}(C^*)$ values, coloured by DBE.

**Figure 9:** Mass defect plots for (a) non-event, (b) full-event, and (c) burst-event periods, data
taken from 10:00 – 15:00 on the days 11/07/2018, 16/07/2018 and 15/07/2018
respectively. Size corresponds to mass spectral peak area. Ions are coloured according
to identified chemical composition. *Blue* points correspond to HOMs containing all
organic species with ≥5 carbon atoms and ≥6 oxygen atoms, and an O:C ratio of >0.6.
*Purple* points correspond to the same but for species containing 1-2 nitrogen atoms.
Species not meeting this HOM criterea were classed generally as OVOCs which are
coloured *brown,* with the nitrogen containing OVOCs coloured *orange*. Sulphur acids
(*red*) include ions $HSO_4^-$, $CH_3SO_3^-$ and $SO_5^-$, as well as the sulphuric acid dimer. Iodine
acids (*green*) contains both $IO_3^-$ and $I^-$ (the latter presumably deprotonated hydrogen
iodide). Unidentified points are left uncoloured.


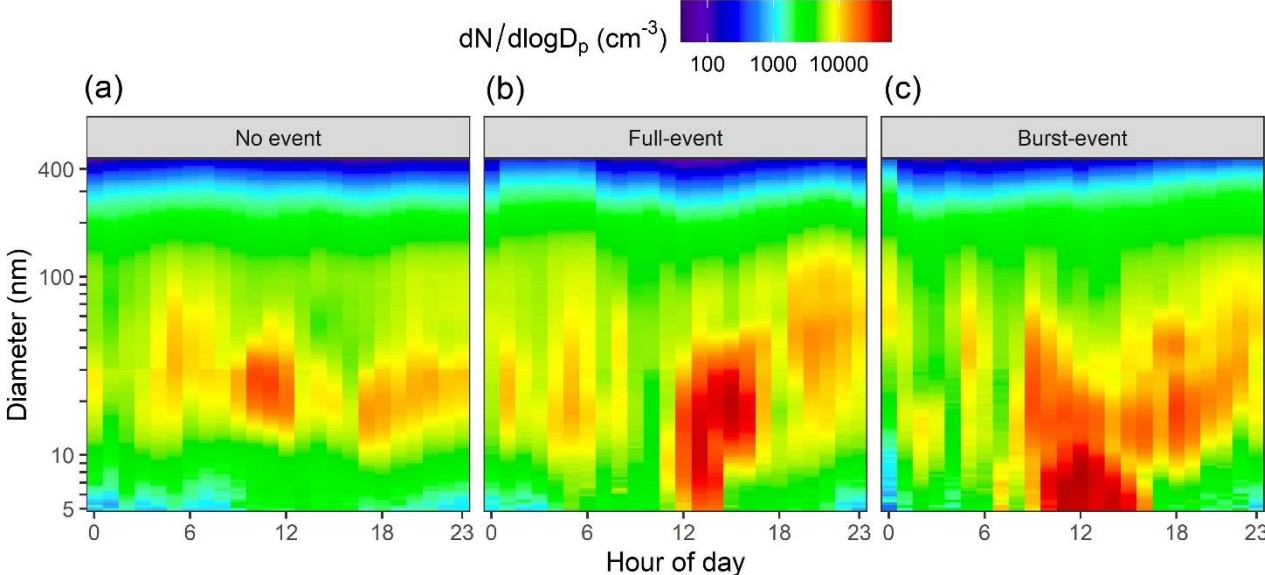


**Figure 1:** Average SMPS contour plots for (a) non-event days, (b) full-event days and (c) burst-event
days.


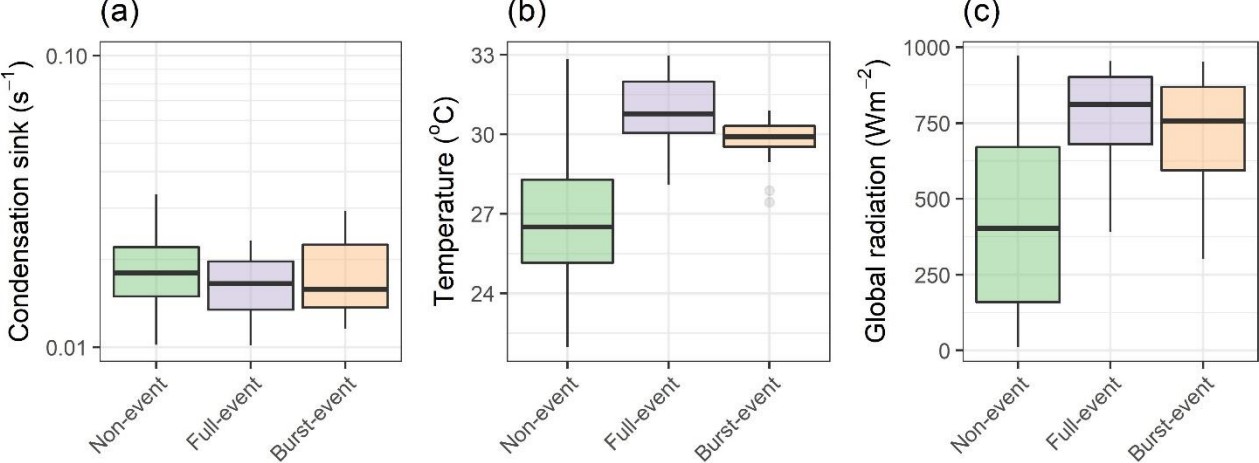

**Figure 2:** Box plots for days of non-event, full-event and burst-event, showing (a) condensation sink,
(b) temperature, and (c) global radiation from hourly data. "Full-event" and "burst-event" include
data across the entire day.

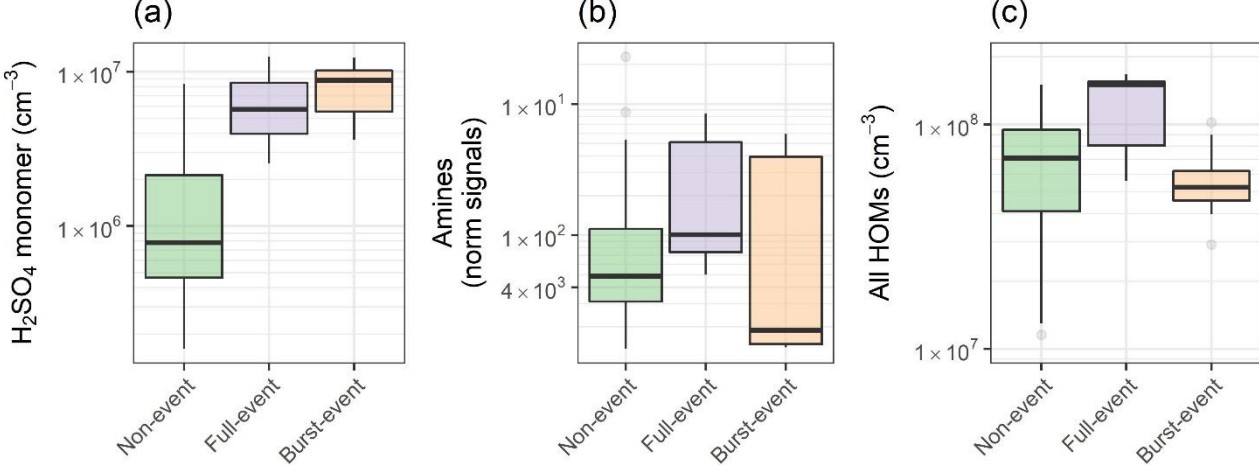

**Figure 3:** Box plots for days of non-event, full-event and burst-event, showing (a) sulphuric acid, (b) $C_2$ and $C_4$ amines, as clustered with the nitrate dimer and trimer, and (c) summed HOM concentration from $C_5+$ from hourly data. Units for ammonia + amines are normalised counts, as no calibration was performed. Event days include data across the full event day.


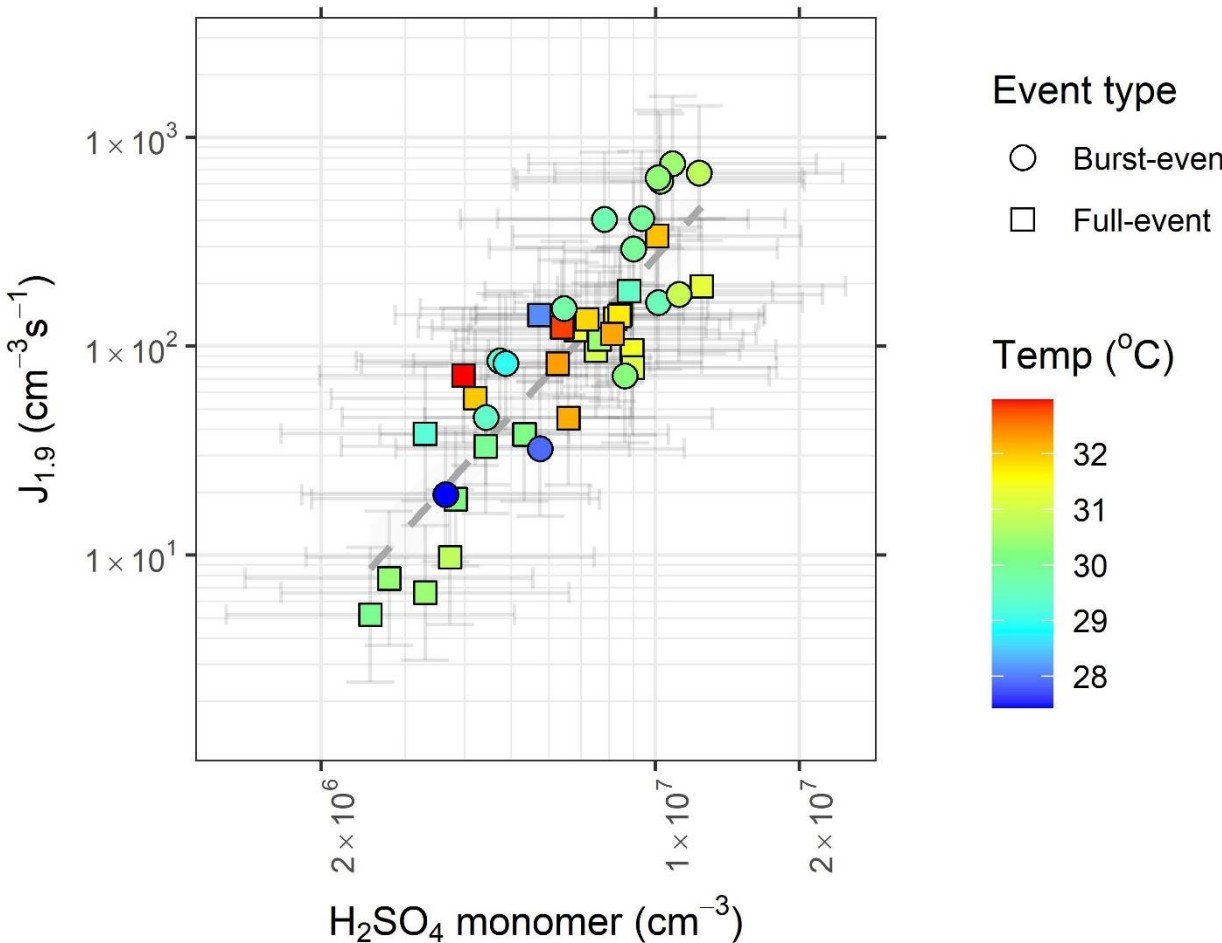


**Figure 4:** Formation rate ($J_{1.9}$) plotted against sulphuric acid monomer concentration, coloured by
condensation sink. Circles represent burst-events, squares represent full events. Data is for hourly
averages across NPF periods, typically within the hours 08:00 – 16:00. Slope of the line = $4.9 \cdot 10^{-5}$ s$^-$
$^1$. Error bars represent systematic uncertainties on [H2SO4] and $J_{1.9}$

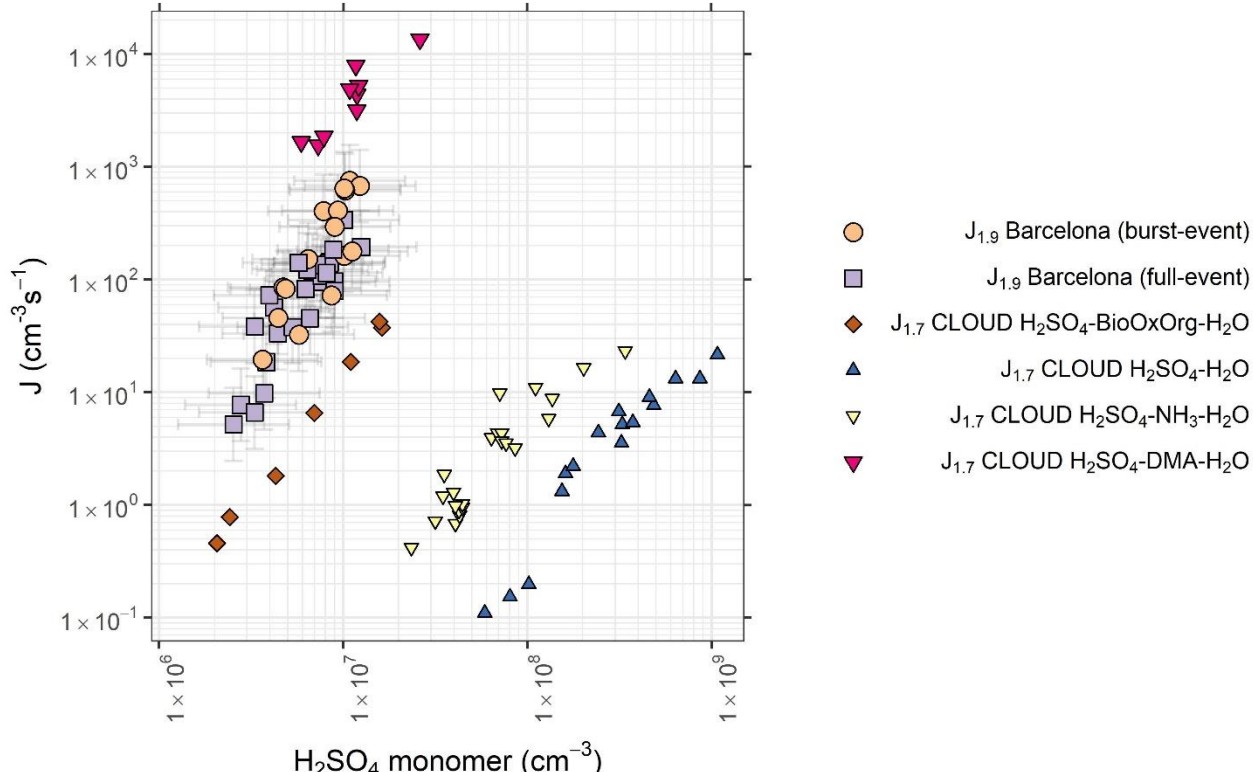


**Figure 5:** Formation rate plotted against sulphuric acid monomer concentration for data collected
from Barcelona. Tan circles represent burst-events, purple squares represent full events. as well as
that for the $H_2SO_4$-$H_2O$ (blue inverted triangles), $H_2SO_4$-$NH_3$-$H_2O$ (yellow inverted triangles),
$H_2SO_4$-DMA-$H_2O$ (pink triangles), and $H_2SO_4$-BioOxOrg-$H_2O$ (brown diamonds) systems from the
CLOUD chamber (Kürten et al., 2018 Kirkby et al., 2011; Riccobono et al., 2014). CLOUD
chamber experiments were performed at 278 K and 38 – 39 % RH. Data is for hourly averages
across NPF periods, typically within the hours 08:00 – 16:00. Error bars represent systematic
uncertainties on [$H_2SO_4$] and $J_{1.9}$.

1194
1195

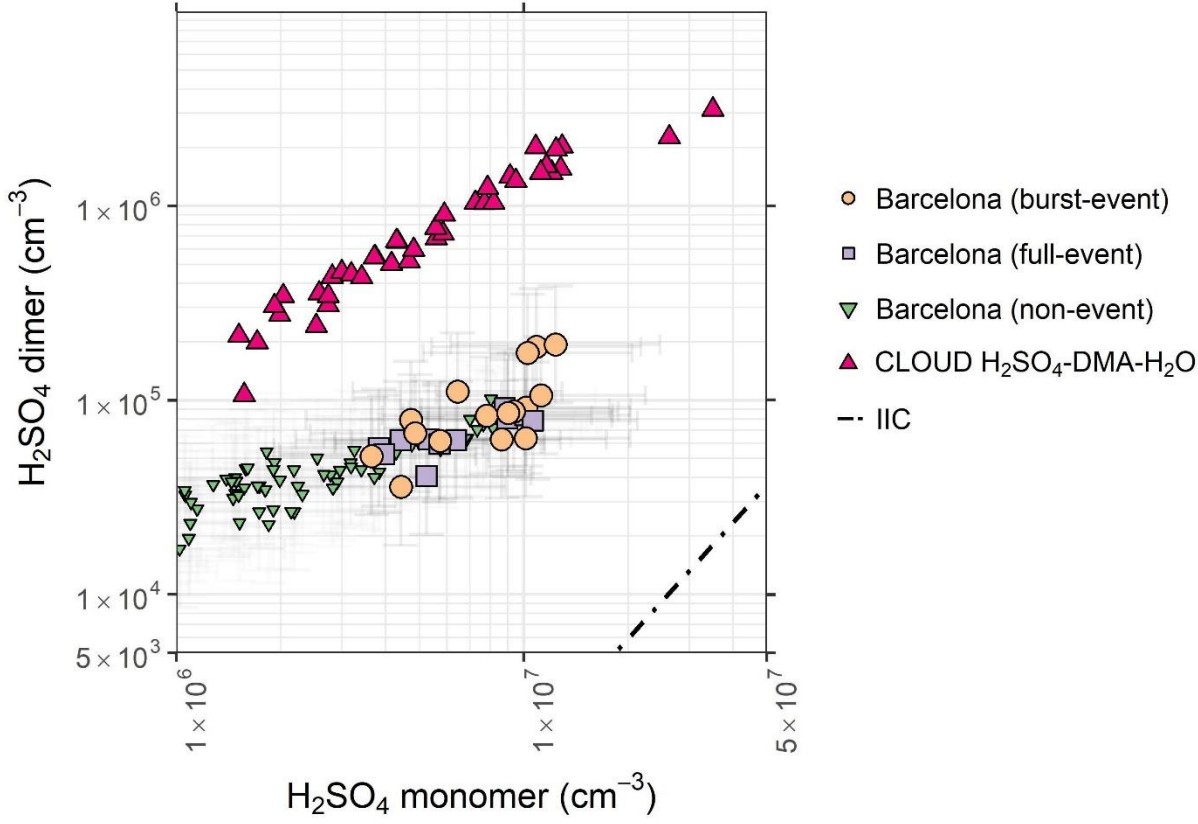

1196
**Figure 6:** Sulphuric acid dimer concentration plotted against monomer concentration, showing burst-event periods (tan circles), full event periods (purple squares), non-event periods (green inverted triangles), and the ratio of sulphuric acid dimer:monomer in the CLOUD chamber for the $H_2SO_4$-$H_2O$-DMA system (pink triangles) (Almeida et al., 2013). Dashed line represents the dimer concentration produced by ion induced clustering in the chemical ionization unit (Zhao et al., 2010). CLOUD chamber experiments were performed at 278 K and 38 – 39 % RH. Data is for hourly averages across NPF periods, typically within the hours 08:00 – 16:00. Error bars represent systematic uncertainties on $[H_2SO_4]$ and $[(H_2SO_4)_2]$.

1205

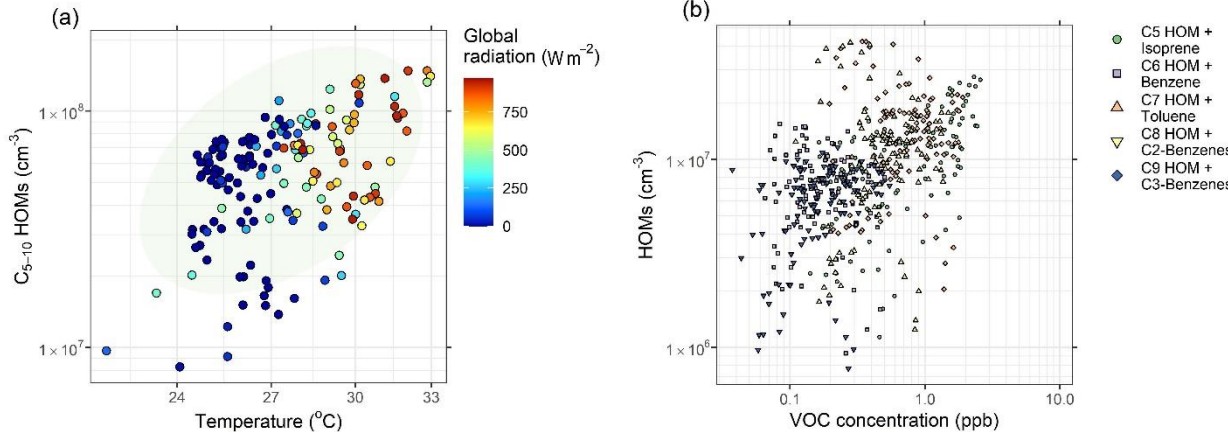

**Figure 7:** Influencing factors on HOM concentration, showing (a) $C_{5-10}$ HOM concentration plotted against temperature, coloured by global radiation. Ellipsis shows 95% confidence on a multivariate t-distribution. (b) HOM concentration by carbon number potted against parent VOC mixing ratio. These are segregated by carbon number/VOC, i.e, $C_7$ HOMs plotted against toluene, under the assumption that toluene oxidation is the main producer of $C_7$ HOMs. Time for both plots is of hourly time resolution.


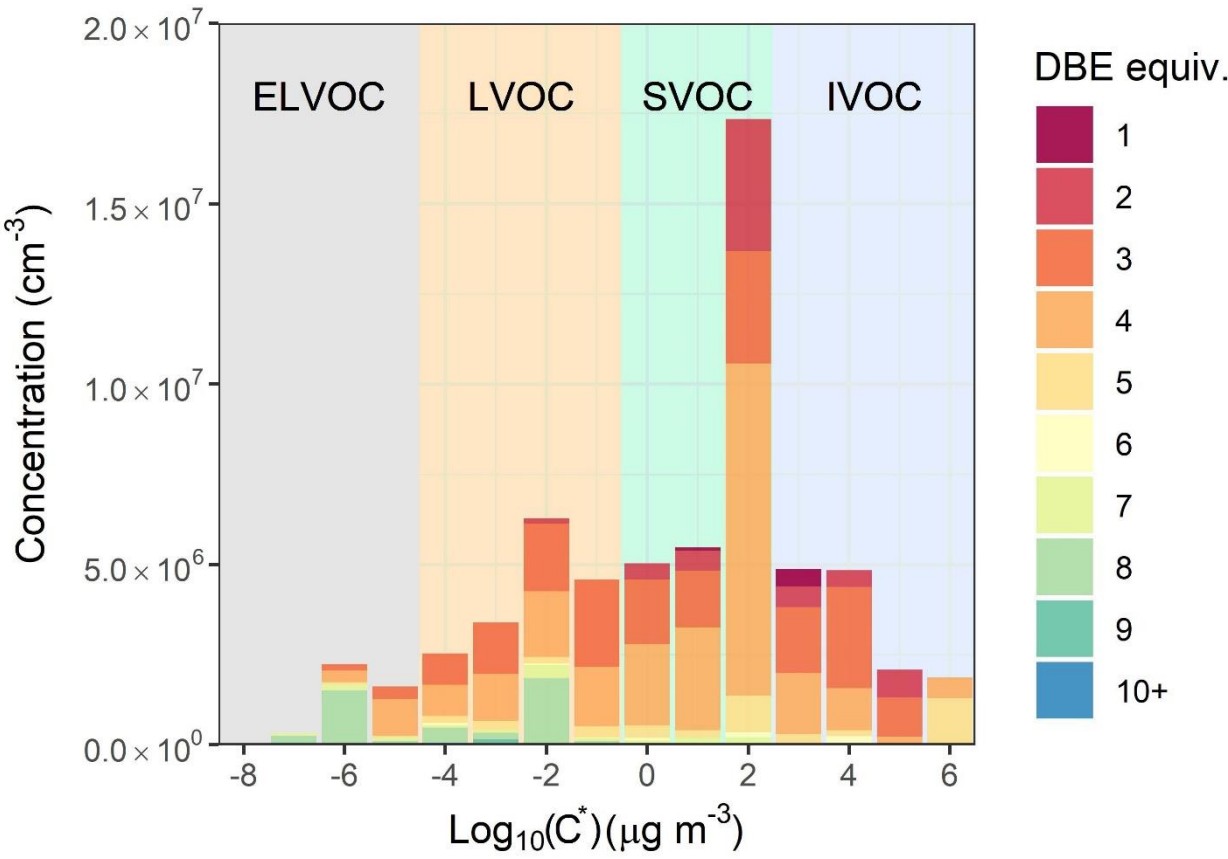


**Figure 8**: Concentrations of all oxygenated organic molecules and HOMs binned to integer $Log_{10}(C^*)$
values, coloured by DBE.

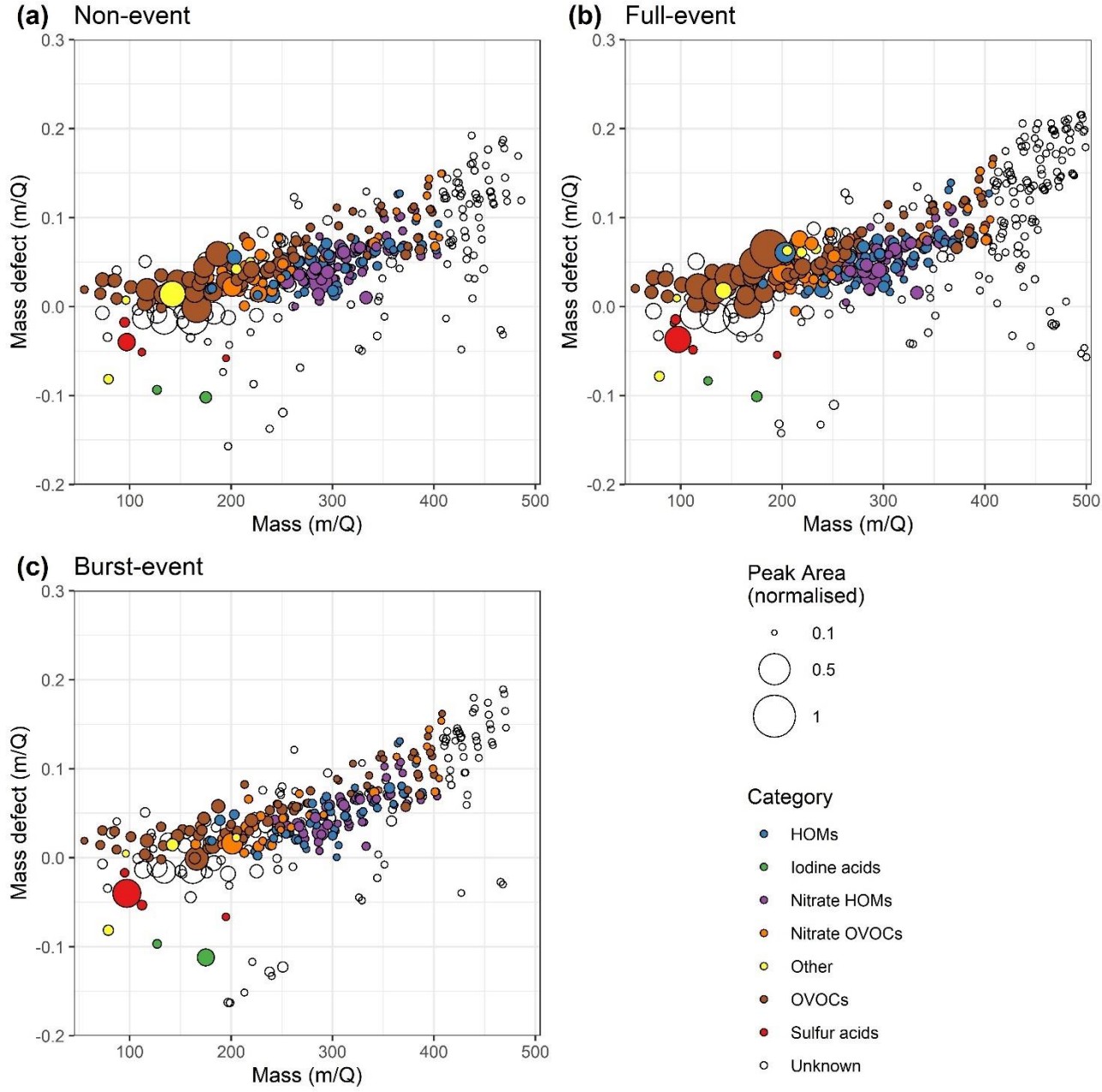


**Figure 9:** Mass defect plots for (a) non-event, (b) full-event, and (c) burst-event periods, data taken from 10:00 – 15:00 on the days 11/07/2018, 16/07/2018 and 15/07/2018 respectively. Size corresponds to mass spectral peak area. Ions are coloured according to identified chemical composition. *Blue* points correspond to HOMs containing all organic species with $\geq 5$ carbon atoms and $\geq 6$ oxygen atoms, and an O:C ratio of >0.6. *Purple* points correspond to the same but for species containing 1-2 nitrogen atoms. Species not meeting this HOM criterea were classed generally as OVOCs which are coloured *brown,* with the nitrogen containing OVOCs coloured *orange*. Sulphur acids (*red*) include ions $HSO_4^-$, $CH_3SO_3^-$ and $SO_5^-$, as well as the sulphuric acid dimer. Iodine acids (*green*) contains both $IO_3^-$ and $I^-$ (the latter presumably deprotonated hydrogen iodide). Unidentified points are left uncoloured.

1230

