# Peer review of "Molecular Insights into New Particle Formation in Barcelona, Spain"

_Atmospheric Chemistry and Physics, 2020_

## Referee Comment (RC1) · Anonymous Referee #1 · 3 Apr 2020

**General comments**

The authors present a set of data collected in summer 2018 in Barcelona, Spain. Using state-of-the-art instrumentation, they study new particle formation (NPF), analyzing particle formation rates, gas concentrations and composition of highly oxygenated organic molecules (HOM). The authors conclude that the NPF at the site proceeds via SA-H2O-DMA and/or SA-H2O-Organic nucleation, while HOM are needed for particles to grow beyond 10nm. The results are novel and valuable for understanding physics and chemistry of NPF in different environments. The text is well-written. However, few discussion and conclusions need to be addressed before paper can be published.

The paper starts very well with comparison of formation rates and sulfuric acid dimer-to-monomer ratios to chamber studies. In addition, gas concentrations and condensation sink on different days are compared. When discussion goes to composition of HOM, their sources, and their contribution to particle growth, it becomes somewhat out of focus, the reader is lost in HOM influence on NPF versus growth. Some conclusions are presented in much stronger way than the results can support. The text suffers from too many general comparisons, such as "relatively larger/smaller/low", "broadly similar", "broadly dependent", which makes the results less clear and leaves a lot of figure interpretation to the reader.

I suggest that authors 1) check their calculations, 2) provide a clearer (consistent) discussion and interpretation of HOM results and 3) rethink how final conclusions are presented regarding "molecular insights" in accordance with presented evidence. Please see specific comments below.

**Specific comments**

Major:

1. **Values for condensation sink (CS), J1.5, J5, growth rates (GR).**
There is a mistake in CS formula (line 181). The coefficient 4 should be 2, as in Kulmala et al. (2001).

J1.5 and J5 are very high, but in line (within some uncertainty) with calculated kinetic limit sulfuric acid nucleation (Kurten et al. 2018). However, during the observed NPF, the temperature is close to 30 degC. At this high temperature, I would expect lower formation rates. Please, check the values and provide a comment on this.

It would be beneficial, if more information was included how CoagS and GR were calculated (shortly from which instrument and with which method GR was calculated). Would it be beneficial to include GR values to results as important physical parameter of NPF?

2. **"Burst event"/ "full-event" discussion'**
The discussion what separates these event types is hard to follow for the following

reasons

1) In Fig. 2-6, how the data was produced? Are these values during certain part of the day or is whole day included? Are these hourly values? Please specify this in Methods or Results.

For instance, in terms of CS comparison, CS over whole day may not show any difference (lines 205-206, 209-210 and Fig. 2a). On the other hand, daily profile of CS may show that on "burst event" (Figure 1c), 100nm mode appears around 14:00, which I assume increases afternoon CS during that day.

2) The discussion about differences in meteorology is incomplete. In lines 246 and 397, it is mentioned that south and south-west air masses were responsible for NPF, while in Fig. S6, air masses on 3/5 NPF days came from North. Could change in air mass (wind direction) be responsible for no growth during "burst events"? Please clarify.

3) Please provide information on which day, what type of NPF was seen.

3. **Discussion about HOM sources/formation/influence on particle growth**
The presentation and discussion on HOM results is very valuable. To make the section 3.3 more clear, I suggest to separate it in two subsection: 1) HOM composition and sources and 2) How HOM change on different days in relation to NPF. The following are my main concerns regarding HOM discussion:

1) Interpretation of Figure 7. Are there really dependencies seen?

Fig. 7a to me shows correlation between T and global radiation. If night and day are separated, there is no dependency on T, or is there?

In Fig. 7b, dependency is discussed when all points are taken together. But is there a point to do that if different groups of HOMs are plotted against different VOCs? For C7/C9 I don't see any dependence.

Therefore, the conclusion that HOM has "strong dependence of their concentration on

both temperature and VOC concentration" (line 394) is not supported.

In Fig. 7a HOM C5-10 are plotted, while in Fig. 7b, only C5/C7 /C9 are shown. What is the reason for excluding for instance C6,C8,C10?

Please clarify this section and harmonize the text (abstract-discussion-conclusions).

2) Sudden statement in conclusion regarding HOM effect on NPF and growth:

In lines 395-399, it is stated "aromatic organic compounds are more important compounds which initiate new particle formation and growth than oceanic emissions". Which data leads to this conclusions? Especially, the part in which aromatic compounds initiate NPF. In line 377, it is actually mentioned that biggest difference between the two event types are in fact C9/C10 compounds. Please clarify this aspect.

3) Please clarify what is the effect of HOM on NPF vs growth, if such can be distinguished from the data.

4. **Strong conclusions regarding pathway for NPF**
Based on the results from the analysis, the results are not as conclusive as claimed both by title ("molecular insights") and in the text (slightly different in abstract/discussion/conclusions). I would, therefore, suggest to rephrase the statements in text.

line 111: related to problem with title: "we examine molecular level evidence ... at the critical diameter"- even the authors state "cluster identity cannot be identified" (line 276), so this wording is not appropriate.

line 293: It is clear that the results are similar to Kurten 2016 (except really high J). Mind that they conclude that SA-H2O-DMA or other pathway could not be confirmed with their data.

lines 301-302: I suggest softening this conclusion. Also check abstract (lines 38-39) for consistency.

[Figure]

line 387: The first and main conclusion is that NPF "plausibly proceeds by the formation of clusters involving sulphuric acid and highly oxygenated organic molecules, with likely involvement of bases". You clearly state earlier in line 276 that cluster identity cannot be directly identified and suggest that HOM is important for growth (not NPF, as far as I understood from text). So I am not sure what "plausibly" refers to. Please rephrase to reflect that results are not so conclusive, but the hypothesis is as follows (. . .).

Minor:

line 63: I appreciate the use of very recent references, but surely there are older references/review that appropriate to be referenced here (from other groups).
line 74: near-kinetic limit is achieved at certain DMA concentrations, can be specified here (Almeida et al. 2013).
line 143: In Brean et al. 2019, calibration coefficient of 7e8 cm-3 was used. Were any further adjustments done in this work to get 3e9 cm-3?
lines 216-217: Sulfuric acid time series/dial profile are discussed, which is not shown anywhere. Consider including time series of measured parameters in the Supplementary.

Figure 3: Was it possible to reliable fit amines and ammonia from the spectra? Kurten et al. 2016 mentions that at high RH, nitrate-water cluster interferes with the fitting of C2 amines. Please comment and preferable provide a figure of the peak fitting. Please also include Kurten et al. 2016 into references when talking about detecting ammonia with nitrate CI-APi-TOF.

lines 234-236: You probably mean to say that TMA won't efficiently cluster with nitric acid and won't be detected. It is not very clear, please rephrase.
line 267 and Fig. 5: I think CLOUD data presented here is J1.7 for 278 degC, isn't it? Please mention it in the text and/or caption.
line 282-283: "Sulphuric acid dimer roughly represents the strength of sulphuric acid

clustering in the nitrate CI-APi-TOF". - I am not sure what this means. Did you mean it tracks atmospheric clustering?

line 287: "lower limit" – do you mean upper? Please explain how IIC was calculated, or was it taken directly from Kurten et al. (2016)? I would like to note that the reaction time in drift tube in Kurten et al. (2016), is 50ms, what is it in your instrument?

line 318: Is this a relevant reference for C10 HOMs that are coming with inland air mass?

lines 323-328: a series of unconnected sentences. please rephrase to make more logical. "condensational growth being a reversible, step-wise kinetic process" – I am not sure what you mean, please rephrase. Do you mean "higher HIO3 cannot account for stopped growth beyond 10nm?. rephrase it to fit in HOM discussion or move it to section 2.2.

In line 334, it is said that "precursors for these HOMs are presumed to be largely isoprene, alkylbenzenes and monoterpenes". However, in abstract (line 40) and in conclusions (line 394), isoprene is lost.

line 346: include also isoprene reference.

line 369: you mention that HOM>500 m/Q are increased during "full-events", however no data above 500m/Q is shown. Or do you mean HOM within 400-500 m/Q? This has to be clarified as this is also mentioned in conclusions. It can be shown as time series (or dial) of the sum of unit mass resolution peaks above 500 m/Q or box plot (I understand high masses are hard to fit/identify).

line 389-390: "multiplicity of mechanisms has been shown to occur in chamber studies but has not been observed in the real atmosphere previously."- for instance Yan et al. (2018) showed two regimes for NPF at SMEAR II station, Finland. Kurten et al. 2016 shows very similar results to this study and analogously (lines 390-391) points out high J, similarity to CLOUD for both SA-DMA-H2O and SA-OxOrg studies. Please rephrase.

Figure 7: Why HOM is shown in normalized signal, while in Fig. 3 it is in cm-3? How many days and in what time resolution are the data in this plot? Caption: "Influencing

factors on VOC concentration" – on HOM concentration?

Figure 8: Please provide a list of identified peaks in Supplementary and please explain what are the HOM observed at m/Q below 200 (quite high signals). Are they deprotonated species fitting with the "HOM" criteria?

I suggest the authors would look into following and add more quantitative interpretation of results in the text.
-line 44: "significantly lower" – here easy to specify exactly how much, a factor of X.
-line 87-89: phrase "despite (extremely) high condensation sinks" repeats twice in a row. For comparison here, it would be nice what is meant by "extremely high".
-line 100: please specify what is "frequent", 40-lines 101-105, 195: "relatively high ozone", "relatively low ozone (high compared with the rest of the year)"-can you rephrase or provide values as example? I also got lost in logic related to "relatively high" ozone during NPF and "maximum ozone episodes" with no NPF.
-line 244-245: "HOM ... greatly enhanced", "lower during burst". Please provide values or a factor for HOM difference between days.
-line 248: "...HIO3 and ...MSA are low". I assume you refer to Fig S5d and e here? Relative to what are these concentrations low?
-line 270: instead of "extremely high" provide actual numbers from Fig. 5.
-line 271: what does "broadly similar" mean?
-line 349: what does "broadly dependent" mean?

**Technical corrections**

line 125: "interface" with small i.
line 222: remove empty line
line 333: "temperature plotted against the signal of HOMs". Isn't it the other way around? Also see caption to Fig. 7.
line 368: insert: "enhancement" in comparison "to smaller alkylbenzene derived

HOMs".

line 401: repetition of line 394-395.

Figure 8. In-text (line 355), you mention that Fig. 8 has same days as Fig. 1. On Fig 1, there is 11/07/2018, while caption to Fig. 8 says 12/07/2018. Is it a typo? Please fix.

References:

Almeida et al. 2013. Nature. 502,Âă 359–363, https://doi.org/10.1038/nature12663

Brean et al. 2019. Atmos. Chem. Phys., 19, 14933–14947. https://doi.org/10.5194/acp-19-14933-2019

Kulmala et al. 2001. Tellus, 53B, 479–490

Kurten et al. 2016. Atmos. Chem. Phys., 16, 12793–12813. https://doi.org/10.5194/acp-16-12793-2016

Kurten et al. 2018. Atmos. Chem. Phys., 18, 845-863. doi:10.5194/acp-18-13231-2018 https://doi.org/10.1029/2018JD029356

Yan et al. 2018. Atmos. Chem. Phys., 18, 13231-13243. https://doi.org/10.5194/acp-18-13231-2018

---

## Referee Comment (RC2) · Anonymous Referee #2 · 23 Apr 2020

The manuscript "Molecular Insights into New Particle Formation in Barcelona, Spain" by Brean et al. discusses the observations of summer-time new-particle formation (NPF) events in Barcelona, Spain. A wide scope of capable instrumentation was deployed, allowing for deriving a decently comprehensive picture of the mechanisms that likely facilitated observed NPF events. Consequently, premise and results of this study are certainly of interest for the ACP audience, and in principle I would like to recommend their publication. However, I have substantial concerns regarding presentation and conclusions, which I feel need to be addressed.

First off, the introduction is pleasant to read, and well structured overall (except for some technicalities, see minor/technical comments below). I also only have minor/technical comments on Section 2 (Methods). My main concern starts with the

interpretation of the "burst events" in Section 3.

MAJOR COMMENTS:

1) "Burst" events and air mass histories It is well established that NPF events for which formation followed by continuous growth over many hours can be observed (here called "full event"), are observable as such at a point site because NPF takes place more or less in sync across a fairly wide area (i.e. regional-scale NPF). Therefore, I wonder if the so-called "burst events" could be due to NPF that occurs more localized? I.e., that the more localized character of the NPF is why growth beyond 10 nm is not observed (with smallest observed particles having formed closest-by, whereas the largest observed new particles would have formed farthest away within the NPF area, upwind). The time resolution of the particle sizing measurements was quite low, but an estimate of the overall growth rate of the newly-formed particles could anyway be made, and from that, plus wind back trajectories, even the size of the hypothesized local NPF area could be estimated. Related to that: The manuscript mentions at least twice that NPF events, both "full" and "burst" types, are associated with "southerly and south-westerly air masses" (e.g. L 396-399). But Fig. S6 (showing back trajectories) clearly contradicts that statement! I suspect the mistake is in the text? That suspicion is (a) because I would expect a regional-scale BVOC-HOM-driven NPF event to have air masses NOT arriving from sea+town, and rather from inland, as suggested, presumably, in Fig. S6; and (b) upon noticing that the burst event shown in more detail (e.g. Figs. 1 and 8; July 15) is indeed the only event that does actually show a southerly back trajectory in Fig. S6, which puts the trajectory mostly over the ocean, with the exception of its final path across town (and, possibly critically, over shipping, as the authors also point out) towards the measurement site.

2) Number of events and data points Line 204 states that there was only 2 "burst" events observed. Then, it would be very useful to also learn how many "full" event were observed (and how many were non-event days)? However, with only 2 samples for the burst events, how can Figs. 2 and 3 include a boxplot for conditions during those

two events? That may be down to the too sparse explanation of what data were used for these boxplots. E.g., data from which times were used in each event class (a vs b vs c)? Is the boxplot based on event-wise averages, or something else? But then again, Figs. 4+ suggest there is many more events of each type (judging from # of markers). A Fig. S6 again suggests again there's only been 5 events total!? I have some suspicions, but overall I am left rather confused. So, all that clearly should be presented more clearly, i.e. data quantity and usage in figures, resolved by event type.

3) Section 2.2: Was "sensitivity" applied to any detected compound? From the rest of the paper I conclude that it was applied to $H_2SO_4$ and HOMs, but not to ammonia and amines. Overall, some more detailed discussion of concentration quantifications based on CIMS signals would be interesting. See also next comment.

4) Ammonia and amines: Section 2.2: Are there any estimates on detection limits or sensitivity regarding ammonia and amines? (And which amines are expected to be detectable?) There is some discussion of amine detection late in Section 3.1. But it would be helpful if that more general instrumental aspects of their detection would already be at least mentioned in the Methods section. Section 3.1: Which amines were actually detected? I think that information is only provided in the caption of Fig. 3 and in Fig. S1 (implying $C_2H_7N$ and $C_4H_{11}N$), and that should be mentioned also in the main text, and much earlier. (And is that in agreement with expectations from amine abundances and amine-specific sensitivities indicated by previous studies (if any)?) The authors do discuss relative sensitivities later in Section 3.1, implying that sensitivities to all those small bases correlate to NPF enhancement potential. That argument could be brought sooner. Figure 3, and related discussion (Section 3.1, page 9): From the beginning of the discussion of Fig. 3b, I have been wondering why are ammonia & amine signals only presented as a sum of all signals? Even if the concentrations of these compounds could not be quantified in these measurements, it still appears it would be more insightful not to lump them together. Especially given the respective differences in NPF enhancement capabilities. Also, if lumped together,

there is the chance that differences in atmospheric abundance of ammonia vs. amines (almost certain, as also described in Section 3.1) or in instrument sensitivity to these compounds (plausible) lead to one compound (or group of compounds) controlling the lumped signal... Only very late in Section 3.1, L 241 mentions that all the bases' time series correlated quite well, thus justifying the lumping. That argument for lumping should be made more clearly and earlier. And supported by a figure or two in the Supplement. L 217: From looking at Figure 3, I am not sure I would also arrive at the conclusion that ammonia+amine concentrations were enhanced for full events. The medians are practically identical. Have the authors applied some quantitative measure of statistical significance regarding differences of the parameters presented in Figs. 2-3?

5) Figure 5 and/or related discussion Environmental conditions should be mentioned and discussed, in particular in light of the CLOUD experiments being compared to: T, RH, compound concentrations (as known), as all those will have affect formation rates. Also: given available parametrizations of NPF rates as a function of sulfuric acid, HOM (or "BioOxOrg") and amine/ammonia concentrations, are the authors able to explain observed NPF rates and infer concentrations of either involved BioOxOrg or amines? Combined with expected amine concentrations and measured VOC and HOM concentrations, some closure could be attempted. Some hand waving may be necessary, but the attempt could be quite interesting. And it could strengthen the paper, especially if it can be argued that closure ∼works out. Similarly, growth rates of new particles could be estimated (see also 1st comment) and evaluated against HOM concentrations.

6) Figure 6 and related discussion (in particular L 289-293): Could the SA dimer signal also be affected by instrument settings? It is conceivable that some instruments or settings would fragment a certain fraction of SA dimer ions (that is, at some point after their formation by NO3-ionization), and that that fraction is instrument- or tuning-specific. If so, conclusions can likely still be made anyway from comparisons between

measurements within the same campaign (provided settings remained the same), and probably even from comparisons across campaigns/longer time periods as long as the same instrument was used. But could such instrumental differences cause (part of) the discussed discrepancies between dimer signals here and results shown from other field and lab experiments? My feeling is that the SA dimer anion is stable enough that such instrumental fragmentation should not be expected, but I would ask the authors to at least point that out (i.e., if my "feeling" can be defended based on previous studies – I apologize for not remembering expected cluster stabilities vs the fragmentation potency of the APi-TOF instrument), OR recognize/discuss potential issues.

MINOR COMMENTS:

Abstract: a) I would already here mention roughly the most important measurement methods used for the study. No details, and a side sentence may be enough. Especially as there is talk about sulfuric acid monomer and dimer concentrations, I would have liked to be informed already here that those are based on NO3-CIMS measurements. b) Growth beyond 10 nm: my immediate thought already here was that limited growth could also be a sign of more local NPF (vs. regional, i.e. on a larger geographical scale), in which case the observed lower concentrations of low-volatility organics could be irrelevant (see 1st major comment). So I would already here, in a compact way, give the reasoning for claimed conclusion.

Introduction: The last paragraph (review of NPF observations in the Barcelona area) is a somewhat confusing to read. Should be restructured for clarity.

I found the usage of the term "background" not clear (end of introduction, beginning of methods, beginning of results).

L 143: The meaning of "sensitivity" here (in previous studies called also "sensitivity coefficient" or "calibration factor") remains a mystery for any reader who is not fairly intimate with operation of that CIMS instrument. So, I would at least cite some paper where that meaning is discussed, e.g. Kürten et al., 2012 (10.1021/jp212123n). Regarding the "sensitivity" value used, the authors cite here a previous calibration, paper also led by UBirmingham. However, I could not find, in that paper, how that calibration was performed. Indeed that paper contains the statement (in their section 2.2) "No sensitivity calibration was performed for these measurement..." Bottom line is that I have remained wondering where the used value (3e9 cm$^{-3}$) derives from. (Or simply from comparison to the sulfuric acid proxy?)

L 221: It is not clear if this is a general statement regarding amine concentrations, or specific to observations (which, however, I understand could not be quantified, so I assume it's the former?)? Should be clarified.

Figure 1 is lacking all labels for the axes. At least the somewhat-less-obvious vertical axis should be labelled.

L 244: Should start new paragraph when starting discussion HOMs.

L244-245: It appears odd to discuss HOMs (Fig. 3b) with only 1 sentence, following a page of discussion of ammonia/amines! Should at least add a reference to a later section, where organics, including HOM, are being discussed.

L247-248: This last sentence of Section 3.1 is very vague. How high (or low) are those concentrations of marine compounds? (Or estimated to be?) What observations would have been considered evidence FOR an influence of oceanic emissions on NPF/growth?

Fig. 4: How steep ist that slope? That is usually interesting information, at least for comparing with other studies.

L 261: Does "losses" refer to amines? If so, I don't see how photochemical reductions of amine mixing ratios could mask an actual dependence of NPF rates on those mixing ratios. (If I caught the inference correctly.)

L 273: Suggest rephrasing to make it clear immediately that the discussion shifts from literature results to new results. And it is not clear which observations the last part of

the sentence refers to ("on these days").

L 282-283: ambiguous what is meant by "strength of sulfuric acid clustering". It becomes clear thereafter, but if I understand correctly, "strength" is not the right word.

L 293+: Could the point raised in major comment (6) explain the flatter slope observed here vs. the Germany observations?

L 337: Getting confused here. Should it read "not largely radiation dependent" instead of "not largely temperature dependent"?

L 342: Should they be transportent FROM inland by the land breeze?

L 357, 361: Should be explained what is meant by "detailed criteria", and by "updated criteria" (i.e. what are the respective criteria).

L 367: could be informative to point out some of those formulae explicitly

L 371-372: would be instructive to be more specific regarding "large" and "smaller"

L 378-382: Something went wrong with this sentence, especially the first part. Think I get the idea, but not sure.

L 381: Please state the size range the ACSM is sensitive to

L 393: Could the mechanisms also support each other (i.e. be combined) rather than be in competition?

Fig. S6: Please include information on which kind of events are shown, and when (... see also major comments (1) and (2))

TECHNICAL COMMENTS

L 61: i.e. should be e.g.?

L 79/80: those "loss processes" haven't been mentioned yet, so would be instructive to name the most important ones (here, for < 50 nm particles)

L 81: "at these diameters" ... rather "from these diameters onward"

L 86: I believe that range should read "0.5-11" (i.e., 11 instead of 1.1)

L 86-90: two redundant consecutive sentences. combine.

L 138: if I remember correctly, the flow containing the reagent ions is not "guided into the sample flow", but rather only the ions are guided there (electrically).

L 162: redundant mentioning of "4 flows"

Fig. 3, a and b: the "+" in the exponents (tick labels) are conventionally omitted

L 371: compromise -> comprise

Figs. S6 and S7 are not referred to in the main text (not sure if that's a problem).

---

## Short Comment (SC1) · 6 May 2020

Since molecular level knowledge of new particle formation in urban areas is still very poor I read this paper with high interest. I strongly support the reviewers asking for a better description of the analysis methods and more quantitative information. For example in Figure 1 b no distinct particle evolution is seen. This looks more like an advection of an air mass. It would be worthwhile to show in an example how the nucleation rate was determined? The same applies to the growth rate calculation. GRs seem to be quite high but time resolution of measurements rather low for such events. For example, the time resolution of the PSM measurements is 10 min. Thus for GR>6 nm/h no growth rate in the sub-3nm range can be determined. It is hard to believe that J5 and J1.5 are almost similar. At H2SO4 = 2E06 the reported J5 is even ten times

higher than J1.5. What are the uncertainties of J1.5, J5 and GR at different sizes? As pointed out by Kulmala et al. (2017) nucleation and growth rate are connected. It would be very helpful for the reader if the authors would also report the growth rates, as they have been determined anyhow. Kulmala et al. (2017) also show that the survival probability of small clusters becomes very low at such high coagulation/condensation sinks as reported in this study. A discussion of this phenomenon should be included. The authors strongly stress the role of organics in NPF. Measurements of the HOMs and their chemical composition are available. Thus, the authors could estimate if the concentration of very low volatility HOMs is high enough to account for the growth of few nm-sized particles. Such an analysis would support their conclusions on what drives NPF. In Figure 8 it is hard to see differences in the mass spectra at m/z>250. Thus, most differences are at lower m/z and for compounds with high mass defect, that is low oxygen content, and thus high volatility. The authors conclude: "We show new particle formation rates in Barcelona are linearly dependent upon the sulphuric acid concentrations, and this mechanism plausibly proceeds by the formation of clusters involving sulphuric acid and highly oxygenated organic molecules, with likely involvement of bases". Where do the authors see the clusters between organics and sulfuric acid in Figure 8? Figure 5 shows the H2SO4/DMA nucleation rates from CLOUD by Almeida et al. This data has been revised by Kürten et al. 2018. The new values would be at least an order of magnitude higher than Barcelona, which could eventually be explained by the higher ambient temperature in Barcelona. From Figure 7a the authors claim a temperature dependence of HOM formation. However, the higher HOM concentrations at high temperature are also accompanied by higher global radiation. Thus, this dependence could just represent day-night time chemistry and the dependence on OH concentration. A well documented report of urban NPF events is highly valuable for further comparison with laboratory and other field studies. Thank you for considering this short comment. Sincerely, Josef Dommen

Kulmala et al., Faraday Discussions 2017, DOI: 10.1039/c6fd00257a Kürten et al., Atmos. Chem. Phys., 18, 845–863, 2018

---

## Author Response (AR1)

Journal: ACP - MS No.: acp-2020-84
Title: Molecular Insights into New Particle Formation in Barcelona, Spain
Author(s): Brean et al.

**RESPONSE TO REVIEWERS**

We thank all reviewers for their insightful comments on our work and are pleased to respond as below:

Note: Referee comments in this text will appear in blue, responses in black, and where new text from the manuscript is quoted, it will appear in red, and **bold** where text has been inserted into a larger section

**Referee 1**

*Major comment 1: Values for condensation sink (CS), J1.5, J5, growth rates (GR). There is a mistake in CS formula (line 181). The coefficient 4 should be 2, as in Kulmala et al. (2001).*

*J1.5 and J5 are very high, but in line (within some uncertainty) with calculated kinetic limit sulfuric acid nucleation (Kurten et al. 2018). However, during the observed NPF, the temperature is close to 30 degC. At this high temperature, I would expect lower formation rates. Please, check the values and provide a comment on this.*

*It would be beneficial, if more information was included how CoagS and GR were calculated (shortly from which instrument and with which method GR was calculated). Would it be beneficial to include GR values to results as important physical parameter of NPF?*

**Response 1:** Equation 14 in Kulmala et al., 2001 states $4\pi D$, this is consistent across more recent publications also (Kulmala et al., 2012). We have included a reference to the latter just before this equation also.

We agree our formation rates are very high – these calculations have been double checked for clarity. We note, however, that such high formation rates at high temperatures have been observed previously, examples being from Mexico City ((Iida et al., 2008; Kuang et al., 2008), temperatures of 19.2 and 24.1 °C are quoted in the former, high formation rates relative to sulphuric acid are quoted in the latter. Similarly, Kürten et al, (2016) report high formation rates on days with peak temperatures just below 25 °C. Measurements in Beijing show a rather sharp temperature dependence, but particle formation rates are shown to reach $10^2$ cm$^{-3}$ s$^{-1}$ at temperatures similar to our own at comparable sulphuric acid concentrations (Yu et al., 2016). We include the following discussion in the text addressing this.

*"Model studies of sulphuric acid-amine nucleation show a decline in nucleation rate with temperature (Almeida et al., 2013; Olenius et al., 2017), as the evaporation rate of sulphuric acid-amine clusters will increase with temperature (Paasonen et al., 2012). Conversely, evaporation rates of such small clusters, and resultant nucleation rates tend to increase modestly with increases in relative humidity, most pronounced at lower amine concentrations (Almeida et al., 2013; Paasonen et al., 2012). Despite this, high nucleation rates at temperatures nearing 300 K have been reported previously (Kuang et al., 2008; Kürten et al., 2016), although these tend to show a temperature dependence (Yu et al., 2016)."*

We also utilise updated $H_2SO_4$-DMA-$H_2O$ data in our Figure 5 from Kürten et al. (2018), where we show our measurements fall short of the J values from the CLOUD chamber experiments, and thus we edit our conclusions accordingly and suggest the discrepancy arises from temperature.

*"while formation rates far exceed that of $H_2SO_4$-BioOxOrg-$H_2O$ nucleation, they fall short of those of $H_2SO_4$-DMA-$H_2O$ nucleation at 278 K, as does the sulphuric acid dimer:monomer ratio, possibly explained by cluster evaporation due to high temperatures in summertime Barcelona (303 K during events), and limited pools of gas-phase amines. These results are similar to reports of nucleation rates in rural Germany (Kürten et al., 2016). As no higher-order clusters were directly measured, we cannot determine nucleation mechanisms with certainty, and an involvement of HOMs in nucleation is plausible."*

[Figure]

Further to this, we present an expanded methods section

*"$J_5$ was calculated using the data between 5-10 nm, and $J_{1.9}$ was calculated using the measurements between 1.9 − 4.5 nm. We also calculated $J_{1.9}$ from our NanoSMPS data, employing the equations of Lehtinen et al. (2007). $J_{1.9}$ from both methods showed reasonable agreement ($R^2$ = 0.34). Agreement between $J_5$ and $J_{1.9}$ for each method was similar ($R^2$ = 0.37 and $R^2$ = 0.38 for $J_{1.9}$ calculated from PSM data and from Lehtinen et al. (2019) respectively) $J_{1.9}$ is greater than $J_5$ as predicted from equation (2) by around a factor of 20. See Kulmala et al. (2001) for more information on calculation of coagulation sinks and formation rates. Growth rates between 4.5 − 20 nm were calculated according to the lognormal distribution function method (Kulmala et al., 2012), whereas those between 1.9 and 4.5 nm were calculated from PSM data using a time-delay method between PSM and NanoSMPS data."*

We also include GR values in section 3.1

*"$GR_{4.5-20}$ ranged between 2.47 − 7.31 nm $h^{-1}$ (4.69 ± 2.03 nm $h^{-1}$), $GR_{1.9-4.5}$ ranged between 3.12 − 5.2 nm $h^{-1}$ (4.36 ± 1.02 nm $h^{-1}$)."*

***Major comment 2: "Burst event"/ "full-event" discussion***

*The discussion what separates these event types is hard to follow for the following 2 reasons*

*1) In Fig. 2-6, how the data was produced? Are these values during certain part of the day or is whole day included? Are these hourly values? Please specify this in Methods or Results.*

*For instance, in terms of CS comparison, CS over whole day may not show any difference (lines 205-206, 209-210 and Fig. 2a). On the other hand, daily profile of CS may show that on "burst event" (Figure 1c), 100nm mode appears around 14:00, which I assume increases afternoon CS during that day.*

*2) The discussion about differences in meteorology is incomplete. In lines 246 and 397, it is mentioned that south and south-west air masses were responsible for NPF, while in Fig. S6, air masses on 3/5 NPF days came from North. Could change in air mass (wind direction) be responsible for no growth during "burst events"? Please clarify.*

*3) Please provide information on which day, what type of NPF was seen.*

**Response 2:**

1) The periods of data that have been included have been clarified in the following sections

- Figure captions for Figures 2 + 3 include the sentence *"Event days include data across the full event day."*
- Data coverage is a little misleading. Figures 4 – 6 have been amended for consistency, containing the phrase *"Data is for hourly averages across NPF periods, typically within the hours 08:00 – 16:00."*

Further, we include a new Figure S2 showing the diurnal evolution of condensation sinks on non-event days, as well as both types of event days, alongside this we present an extended discussion where we include the following

*"Figure 2 contains box plots showing condensation sink, temperature and global radiation for all 3 NPF types **across the entire day (diurnal profiles plotted in Figure S1). Condensation sinks during NPF periods of both types (Figure 1(b) & 1(c)) were not significantly lower than in non-event periods. Condensation sinks were supressed prior to the beginning of an event for full-events, increasing relative to non-events through the afternoon period. Of the two burst-events, one was similarly characterised by a suppression to condensation sink, whereas the other showed a sharp rise in the midday"***

2) This discussion was based upon an analysis of local wind direction + speed measurements. From polar plots it was inferred that air masses arising from the south were dominant during NPF; however, we find the air mass history from HYSPLIT to be more informative to our analyses, and thus these were included. The wind direction and velocity across the burst event days is shown below, where it can be seen that air masses are stable following the appearance of the new mode, and persist across the period of multiple hours, giving us confidence that this is not due to air mass advection.

[Figure]

3) This information is now available in Figure S8, where air mass history is presented by day, with each class of day stated in the figure caption.

*Major Comment 3: Discussion about HOM sources/formation/influence on particle growth*

*The presentation and discussion on HOM results is very valuable. To make the section 3.3 more clear, I suggest to separate it in two subsection: 1) HOM composition and sources and 2) How HOM change on different days in relation to NPF. The following are my main concerns regarding HOM discussion:*

*1) Interpretation of Figure 7. Are there really dependencies seen?*

*Fig. 7a to me shows correlation between T and global radiation. If night and day are separated, there is no dependency on T, or is there?*

*In Fig. 7b, dependency is discussed when all points are taken together. But is there a point to do that if different groups of HOMs are plotted against different VOCs? For C7/C9 I don't see any dependence.*

*Therefore, the conclusion that HOM has "strong dependence of their concentration on both temperature and VOC concentration" (line 394) is not supported.*

*In Fig. 7a HOM C5-10 are plotted, while in Fig. 7b, only C5/C7 /C9 are shown. What is the reason for excluding for instance C6,C8,C10?*

*Please clarify this section and harmonize the text (abstract-discussion-conclusions).*

*2) Sudden statement in conclusion regarding HOM effect on NPF and growth:*

*In lines 395-399, it is stated "aromatic organic compounds are more important compounds which initiate new particle formation and growth than oceanic emissions". Which data leads to this conclusions? Especially, the part in which aromatic compounds initiate NPF. In line 377, it is actually mentioned that biggest difference between the two event types are in fact C9/C10 compounds. Please clarify this aspect.*

*3) Please clarify what is the effect of HOM on NPF vs growth, if such can be distinguished from the data.*

**Response 3:**

We have separated these sections as suggested, and provide the following answers below

1) If we take just the daylight hours, or hours where insolation is $>100$ W m$^{-2}$, the correlation between temperature and signal gets slightly better $R^2 = 0.29 \rightarrow R^2 = 0.30$. Taking the periods where insolation is $<100$ W m$^{-2}$ then there is no correlation seen. However, the correlation between HOMs and insolation is poor ($R^2 = 0.15$), and we maintain that there exists a dependence of HOM concentration on temperature within this dataset.

We excluded $C_6$, $C_8$ and $C_{10}$ from this plot so as to not overcomplicate the image. In the case of $C_{10}$ also, we measured $C_{10}$ aromatics and monoterpenes at approximately a 1:3 ratio, and these will both produce oxidation products of similar chemical formulae, thus including $C_{10}$ in the plot comes with some uncertainties. However, we produce a new version of the plot including isoprene, benzene, toluene, $C_2$ aromatics and $C_3$ aromatics. We concur with the skepticism about our conclusions from these figures and have updated this entire section. Specifically, regarding figure 7B we write

*"Operating under the assumption that $C_5$, $C_6$, $C_7$, $C_8$, and $C_9$ HOMs primarily arise from isoprene, benzene toluene, $C_2$-alkylbenzene $C_3$-alkylbenzene oxidation respectively (Molteni et al., 2018; Wang et al., 2017), HOM signals plotted against parent VOC concentration indicate their dependence upon that VOC. Here, a $C_7$ HOM is thought to follow the formula $C_7H_{8-12}O_{5-10}N_{0-2}$. We have plotted HOM concentrations against VOC concentrations in Figure 7(b). $C_{10}$ HOMs are not included in these analyses as these may primarily arise from $C_{10}H_{12-14}$ alkylbenzene, or monoterpene oxidation. Concentration appears mostly independent of VOC concentration, with the exception of isoprene, for which emissions are highly temperature dependent, and thus this is likely a function of the effect of temperature on HOM formation (Figure 7(a)). A lack of correlation between other VOCs and their HOMs confirms that this relationship between HOMs and temperature is not a function of enhanced source fluxes from, for example, evaporation, except in the instance of isoprene"*

2) In the text we omit this statement, and provide a much revised form of these sections, as well as our conclusions re: NPF mechanisms (see response to major comment 1), this particular sentence now reads as follows.

*"The burst-event day has significantly lower concentrations of OVOCs and HOMs, and to a lesser degree, their nitrogen containing counterparts (N-OVOCs and N-HOMs), with significantly fewer compounds >400 m/Q. The most significant difference between full and burst-event days is in the SVOCs, accounting for a factor of two difference in concentration. The sulphur containing acids all have similar peak areas to the full-event day"*

3) The effect of HOM on NPF vs growth is discussed in the conclusions now, through the sentences

*"High concentrations of OVOC and HOMs were measured by CI-APi-ToF. Of these, the SVOC arose from mostly isoprene and alkylbenzene oxidation, whereas LVOC and ELVOC arose from alkylbenzene, monoterpene and PAH oxidation, with a dependence of their concentration on temperature, but less so on VOC concentration. Concentrations of species associated with coastal and oceanic sources such as MSA and $HIO_3$ were low. High HOM signals are seen to be a necessary condition for new particle growth past 10 nm, with the most significant difference between days with and without particle growth being SVOC concentrations (factor of 2 difference), while modelled growth rates from condensation of these organic compounds, alongside $H_2SO_4$, MSA and $HIO_3$ were shown to match growth rates within measurement error. Thus, oxidation of traffic derived alkylbenzenes and PAHs, and to a lesser degree, isoprene and monoterpene emissions are significant for new particle growth in this environment."*

These conclusions are based upon analysis of volatility and double bond equivalency (DBE) in our measured organic molecules. We have calculated DBE per molecule and saturation mass concentration as according to 2D-VBS, explained in the methods section as follows

*"The double bond equivalent (DBE) describes the degree of unsaturation of an organic molecule and is defined simply as*
$$DBE = N_C - \frac{N_H}{2} - \frac{N_N}{2} + 1 \qquad (3)$$
*The saturation vapour pressure at 300 K is defined by the 2D-volatility basis set (2D-VBS) as follows, if all nitrogen functionality is assumed to take the form $-ONO_2$ (Bianchi 2019; Donahue 2011; Schervish and Donahue, 2020):*
$$Log_{10}(C^*)(300\ K) = (N_{C0} - N_C)b_C - N_O b_O - 2\frac{N_O N_C}{N_C + N_O}b_{CO} - N_n b_N \qquad (4)$$
*Where $N_C$, $N_H$, and $N_N$, are the number of carbon, hydrogen, and nitrogen atoms respectively. $N_O$ is the number of oxygen atoms minus $3N_N$ to account for $-ONO_2$ groups, $N_{C0}$ is 25 (the carbon number of a 1 µg m$^{-3}$ alkane), $b_C$, $b_O$, $b_{CO}$, and $b_N$ are 0.475, 0.2, 0.9 and 2.5 respectively, and represent*

*interaction and nonideality terms. The final term of equation (4) represents the -ONO₂ groups, each reducing the saturation vapour pressure by 2.5 orders of magnitude. $C^*$ values are calculated at 300 K and not corrected, as 300 K is within 1 K of the campaign average temperature."*

We include the following sections in our discussion of HOM composition and sources

*"DBE as calculated by equation 3 is equal to the number of pi bonds and rings within a molecule. Benzene, toluene, and similar aromatics have DBE = 4, naphthalene = 7 and monoterpenes = 3. DBE can be used as an indicator of sources when considering HOM in bulk. Saturation mass concentration as calculated by equation 4 can help describe capacity of a molecule to both condense onto newly formed particles and participate in nucleation. Figure 8 shows concentrations of HOMs and other oxygenated organic molecules binned to the nearest integer $Log_{10}(C^*)(300 K)$, coloured by DBE. Mean ion signals per carbon number are shown in Figure S7. Most measured molecules fall into the SVOC class $(0.3 < C^*(300 K) < 300\ \mu g\ m^{-3})$ which will mostly exist in equilibrium between gas and particle phase. High SVOC concentrations arise from fingerprint molecules for isoprene oxidation under high $NO_x$ $(C_5H_{10}N_2O_8)$ (Brean et al., 2019), and oxidation of small alkylbenzenes $(C_7H_8O_5, C_8H_{10}O_5)$. LVOC and ELVOC $(3·10^{-5} < C^* < 0.3\ \mu g\ m^{-3}$ and $3·10^{-9} < C^*(300 K) < 3·10^{-5}\ \mu g\ m^{-3}$ respectively) have a greater contribution from molecules with higher DBE, i.e., $C_{10}H_{10}O_8$ arising most likely from PAH oxidation (Molteni et al., 2018), and $C_{10}H_{15}O_7N$, a common molecule arising from monoterpene oxidation in the presence of $NO_x$. The contribution of molecules with carbon number ≤ 9 to these LVOC is modest, and ELVOCs are entirely comprised of molecules with carbon numbers ≥ 10 and DBEs of 8 and 4. No molecules classed as ultra-low volatility organic compounds (ULVOC, $C^*(300 K) < 3·10^{-9}\ \mu g\ m^{-3}$) were observed."*

We discuss these in the context of NPF in section 3.3.2. We present our new figure below

[Figure]

A parametrisation of condensational growth from Nieminen et al. (2010) has been applied to these results, included in the supplement (also included below, where (a) shows $GR_{5-20}$ from various condensation schemes, and (b) shows $GR_{1.9-5}$ from similar schemes. Note the inclusion of SVOC in $GR_{5-20}$), alongside the following statement

*"Calculated growth rates according to the method of Nieminen et al. (2010) are presented in Figure S6 for both $GR_{1.9–5}$ and $GR_{5-20}$. Best agreement for $GR_{5-20}$ is when condensation of SVOC,*

*LVOC, ELVOC, MSA, HIO$_3$ and H$_2$SO$_4$ is considered, and best agreement for GR$_{1.9-5}$ is seen for condensation of all these except SVOC. The uncertainties in this method are large, and assumptions of irreversible condensation of SVOC onto particles of 5 nm likely lead to overestimations; however, these results confirm the essential role of the condensation of organic compounds to produce high growth rates observed in urban environments."*

[Figure]

**Major Comment 4: Strong conclusions regarding pathway for NPF**

*Based on the results from the analysis, the results are not as conclusive as claimed both by title ("molecular insights") and in the text (slightly different in abstract/discussion/conclusions). I would, therefore, suggest to rephrase the statements in text.*

*line 111: related to problem with title: "we examine molecular level evidence . . . at the critical diameter"- even the authors state "cluster identity cannot be identified" (line 276), so this wording is not appropriate.*

*line 293: It is clear that the results are similar to Kurten 2016 (except really high J). Mind that they conclude that SA-H2O-DMA or other pathway could not be confirmed with their data.*

*lines 301-302: I suggest softening this conclusion. Also check abstract (lines 38-39) for consistency.*

*line 387: The first and main conclusion is that NPF "plausibly proceeds by the formation of clusters involving sulphuric acid and highly oxygenated organic molecules, with likely involvement of bases". You clearly state earlier in line 276 that cluster identity cannot be directly identified and suggest that HOM is important for growth (not NPF, as far as I understood from text). So I am not sure what "plausibly" refers to. Please rephrase to reflect that results are not so conclusive, but the hypothesis is as follows (. . .).*

**Response 4:** Our conclusions regarding the potential nucleation mechanisms in Barcelona have changed slightly given the updated parametrisations of $H_2SO_4$-DMA-$H_2O$ nucleation from Kürten et al. (2018). We have made the following changes:

The wording at line 111 has been reworded to the following

*"Here, we examine gas phase mass spectral evidence and particle formation rates at the critical diameter from sulphuric acid in Barcelona"*

Further, at the end of section 3.2 we emphasise the following

*"however, as no higher-order clusters were observed, we cannot establish this with certainty."*

In our conclusions we now make the following statements

*"We show new particle formation rates in Barcelona are linearly dependent upon the sulphuric acid concentrations, and while formation rates far exceed that of $H_2SO_4$-BioOxOrg-$H_2O$ nucleation, they fall short of those of $H_2SO_4$-DMA-$H_2O$ nucleation at 278 K, as does the sulphuric acid dimer:monomer ratio, possibly explained by cluster evaporation due to high temperatures in summertime Barcelona (303 K during events), and limited pools of gas-phase amines. These results are similar to reports of nucleation rates in rural Germany (Kürten et al., 2016). As no higher-order clusters were directly measured, we cannot determine nucleation mechanisms with certainty, and an involvement of HOMs in nucleation is plausible."*

We feel that this better reflects the conclusions of this work, and have made similar changes to the abstract.

*Minor Comments:*

*line 63: I appreciate the use of very recent references, but surely there are older references/review that appropriate to be referenced here (from other groups).*

**Response:** This has been updated with reference to work across some diverse urban environments across the last 10 years, as well as a recent review from Lee et al., (2019), as this contains a section on NPF in polluted conditions with many recent references.

*line 74: near-kinetic limit is achieved at certain DMA concentrations, can be specified here (Almeida et al. 2013).*

**Response:** This section now reads

*"Nucleation of sulphuric acid, DMA and water proceeds at, or near to the kinetic limit **in a chamber at 278 K when DMA mixing ratios are sufficient** (Almeida et al., 2013; **Kürten et al., 2014**)."*

*line 143: In Brean et al. 2019, calibration coefficient of 7e8 cm-3 was used. Were any further adjustments done in this work to get 3e9 cm-3?*

**Response:** In this work, we correlated our uncalibrated data with a sulphuric acid proxy (Mikkonen et al., 2011), $R^2$=0.49 and took the gradient as our calibration constant. We compare with (Brean et al., 2019) to ensure our figures are in the correct ballpark, accounting for differences due to different voltage tune. For clarity, this now reads

*"Signals except for those of amines and ammonia are divided by the sum of reagent ion signals and multiplied by a calibration coefficient to produce a concentration. A calibration coefficient of $3\times10^9$ $cm^{-3}$ was established based upon comparison with a sulphuric acid proxy (Mikkonen et al.,*

*2011) and is in line with a prior calibration with our instrument (Brean et al., 2019). Uniform sensitivity between $H_2SO_4$ and all other species measured by CI-APi-ToF apart from amines and ammonia was assumed in this work. This introduces some uncertainties, as it relies upon both collision rates and charging efficiencies to be the same within the ionisation source for all species. Amine and ammonia signals are normalised to the nitrate trimer signal (Simon et al., 2016). Prior reports of ammonia and amines as measured by CI-APi-ToF employed corona discharge systems, which utilise higher concentrations of nitric acid, thus we report normalised signals."*

*lines 216-217: Sulfuric acid time series/dial profile are discussed, which is not shown anywhere. Consider including time series of measured parameters in the Supplementary.*

**Response:** We now include the diurnal profile of sulphuric acid as Figure S2

*Figure 3: Was it possible to reliable fit amines and ammonia from the spectra? Kurten et al. 2016 mentions that at high RH, nitrate-water cluster interferes with the fitting of C2 amines. Please comment and preferable provide a figure of the peak fitting. Please also include Kurten et al. 2016 into references when talking about detecting ammonia with nitrate CI-APi-TOF.*

**Response:** We provide correlations of each amine as clustered with the dimer + trimer, and see a small RH% dependence on this ratio. The RH interference isn't too significant in this data, as the interference is seen, for example between the $(H_2O)_6(HNO_3)_{0-1}NO_3^-$ ions and the $(DMA)(HNO_3)_{1-2}NO_3^-$ ions. As our system is slightly different to that of Kürten et al. (2016), with lower concentrations of $HNO_3$ in the front end of the system, this $H_2O$ clustering interference is smaller. We also provide intercorrelations of the amines with one another, and example peak fits of $C_2$ amines in Figure S1.

*lines 234-236: You probably mean to say that TMA won't efficiently cluster with nitric acid and won't be detected. It is not very clear, please rephrase.*

**Response:** We realise this is a mistake in the text and rephrase this whole section as follows

*"Although high emission fluxes of TMA are expected in this environment, they are not present in our spectra, although this ion has been reported previously (Kürten et al., 2016)."*

*line 267 and Fig. 5: I think CLOUD data presented here is J1.7 for 278 degC, isn't it? Please mention it in the text and/or caption.*

**Response:** This is correct. We specify this in the text as follows

*"Data from these chamber experiments is for 278 K and 38 - 39 % relative humidity."*

And in our figure captions

*"CLOUD chamber experiments were performed at 278 K and 38 – 39 % RH."*

We discuss the implications of these differing temperatures as in the response to major comment 1.

*line 282-283: "Sulphuric acid dimer roughly represents the strength of sulphuric acid clustering in the nitrate CI-APi-TOF". - I am not sure what this means. Did you mean it tracks atmospheric clustering?*

**Response:** This has been reworded for the sake of clarity also, and now reads

*"The sulphuric acid dimer:monomer ratio is elevated by the presence of gas-phase bases such as DMA, and this elevation is dependent upon both the abundances and proton affinities of such bases*

*(Olenius et al., 2017). Upon charging, evaporation of water and bases from sulphuric acid clusters occurs, and thus these are detected at sulphuric acid dimer (Ortega et al., 2012, 2014)."*

*line 287: "lower limit" – do you mean upper? Please explain how IIC was calculated, or was it taken directly from Kurten et al. (2016)? I would like to note that the reaction time in drift tube in Kurten et al. (2016), is 50ms, what is it in your instrument?*

**Response:** The reaction time in our Nitrate CI system is also 50 ms, so the calculated IIC as according to Zhao et al. (2010) is the same. "Lower limit" here was used to state that this is the lowest ratio that could be expected to be seen from this experimental setup, and has been reworded appropriately to

*"The dashed line represents the **ratio** that would be seen due to ion induced clustering (IIC) in the nitrate chemical ionisation system **for a 50 ms reaction time** (Zhao et al., 2010)"*

*line 318: Is this a relevant reference for C10 HOMs that are coming with inland air mass?*

**Response:** Our reference here is relevant to the discussion presented in Querol et al. (2017) regarding the inland air masses that flow towards the sea at night. This now reads as follows

*"These nighttime HOMs will therefore mostly be derived from biogenic emissions which undergo more rapid nocturnal oxidation, and are likely transported from inland by the land breeze during night (Millán, 2014; Querol et al., 2017)."*

*lines 323-328: a series of unconnected sentences. please rephrase to make more logical. "condensational growth being a reversible, step-wise kinetic process" – I am not sure what you mean, please rephrase. Do you mean "higher HIO3 cannot account for stopped growth beyond 10nm?. rephrase it to fit in HOM discussion or move it to section 2.2.*

**Response:** *"higher HIO3 cannot account for stopped growth beyond 10nm"* was indeed the initial statement, but this sentence has now been absorbed into a more thorough discussion of particle growth.

*In line 334, it is said that "precursors for these HOMs are presumed to be largely isoprene, alkylbenzenes and monoterpenes". However, in abstract (line 40) and in conclusions (line 394), isoprene is lost.*

**Response:** These have been re-added to the abstract and conclusions

*line 346: include also isoprene reference.*

**Response:** We include a reference to Massoli et al., where isoprene HOMs are reported in the ambient environment.

*line 369: you mention that HOM>500 m/Q are increased during "full-events", however no data above 500m/Q is shown. Or do you mean HOM within 400-500 m/Q? This has to be clarified as this is also mentioned in conclusions. It can be shown as time series (or dial) of the sum of unit mass resolution peaks above 500 m/Q or box plot (I understand high masses are hard to fit/identify).*

**Response:** These were repeated errors. This was meant to read >400 m/Q, and has been fixed.

*line 389-390: "multiplicity of mechanisms has been shown to occur in chamber studies but has not been observed in the real atmosphere previously."- for instance Yan et al. (2018) showed two*

*regimes for NPF at SMEAR II station, Finland. Kurten et al. 2016 shows very similar results to this study and analogously (lines 390-391) points out high J, similarity to CLOUD for both SA-DMA-H2O and SA-OxOrg studies. Please rephrase*

**Response:** These sentences have been rephrased. At the end of section 3.2 we emphasise the following

*"however, as no higher-order clusters were observed, we cannot establish this with certainty."*

Further, in our conclusions we have made the following changes

*"while formation rates far exceed that of $H_2SO_4$-BioOxOrg-$H_2O$ nucleation, they fall short of those of $H_2SO_4$-DMA-$H_2O$ nucleation at 278 K, as does the sulphuric acid dimer:monomer ratio, possibly explained by cluster evaporation due to high temperatures in summertime Barcelona (303 K during events), and limited pools of gas-phase amines. These results are markedly similar to reports of nucleation rates in rural Germany (Kürten et al., 2016). As no higher-order clusters were directly measured, we cannot determine nucleation mechanisms with certainty, and an involvement of HOMs in nucleation is plausible"*

We also include a reference to Yan et al. in the following sentence

*"Conversely, research in remote boreal environments show that the mechanism of nucleation can modulate dependent upon the $H_2SO_4$:HOM ratio (Yan et al., 2018)."*

*Figure 7: Why HOM is shown in normalized signal, while in Fig. 3 it is in cm-3? How many days and in what time resolution are the data in this plot? Caption: "Influencingfactors on VOC concentration" – on HOM concentration?*

**Response:** We present an updated figure in the manuscript presenting HOM as molecules/cm$^{-3}$ and have corrected the figure caption to now read

*"Figure 7: Influencing factors on HOM concentration, showing (a) temperature plotted against C5-10 HOM signal, coloured by global radiation. Ellipsis shows 95% confidence on a multivariate t-distribution. (a) VOC concentration plotted against HOM signal. These are segregated by carbon number/VOC, i.e, C7 HOMs plotted against toluene, under the assumption that toluene oxidation is the main producer of C7 HOMs. Time for both plots is of hourly time resolution."*

*Figure 8: Please provide a list of identified peaks in Supplementary and please explain what are the HOM observed at m/Q below 200 (quite high signals). Are they deprotonated species fitting with the "HOM" criteria?*

**Response:** Peak list has been included as requested and referenced in the text as follows

*"All ions identified are listed in Table S1."*

The inclusion of some small OVOCs as "HOM" was a printing error that has been fixed – these molecules are mostly deprotonated OVOCs such as dicarboxylic acids. The largest of these peaks belongs to malonic acid.

*I suggest the authors would look into following and add more quantitative interpretation of results in the text.*

*-line 44: "significantly lower" – here easy to specify exactly how much, a factor of X.*

*-line 87-89: phrase "despite (extremely) high condensation sinks" repeats twice in a row. For comparison here, it would be nice what is meant by "extremely high".*

*-line 100: please specify what is "frequent", 40-lines 101-105, 195: "relatively high ozone", "relatively low ozone (high compared with the rest of the year)"-can you rephrase or provide values as example? I also got lost in logic related to "relatively high" ozone during NPF and "maximum ozone episodes" with no NPF.*

**Response:** Maximum ozone episodes refer to phenomena occurring in Barcelona where high particulate matter and ozone pollution occur concurrently, however, as this has little relevance to our discussion of NPF this has been removed. Our referenced sources here fail to provide a mean $O_3$ value through the year, but the cluster analysis results show elevations compared to the other most common clusters.

*-line 244-245: "HOM . . . greatly enhanced", "lower during burst". Please provide values or a factor for HOM difference between days.*

*-line 248: ". . .HIO3 and . . .MSA are low". I assume you refer to Fig S5d and e here? Relative to what are these concentrations low?*

*-line 270: instead of "extremely high" provide actual numbers from Fig. 5*

*-line 271: what does "broadly similar" mean?*

*-line 349: what does "broadly dependent" mean?*

**Response:** We thank the reviewer for these very useful suggestions and have incorporated them through the whole text with the exception of one (see above).

*Technical corrections*

*line 125: "interface" with small i*

*line 222: remove empty line*

*line 333: "temperature plotted against the signal of HOMs". Isn't it the other way around? Also see caption to Fig. 7.*

*line 368: insert: "enhancement" in comparison "to smaller alkylbenzene derived HOMs".*

*line 401: repetition of line 394-395.*

*Figure 8. In-text (line 355), you mention that Fig. 8 has same days as Fig. 1. On Fig 1, there is 11/07/2018, while caption to Fig. 8 says 12/07/2018. Is it a typo? Please fix.*

**Response:** We again thank the reviewer for pointing out these errors and have incorporated the appropriate corrections in the text.

**Referee 2**

*Major Comment 1: "Burst" events and air mass histories It is well established that NPF events for which formation followed by continuous growth over many hours can be observed (here called "full event"), are observable as such at a point site because NPF takes place more or less in sync across a fairly wide area (i.e. regional-scale NPF). Therefore, I wonder if the so-called "burst events" could be due to NPF that occurs more localized? I.e., that the more localized character of the NPF is why growth beyond 10 nm is not observed (with smallest observed particles having formed closest-by, whereas the largest observed new particles would have formed farthest away within the NPF area, upwind). The time resolution of the particle sizing measurements was quite low, but an estimate of the overall growth rate of the newly-formed particles could anyway be made, and from that, plus wind back trajectories, even the size of the hypothesized local NPF area could be estimated. Related to that: The manuscript mentions at least twice that NPF events, both "full" and "burst" types, are associated with "southerly and south-westerly air masses" (e.g. L 396-399). But Fig. S6 (showing back trajectories) clearly contradicts that statement! I suspect the mistake is in the text? That suspicion is (a) because I would expect a regional-scale BVOC-HOM-driven NPF event to have air masses NOT arriving from sea+town, and rather from inland, as suggested, presumably, in Fig. S6; and (b) upon noticing that the burst event shown in more detail (e.g. Figs. 1 and 8; July 15) is indeed the only event that does actually show a southerly back trajectory in Fig. S6, which puts the trajectory mostly over the ocean, with the exception of its final path across town (and, possibly critically, over shipping, as the authors also point out) towards the measurement site.*

We discuss potential uncertainties arising from such local sources, following an extended discussion of methods used to derive *J* and *GR*. The data do not support the concept of burst events arising from an event of limited spatial extent. If they did, we would expect to see the disappearance of the mode associated with the burst and appearance of a new size distribution reflective of the new air mass. In our observations, the growth curve levels off – a mode of particles appears <10 nm and stays around that diameter, persisting for the course of hours. The argument of growth processes being limited is consistent with low concentrations of SVOC/LVOC/ELVOC on these days.

*"The above calculations rely on the assumption of homogeneous air masses, and while air mass advection, as well as primary particle emissions can cause errors in estimations of temporal changes in particle count and diameter, the appearance and persistence of a new mode of particles across a period of several hours is typically indicative of a regional process."*

The discussion of south-westerly air masses was based upon an analysis of local wind direction + speed measurements. From polar plots it was inferred that air masses arising from the south were dominant during NPF, however, we find the air mass history from HYSPLIT to be more informative for our analyses, and thus these were included. The wind direction and velocity across the burst event days is shown below, where it can be seen that air masses are stable following the appearance of the new mode, and persist across the period of multiple hours, giving us confidence that this is not due to advection of a different air mass.

[Figure]

*Major Comment 2: Number of events and data points Line 204 states that there was only 2 "burst" events observed. Then, it would be very useful to also learn how many "full" event were observed (and how many were non-event days)? However, with only 2 samples for the burst events, how can Figs. 2 and 3 include a boxplot for conditions during those two events? That may be down to the too sparse explanation of what data were used for these boxplots. E.g., data from which times were used in each event class (a vs b vs c)? Is the boxplot based on event-wise averages, or something else? But then again, Figs. 4+ suggest there is many more events of each type (judging from # of markers). A Fig. S6 again suggests again there's only been 5 events total!? I have some suspicions, but overall I am left rather confused. So, all that clearly should be presented more clearly, i.e. data quantity and usage in figures, resolved by event type.*

**Response 2:** We answer these questions as below:

The periods of data that have been included have been clarified in the following sections

Figure captions for Figures 2 + 3 include the sentence

*"[F]rom hourly data. Event days include data across the full event day."*

And further, we address potential issues arising from producing box plots from a limited dataset by also including a complementary Figure S2, containing diurnal cycles of all the included parameters of Figures 2 and 3. Data coverage for some figures was a little misleading. Labels for Figures 4 – 6 have been amended for consistency, containing the phrase

*"Data is for hourly averages across NPF periods, typically within the hours 08:00 – 16:00."*

And in the start of section 3.1 we include the following

*"Figure 1(b) shows a nucleation day with growth to larger sizes >10 nm, termed "full-event", showing the growth through the course of the day. These fulfil all the criteria of Dal Maso et al. (2005). 4 events of this type were observed with CI-APi-ToF data coverage. Figure 1(c) shows a day with nucleation occurring, but no growth past 10 nm. These days are referred to as "burst-event" days. Here, NPF is seen to occur, but particles fail to grow past the nucleation mode. 2 such events were seen in this data with CI-APi-ToF data coverage, and both are accompanied by a distinct mode appearing beforehand at ~20-40 nm."*

We also present an updated Figure S5 containing the identity of each event day .

*Major Comment 3: Section 2.2: Was "sensitivity" applied to any detected compound? From the rest of the paper I conclude that it was applied to H2SO4 and HOMs, but not to ammonia and amines. Overall, some more detailed discussion of concentration quantifications based on CIMS signals would be interesting. See also next comment.*

**Response 3:** We expand our discussion of quantifications in the following sentences

*"Signals except for those of amines and ammonia are divided by the sum of reagent ion signals and multiplied by a calibration coefficient to produce a concentration. A calibration coefficient of $3\times10^9$ cm$^{-3}$ was established based upon comparison with a sulphuric acid proxy (Mikkonen et al., 2011) and is in line with a prior calibration with our instrument (Brean et al., 2019). Uniform sensitivity between $H_2SO_4$ and all other species measured by CI-APi-ToF bar amines and ammonia was assumed in this work. This introduces some uncertainties, as it relies upon both collision rates and charging efficiencies to be the same within the ionisation source for all species. Amine and ammonia signals are normalised to the nitrate trimer signal (Simon et al., 2016). Prior reports of ammonia and amines as measured by CI-APi-ToF employed corona discharge systems, which*

*__Major Comment 4:__ Ammonia and amines: Section 2.2: Are there any estimates on detection limits or sensitivity regarding ammonia and amines? (And which amines are expected to be detectable?) There is some discussion of amine detection late in Section 3.1. But it would be helpful if that more general instrumental aspects of their detection would already be at least mentioned in the Methods section. Section 3.1: Which amines were actually detected? I think that information is only provided in the caption of Fig. 3 and in Fig. S1 (implying C2H7N and C4H11N), and that should be mentioned also in the main text, and much earlier. (And is that in agreement with expectations from amine abundances and amine-specific sensitivities indicated by previous studies (if any)?) The authors do discuss relative sensitivities later in Section 3.1, implying that sensitivities to all those small bases correlate to NPF enhancement potential. That argument could be brought sooner. Figure 3, and related discussion (Section 3.1, page 9): From the beginning of the discussion of Fig. 3b, I have been wondering why are ammonia & amine signals only presented as a sum of all signals? Even if the concentrations of these compounds could not be quantified in these measurements, it still appears it would be more insightful not to lump them together. Especially given the respective differences in NPF enhancement capabilities. Also, if lumped together, there is the chance that differences in atmospheric abundance of ammonia vs. amines (almost certain, as also described in Section 3.1) or in instrument sensitivity to these compounds (plausible) lead to one compound (or group of compounds) controlling the lumped signal... Only very late in Section 3.1, L 241 mentions that all the bases' time series correlated quite well, thus justifying the lumping. That argument for lumping should be made more clearly and earlier. And supported by a figure or two in the Supplement. L 217: From looking at Figure 3, I am not sure I would also arrive at the conclusion that ammonia+amine concentrations were enhanced for full events. The medians are practically identical. Have the authors applied some quantitative measure of statistical significance regarding differences of the parameters presented in Figs. 2-3?*

**Response 4:** Unfortunately, the two prior reports of amines (Simon et al., 2016) and both ammonia and amines (Kürten et al., 2016) from $NO_3^-$ ionisation used a corona discharge CI-APi-ToF system, which is distinct from our own for these purposes as the amount of reagent ion in the front end of the instrument is higher. The result of this is that detection limits and sensitivities are likely significantly different to our own. We can apply the sensitivities of Simon et al. (2016), where a sensitivity a factor of 10 different from that for $H_2SO_4$ is applied, and this gives us a mean $C_2$ amine concentration of just under 10 pptv across the campaign.

Amines and ammonia have been considered as one "sum" here as they follow similar trends. Below we tabulate the Pearson correlation coefficients between ammonia and amines, as well as a plot of ammonia plotted against $C_2$ amine signals, coloured by $C_4$ amine signals. We also include all of this information in Figure S1, which is included below (see panel (b)) as well as an expanded Methods section (see quoted text above) to clear issues of placement of this information within the manuscript, while the following is included in the text.

*"We present correlations of each of these bases clustered with the nitrate dimer plotted against measurements with the nitrate trimer, as well as their intercorrelations and example peak fits across Figure S1"*

| | $C_2$ am. | $C_4$ am. |
| --- | --- | --- |

[Figure]

| | | |
|---|---|---|
| $NH_3$ | 0.71 | 0.58 |
| $C_2$ am | | 0.94 |

[Figure]

We also present individual diurnals for these compounds in a new Figure S2. Fortunately, the magnitude of each signal of each of these compounds is similar, and thus our Figure 2 is not weighted towards ammonia. This is not as significant as the difference expected in typical atmospheric mixing ratios (Ge et al., 2011). We realise this is evidently a function of an insensitivity towards ammonia. We remove ammonia from this figure for both this reason, and because the presence of alkylamines even in low mixing ratios will likely be more significant for nucleation processes than ammonia (Almeida et al., 2013; Yao et al., 2018) as even modest mixing ratios of alkylamines can substitute ammonia in sulphuric acid-ammonia clusters (Kupiainen et al., 2012), and thus argue that these are the important aspect of these measurements. The argument of significance was made based upon a simple t-test.

*Major Comment 5: Figure 5 and/or related discussion Environmental conditions should be mentioned and discussed, in particular in light of the CLOUD experiments being compared to: T, RH, compound concentrations (as known), as all those will have affect formation rates. Also: given available parametrizations of NPF rates as a function of sulfuric acid, HOM (or "BioOxOrg") and amine/ammonia concentrations, are the authors able to explain observed NPF rates and infer concentrations of either involved BioOxOrg or amines? Combined with expected amine concentrations and measured VOC and HOM concentrations, some closure could be attempted. Some hand waving may be necessary, but the attempt could be quite interesting. And it could strengthen the paper, especially if it can be argued that closure ~works out. Similarly, growth rates of new particles could be estimated (see also 1st comment) and evaluated against HOM concentrations.*

**Response 5:** Regarding the temperature and relative humidity dependencies, we include the following

*"Model studies of sulphuric acid-amine nucleation show a decline in nucleation rate with temperature (Almeida et al., 2013; Olenius et al., 2017), as the evaporation rate of sulphuric acid-amine clusters will increase with temperature (Paasonen et al., 2012). Conversely, evaporation rates of such small clusters, and resultant nucleation rates tend to increase modestly with increases in relative humidity, most pronounced at lower amine concentrations (Almeida et al., 2013; Paasonen et al., 2012). Despite this, high nucleation rates at temperatures nearing 300 K have been reported previously (Kuang et al., 2008; Kürten et al., 2016), although these tend to show a temperature dependence (Yu et al., 2016)."*

We also emphasise the differing conditions of these chamber results in our figure captions. Included in the new version of the manuscript is an updated parametrisation of $H_2SO_4$-DMA-$H_2O$ nucleation from Kürten et al. (2018), which somewhat changes our conclusions. The results of Kürten et al. (2018) involve re-analysis of the CLOUD data and produce nucleation rates around an order of magnitude higher than in Almeida et al. (2013). Thus, we produce a new Figure 5 included below. This produces a far more consistent picture, and we now suggest a nucleation pathway possibly involving sulphuric acid and amines, with rates reduced by around an order of magnitude by elevated temperatures, consistent with prior evidence (Olenius et al., 2017). We therefore include the following sections

*"Particle formation plausibly operates by sulphuric acid-amine nucleation involving the measured C2 and C4 amines in our data, with nucleation rates hindered relative to those measured in the CLOUD experiments by elevated temperatures, and a decline to the sulphuric acid dimer:monomer ratio indicates that base concentrations may be limited. We cannot rule out an involvement of HOMs in particle formation processes, and, as no higher-order clusters were observed, we cannot establish sulphuric acid-amine nucleation with certainty."*

And conclude

*"while formation rates far exceed that of $H_2SO_4$-BioOxOrg-$H_2O$ nucleation, they fall short of those of $H_2SO_4$-DMA-$H_2O$ nucleation at 278 K, as does the sulphuric acid dimer:monomer ratio, possibly explained by cluster evaporation due to high temperatures in summertime Barcelona (303 K during events), and limited pools of gas-phase amines. These results are similar to reports of nucleation rates in rural Germany (Kürten et al., 2016). As no higher-order clusters were directly measured, we cannot determine nucleation mechanisms with certainty, and an involvement of HOMs in nucleation is plausible."*

We would argue that the necessary hand waving might be too great to justify such a parametrisation as suggested above. An example of such a parametrisation would be that of Hanson et al., (2017), wherein a simple equation to derive *J* during $H_2SO_4$-DMA-$H_2O$ nucleation is proposed, however, this requires a quantified DMA mixing ratio, and to assume one would introduce quite some uncertainties into this. However, we hope the reviewer find our updated discussion of potential mechanisms of nucleation compelling.

[Figure]

***Major Comment 6:*** *Figure 6 and related discussion (in particular L 289-293): Could the SA dimer signal also be affected by instrument settings? It is conceivable that some instruments or settings would fragment a certain fraction of SA dimer ions (that is, at some point after their formation by NO3-ionization), and that that fraction is instrument- or tuning specific. If so, conclusions can likely still be made anyway from comparisons between measurements within the same campaign (provided settings remained the same), and probably even from comparisons across campaigns/longer time periods as long as the same instrument was used. But could such instrumental differences cause (part of) the discussed discrepancies between dimer signals here and results shown from other field and lab experiments? My feeling is that the SA dimer anion is stable enough that such instrumental fragmentation should not be expected, but I would ask the authors to at least point that out (i.e., if my "feeling" can be defended based on previous studies – I apologize for not remembering expected cluster stabilities vs the fragmentation potency of the APi-TOF instrument), OR recognize/discuss potential issues.*

**Response 6:** The region of the instrument most likely to produce fragmentation of the sulphuric acid dimer is the collision dissociation chamber (CDC). A factor of 1.2 difference in sulphuric acid dimer concentration compared to that expected from ion induced clustering in the chemical ionization source has been shown in a chamber study (Kürten et al., 2015). This is consistent with the high binding energies of the $H_2SO_4HSO_4^-$ cluster, in excess of 40 kcal mol$^{-1}$ (Curtius et al., 2001), around a factor of two higher than other commonly measured ions such as $HOM-HSO_4$ or $HOM-NO_3^-$ clusters (Froyd and Lovejoy, 2012; Kurtén et al., 2007; Zanca et al., 2020). The result of this is an evaporation rate of around $2.7\cdot10^{-15}$ s$^{-1}$ compared to an evaporation rate of DMA out of the $(DMA)_2H_2SO_4HSO_4^-$ ion of $2.7\cdot10^{-1}$ s$^{-1}$). Further evidence is that when a negatively charged sulphuric acid trimer makes its way into the instrument it is fragmented largely into a charged sulphuric acid dimer, and a neutral monomer (Passananti et al., 2019). To our knowledge there have been no results published on the effect of instrument tuning on the sulphuric acid dimer (just those on the sulphuric acid trimer and HOMs as referenced above). We do, however, include the following sentence in our manuscript (with further mention of the potential effect of condensation sinks

*"The binding energy of the bisulphate-$H_2SO_4$ ion is in excess of 40 kcal mol$^{-1}$ (Curtius et al., 2001), and thus minimal declustering of the dimer is expected within the CI-APi-ToF instrument – however, declustering of higher order sulphuric acid clusters has been shown to be sensitive to voltage tune (Passanati et al., 2019), and this likely extends to the dimer also, and as such discrepancies between sets of results due to instrument setup cannot be ruled out. The ratio of sulphuric acid dimer:monomer is also highly sensitive to condensation sinks, with a difference in dimer concentration of approximately a factor of 4 expected at $10^7$ cm$^{-3}$ between 0.001 s$^{-1}$ (a clean environment) and 0.03 s$^{-1}$ (our calculated condensation sinks during NPF events) (Yao et al., 2018) and thus our low dimer:monomer ratio can, in part, be explained by elevated condensation sinks."*

***Minor comments:***

*Abstract: a) I would already here mention roughly the most important measurement methods used for the study. No details, and a side sentence may be enough. Especially as there is talk about sulfuric acid monomer and dimer concentrations, I would have liked to be informed already here that those are based on NO3-CIMS measurements.*

**Response:** We have included the following in the abstract

*"Here, we study summertime NPF in urban Barcelona in NE Spain **utilising particle counting instruments down to 1.9 nm and Nitrate CI-APi-ToF**."*

*b) Growth beyond 10 nm: my immediate thought already here was that limited growth could also be a sign of more local NPF (vs. regional, i.e. on a larger geographical scale), in which case the observed lower concentrations of low-volatility organics could be irrelevant (see 1st major comment). So I would already here, in a compact way, give the reasoning for claimed conclusion.*

**Response:** This has been included.

*Introduction: The last paragraph (review of NPF observations in the Barcelona area) is a somewhat confusing to read. Should be restructured for clarity.*

**Response:** This has been restructured – we have removed the mention of high ozone events as they have little relevance to NPF, as well as mention of ozone across other urban environments. This section now reads

*"Urban Barcelona sees frequent, strong summer-time NPF events occurring on 28% of days. These events are associated with high insolation and elevated ozone (~60 µg m$^{-3}$) when considering the whole year (Brines et al., 2014, 2015). Ground-level observations report NPF events starting typically at midday, and either occurring in urban Barcelona and the surrounding regional background simultaneously, or isolated to either urban Barcelona or just the regional background (Dall'Osto et al., 2013). Vertical profiles over urban Barcelona reveal that NPF occurs at higher altitudes, and starts earlier in the day, as at a given altitude these events are not suppressed by early traffic peaks contributing to particle load (Minguillón et al., 2015)."*

*I found the usage of the term "background" not clear (end of introduction, beginning of methods, beginning of results).*

**Response:** The inclusion of the term "urban background" refers to the urban environment over the scale of several km$^2$. An "urban background site" is an urban site away from direct emission sources that is descriptive of a well mixed urban region. "Regional background" refers to Barcelona and its surrounding regions, here on the scale of tens of kilometres. This terminology is well accepted in the air pollution literature.

*L 143: The meaning of "sensitivity" here (in previous studies called also "sensitivity coefficient" or "calibration factor") remains a mystery for any reader who is not fairly intimate with operation of that CIMS instrument. So, I would at least cite some paper where that meaning is discussed, e.g. Kürten et al., 2012 (10.1021/jp212123n). Regarding the "sensitivity" value used, the authors cite here a previous calibration, paper also led by UBirmingham. However, I could not find, in that paper, how that calibration was performed. Indeed that paper contains the statement (in their section 2.2) "No sensitivity calibration was performed for these measurement..." Bottom line is that I have remained wondering where the used value (3e9 cmˆ-3) derives from. (Or simply from comparison to the sulfuric acid proxy?)*

**Response:** To answer, and provide more detail re: calibrations, sensitivities etc., we include the following

*"Signals except for those of amines and ammonia are divided by the sum of reagent ion signals and multiplied by a calibration coefficient to produce a concentration. A calibration coefficient of $3 \times 10^9$ cm$^{-3}$ was established based upon comparison with a sulphuric acid proxy (Mikkonen et al., 2011) and is in line with a prior calibration with our instrument (Brean et al., 2019). Uniform sensitivity between $H_2SO_4$ and all other species measured by CI-APi-ToF apart from amines and ammonia was assumed in this work. This introduces some uncertainties, as it relies upon both collision rates and charging efficiencies to be the same within the ionisation source for all species. Amine and ammonia signals are normalised to the nitrate trimer signal (Simon et al., 2016). Prior reports of ammonia and amines as measured by CI-APi-ToF employed corona discharge systems, which utilise higher concentrations of nitric acid, thus we report normalised signals. We present correlations of each of these bases clustered with the nitrate dimer plotted against measurements with the nitrate trimer, as well as their intercorrelations and example peak fits across Figure S1."*

*L 221: It is not clear if this is a general statement regarding amine concentrations, or specific to observations (which, however, I understand could not be quantified, so I assume it's the former?)? Should be clarified.*

**Response:** This is indeed a general statement, we have included a reference to an appropriate source here.

*Figure 1 is lacking all labels for the axes. At least the somewhat-less-obvious vertical axis should be labelled.*

**Response:** Figure 1 has been amended (and individual days have been replaced with the average contour for each of these day-classes.

*L 244: Should start new paragraph when starting discussion HOMs.*

**Response:** Done

*L244-245: It appears odd to discuss HOMs (Fig. 3b) with only 1 sentence, following a page of discussion of ammonia/amines! Should at least add a reference to a later section, where organics, including HOM, are being discussed.*

**Response:** This section now reads

*"HOM concentrations were greatly enhanced during full-event periods (factor of 1.5 higher compared to non-NPF mean), but lower during burst-event periods (factor of 1.2 lower compared*

*to non-NPF mean), implying their necessity for growth. The sources and implications of these HOMs are discussed in section 3.3"*

*L247-248: This last sentence of Section 3.1 is very vague. How high (or low) are those concentrations of marine compounds? (Or estimated to be?) What observations would have been considered evidence FOR an influence of oceanic emissions on NPF/growth?*

**Response:** These have been stated and this section now reads

*"concentrations of iodine and DMS-derived acids such as iodic acid ($HIO_3$) and methanesulphonic acid (MSA) are low ($7.8·10^5$ and $3.3·10^5$ $cm^{-3}$ respectively), indicating a small influence of oceanic emissions on particle nucleation/growth"*

Evidence for the influence of oceanic emissions would, in this context, be taken as the presence of high mixing ratios of oceanic-emission derived acids. An example would be a high $MSA:H_2SO_4$ ratio as both arise plentifully from DMS oxidation (Hoffmann et al., 2016), or high concentrations of iodine acids derived from biogenic iodine vapours (Sipilä et al., 2016).

*Fig. 4: How steep ist that slope? That is usually interesting information, at least for comparing with other studies.*

**Response:** This has been included in the figure caption. The value of the slope is $4.9·10^{-5}$ $s^{-1}$

*L 261: Does "losses" refer to amines? If so, I don't see how photochemical reductions of amine mixing ratios could mask an actual dependence of NPF rates on those mixing ratios. (If I caught the inference correctly.)*

**Response:** That was indeed the inference, but this point is secondary to the more important one made before. This section now reads

*"In the example of alkylamines, their gas phase concentration may decrease due to clustering with elevated sulphuric acid, **as they cluster at around a 1:1 ratio at high amine mixing ratios (Kürten et al., 2014)** (and therefore they will not be detectable as free amines). **Further, if amines are present at a few pptv, their mixing ratios are significantly higher than our ambient measured sulphuric acid concentrations, and will be sufficient to accelerate nucleation rates (Almeida et al., 2013)**. Photochemical losses will also be greater during the periods of highest NPF rate (Ge et al., 2011b)."*

*L 273: Suggest rephrasing to make it clear immediately that the discussion shifts from literature results to new results. And it is not clear which observations the last part of the sentence refers to ("on these days").*

**Response:** We have clarified this. It reads

*"indicating similar mechanisms of formation, despite lower HOM concentrations on **burst-event days**"*

*L 282-283: ambiguous what is meant by "strength of sulfuric acid clustering". It becomes clear thereafter, but if I understand correctly, "strength" is not the right word.*

**Response:** This has been changed and now reads as follows

*"To further explore the relationship between sulphuric acid clusters and the rate of nucleation, the sulphuric acid dimer:monomer ratio is plotted in Figure 6. **The sulphuric acid dimer:monomer ratio is elevated by the presence of gas-phase bases such as DMA, and this elevation is dependent***

*upon both the abundances and proton affinities of such bases (Olenius et al., 2017). Upon charging, evaporation of water and bases from sulphuric acid clusters occurs, and thus these are detected as sulphuric acid dimer (Ortega et al., 2012, 2014)."*

*L 293+: Could the point raised in major comment (6) explain the flatter slope observed here vs. the Germany observations?*

**Response:** We include a discussion of this, alongside a discussion of the result of elevated condensation sinks as mentioned in the response to major comment 6

*L 337: Getting confused here. Should it read "not largely radiation dependent" instead of "not largely temperature dependent"?*

**Response:** This section has been modified to read as follows

*"A lack of correlation between other VOCs and their HOMs confirms that this relationship between HOMs and temperature is not a function of enhanced VOC emission fluxes from, for example, evaporation, except in the instance of isoprene."*

*L 342: Should they be transportent FROM inland by the land breeze?*

**Response:** Correct. This has been amended.

*L 357, 361: Should be explained what is meant by "detailed criteria", and by "updated criteria" (i.e. what are the respective criteria).*

**Response:** We update this section to the following, and move it slightly earlier

*"Oxygenated volatile organic compounds (OVOC) are defined as species visible in the nitrate CI-APi-ToF that do not classify as HOM. Here, the first of the three criteria provided by Bianchi et al. (2019), that HOM must be formed by peroxy radical autoxidation, cannot be applied to define HOM, as knowledge as to whether a molecule is a result of autoxidation requires sound knowledge of the structure of the precursors, oxidants and peroxy radical terminators present, however, the number of molecules observed with $nN = 2$ is around an order of magnitude lower than that for $nN = 1$, where the primary source of multiple nitrogen functionalities would be multiple peroxy radical termination reactions from $NO_x$, and we argue that while multiple generations of oxidation have been shown to occur in aromatics (Garmash et al., 2020), it is a small contributor to the concentration of what we class as HOM. The second criterion to define HOM are that they must be formed in the gas phase under atmospherically relevant conditions, which we deem appropriately fulfilled as all CI-APi-ToF measurements are of gas phase compounds, and the final criterion is that HOM must contain more than 6 oxygen atoms. To attempt to satisfy these criteria as best possible, the criteria of both containing 6 oxygen atoms and 5 carbon atoms or greater and having an O:C ratio >0.6 is applied, as these molecules will all plausibly fulfil the updated criteria of "HOM"."*

*L 367: could be informative to point out some of those formulae explicitly*

**Response:** We include the following

*"The full-event day sees enhancements to smaller OVOCs and HOMs compared to the non-event day, especially around 150-200 m/Q, which contains peaks corresponding to dicarboxylic acids and isoprene oxidation products. Some of the largest peaks in the mass spectra correspond to formulae seen arising from the enhanced OH· oxidation of alkylbenzenes (such as $C_7H_7NO_6$) (Molteni et al., 2018; Wang et al., 2017)"*

*L 371-372: would be instructive to be more specific regarding "large" and "smaller"*

**Response:** We include the following

*"During full-event periods, these peaks are both more numerous and larger, with a factor of two difference in total peak area in this m/Q range"*

*L 378-382: Something went wrong with this sentence, especially the first part. Think I get the idea, but not sure.*

**Response:** We edit these sentences to now read

*"These elevations to condensable OVOC and HOMs on particle formation days with growth are consistent with particle composition data as measured by ACSM (Figure S9). Particle composition on full-event days shows an elevation to organic mass concentration in the late evening and night around when new particles from NPF will reach sizes detectable by the ACSM (~75 nm, Ng et al., 2011). Organic mass between 16:00 – 23:00 is 3.5 µg m$^{-3}$ on burst-event days, versus 7.8 µg m$^{-3}$ on full-event days."*

*L 381: Please state the size range the ACSM is sensitive to*

**Response:** This is stated in the methods now

*"Continual monitoring of composition and mass of submicron aerosol >**75 nm** was carried out with an Aerosol Chemical Speciation Monitor (ACSM, Aerodyne, USA) (Ng et al., 2011)."*

*L 393: Could the mechanisms also support each other (i.e. be combined) rather than be in competition?*

**Response:** This statement in the conclusions has been softened, and now reads

*"[F]rom our evidence, we hypothesise a mechanism proceeding by the formation of clusters involving sulphuric acid, with potential involvement of both HOMs and amines, as we measure both small alkylamines, and molecules of class ELVOC, both of which have shown themselves capable of forming particles in conjunction with sulphuric acid. This potential multiplicity of mechanisms has been shown to occur in chamber studies and is markedly similar to reports from rural Germany (Kürten et al., 2016)."*

*Fig. S6: Please include information on which kind of events are shown, and when (... see also major comments (1) and (2))*

**Response:** This figure has been updated with the appropriate information

***Technical comments***

*L 61: i.e. should be e.g.?*

*L 79/80: those "loss processes" haven't been mentioned yet, so would be instructive to name the most important ones (here, for < 50 nm particles)*

*L 81: "at these diameters" ... rather "from these diameters onward"*

*L 86: I believe that range should read "0.5-11" (i.e., 11 instead of 1.1)*

*L 86-90: two redundant consecutive sentences. combine.*

*L 138: if I remember correctly, the flow containing the reagent ions is not "guided into the sample flow", but rather only the ions are guided there (electrically).*

*L 162: redundant mentioning of "4 flows"*

*Fig. 3, a and b: the "+" in the exponents (tick labels) are conventionally omitted*

*L 371: compromise -> comprise*

*Figs. S6 and S7 are not referred to in the main text (not sure if that's a problem)*

**Response:** We thank the reviewer for these comments – these have been amended in the text.

**Referee 3**

Hi Josef, thank you for the insightful comments. We have separated these into individual comments and addressed these in-turn below

*Comment 1: Since molecular level knowledge of new particle formation in urban areas is still very poor I read this paper with high interest. I strongly support the reviewers asking for a better description of the analysis methods and more quantitative information. For example in Figure 1 b no distinct particle evolution is seen. This looks more like an advection of an air mass. It would be worthwhile to show in an example how the nucleation rate was determined? The same applies to the growth rate calculation. GRs seem to be quite high but time resolution of measurements rather low for such events. For example, the time resolution of the PSM measurements is 10 min. Thus for GR>6 nm/h no growth rate in the sub-3nm range can be determined.*

**Response 1:** In line with these suggestions, we have produced an expanded methods section. We use a time-delay method utilising the 1.9 nm size cut of the PSM and compare it to the integrated particle counts from the NanoSMPS + SMPS measurements, giving the time-delay between measuring particles at 1.9 and 5 nm, thus, particle growth rates would need to reach ~18 nm h$^{-1}$ in order to escape the resolution of our measurements. We realise this has its potential uncertainties, and include a discussion of such uncertainties in our results section. Alongside there is an expanded Methods section where we state the following:

*"The formation rate of new particles at size $d_p$ is calculated as follows:*

$$J_{d_p} = \frac{dN_{d_p}}{dt} + CoagS_{d_p} \cdot N_{d_p} + \frac{GR}{\Delta d_p} \cdot N_{d_p} \qquad (2)$$

*Where the first term on the right-hand side comprises the rate at which particles enter the size $d_p$, and the latter two terms represent losses from this size by coagulation and growth respectively. $J_5$ was calculated using the data between 5-10 nm, and $J_{1.9}$ was calculated using the measurements between 1.9 – 4.5 nm. We also calculated $J_{1.9}$ from our NanoSMPS data, employing the equations of Lehtinen et al. (2007). $J_{1.9}$ from both methods showed reasonable agreement ($R^2 = 0.34$) Agreement between $J_5$ and $J_{1.9}$ for each method was similar ($R^2 = 0.37$ and $R^2 = 0.38$ for $J_{1.9}$ calculated from PSM data and from Lehtinen et al. (2019) respectively). See Kulmala et al. (2001) for more information on calculation of coagulation sinks and formation rates. Growth rates between 4.5 – 20 nm were calculated according to the lognormal distribution function method (Kulmala et al., 2012), whereas those between 1.9 and 4.5 nm were calculated from PSM data using a time-delay method between PSM and NanoSMPS data. Systematic uncertainties on our calculated $J_{1.9}$ values include 25% method uncertainty (Yli-Juuti et al., 2017), with a further 25% arising from uncertainties in PSM cutoff (±0.5 nm), as well as a 10% uncertainty in counting errors. A 50% error arising from calculated coagulation sink is also applied (Kurten et al., 2016). The above calculations rely on the*

*assumption of homogeneous air masses, and while air mass advection, as well as primary particle emissions can cause errors in estimations of temporal changes in particle count and diameter, the appearance and persistence of a new mode of particles across a period of several hours is typically indicative of a regional process."*

**Comment 2:** *It is hard to believe that J5 and J1.5 are almost similar. At H2SO4 = 2E06 the reported J5 is even ten times higher than J1.5.*

**Response 2:** We confirm that the initial Figure 4 was a printing error – in the updated form of this manuscript we present an updated Figure 4 using $J_{1.9}$ values. For your consideration below we also include $J_5$ and $J_{1.9}$ on the same plot which differ by around the expected order of magnitude or so, given the high coagulation sinks.

[Figure]

**Comment 3:** *What are the uncertainties of J1.5, J5 and GR at different sizes? As pointed out by Kulmala et al. (2017) nucleation and growth rate are connected. It would be very helpful for the reader if the authors would also report the growth rates, as they have been determined anyhow. Kulmala et al. (2017) also show that the survival probability of small clusters becomes very low at such high coagulation/condensation sinks as reported in this study. A discussion of this phenomenon should be included.*

**Response 3:** The revised manuscript now contains stated growth rates as calculated from our particle count data, as well as a mean survival parameter. We include these in the following sentences

*"The survival parameter (P) as suggested by Kulmala et al. (2017) is defined as CS·10⁻⁴ / GR, and for this data is equal to 82, higher than other European cities. The occurrence of such a high P value should, in theory, inhibit the occurrence of NPF, but we show events happen readily under such conditions, akin to other heavily polluted megacities."*

We also include the updated $J_{1.7}$ values of Kürten et al., (2018) in a new Figure 5, while parameterising our errors as follows: We estimate a 25 % error in GR arising from methods chosen (Yli-Juuti et al., 2017) and a further 25% arising from potential uncertainties in the PSM cutoff of ± 0.5 nm. A 10 % uncertainty has been presumed to arise from errors in counting, and a 50 % error arising from systematic errors in calculating coagulation sinks (Kürten et al., 2016). Similarly, we present a +100% / -50 % error on sulphuric acid condensation arising from systematic errors in our estimation of $C$. We include a discussion of such errors in our methods, and further reference these uncertainties in the respective figure captions. Below, we show a comparison of our own $J_{1.9}$ and $J_5$ values with the CLOUD data from Almeida et al., (2013) and Kürten et al., (2018).

[Figure]

***Comment 4:*** *The authors strongly stress the role of organics in NPF. Measurements of the HOMs and their chemical composition are available. Thus, the authors could estimate if the concentration of very low volatility HOMs is high enough to account for the growth of few nm-sized particles. Such an analysis would support their conclusions on what drives NPF. In Figure 8 it is hard to see differences in the mass spectra at m/z>250. Thus, most differences are at lower m/z and for compounds with high mass defect, that is low oxygen content, and thus high volatility.*

**Response 4:** Regarding the interesting comment on growth rates, we have calculated $Log_{10}(C^*)(300$ K) as according to the simple linear equation provided in Bianchi et al., (2019), an updated form of that first presented in Donahue et al., (2011). As per their recommendations, we include an additional term to account for nitrogen functionality where every -O-NO$_2$ group reduces $Log_{10}(C^*)$ by 2.5. The volatility distribution of products is presented below, and is now included in the text of the paper. We discuss this volatility parametrisation and the implications in the following paragraphs

*"DBE as calculated by equation 3 is equal to the number of pi bonds and rings within a molecule. Benzene, toluene, and similar aromatics have DBE = 4, naphthalene = 7 and monoterpenes = 3. DBE can be used as an indicator of sources when considering HOM in bulk. Saturation mass concentration as calculated by equation 4 can help describe capacity of a molecule to both condense onto newly formed particles and participate in nucleation. Figure 8 shows concentrations of HOMs and other oxygenated organic molecules binned to the nearest integer $Log_{10}(C^*)(300$ K), coloured by DBE. Most measured molecules fall into the SVOC class ($0.3 < C^*(300$ K$) < 300$ µg m$^{-3}$) which will mostly exist in equilibrium between gas and particle phase. High SVOC concentrations arise from fingerprint molecules for isoprene oxidation under high NO$_x$ ($C_5H_{10}N_2O_8$) (Brean et al., 2019), and oxidation of small alkylbenzenes ($C_7H_8O_5$, $C_8H_{10}O_5$). LVOC and ELVOC ($3·10^{-5} < C^* < 0.3$ µg m$^{-3}$ and $3·10^{-9} < C^*(300$ K$) < 3·10^{-5}$ µg m$^{-3}$ respectively) have greater contribution from molecules with higher DBE, i.e., $C_{10}H_{10}O_8$ arising most likely from PAH*

*oxidation (Molteni et al., 2018), and $C_{10}H_{15}O_7N$, a common molecule arising from monoterpene oxidation in the presence of $NO_x$. The contribution of molecules with carbon number ≤ 9 to these LVOC is modest, and ELVOCs are entirely comprised of molecules with carbon numbers ≥ 10 and DBEs of 8 and 4. No molecules classed as ultra-low volatility organic compounds (ULVOC, $C^*$(300 K) < 3·10⁻⁹ µg m⁻³) were observed."*

The following is in our discussion of contribution to growth

*"Early stage particle growth is therefore plausibly driven by measured LVOC and ELVOC as these will be the molecules involatile enough to readily condense down on particles of this size. From 2D-VBS volatility calculations discussed in the previous section, we show that the LVOC and ELVOC we measure in Barcelona plausibly arise from the oxidation of aromatics (particularly PAHs in the case of ELVOC) and monoterpenes."*

And in our conclusions and abstract respectively, state

*"SVOC arose from mostly isoprene and alkylbenzene oxidation, whereas LVOC and ELVOC arose from alkylbenzene, monoterpene and PAH oxidation"*

*"The concentration of these HOMs shows a dependence on temperature, and HOMs primarily fall into the SVOC volatility class. LVOC arise from oxidation of alkylbenzenes, PAHs and monoterpenes, whereas ELVOC appear to arise from primarily PAH and monoterpene oxidation"*

From these data we also calculate growth rates as according to Nieminen et al., (2010) from $H_2SO_4$ condensation, from condensation of $H_2SO_4$, MSA and $HIO_3$, which were present at low concentration, and from condensation of organics by volatility class. Although we understand binning growth by volatility as such is a rather large oversimplification, we got good agreement between our measured $GR_{5-20}$ from condensation of SVOC, LVOC, ELVOC, MSA, $HIO_3$ and $H_2SO_4$, and $GR_{1.9-5}$ from LVOC, ELVOC, MSA, $HIO_3$ and $H_2SO_4$. The details of these calculations are included in the methods section, and are as follows

*"Growth rates from irreversible condensation of various vapours were calculated according to the method of Nieminen et al. (2010). At our measured relative humidity, sulphuric acid favours binding to 2 $H_2O$ molecules (Kúrten et al., 2007). As amine concentrations are likely limited, we presume no mass from amines in the condensing species. $H_2SO_4$ was assigned a density of 1.8 g cm⁻³. For simplicity, the properties of MSA regarding density and hydration were presumed the same as $H_2SO_4$, and $HIO_3$ was presumed to have the same hydration as $H_2SO_4$, with a density of 4.98 g cm⁻³. The density of condensing organic vapours was assumed to be 1.5 g cm⁻³, and concentration-weighted mean mass (~276 g mol⁻¹ for LVOC) and atomic weighted diffusion volumes of organic compounds were used to calculate GRs."*

*"The saturation vapour pressure at 300 K is defined by the 2D-volatility basis set (2D-VBS) as follows, if all nitrogen functionality is assumed to take the form -$ONO_2$ (Bianchi 2019; Donahue 2011; Schervish and Donahue, 2020):*

$$Log_{10}(C^*)(300\ K) = (N_{C0} - N_C)b_C - N_O b_O - 2\frac{N_O N_C}{N_C + N_O}b_{CO} - N_n b_N \qquad (4)$$

*Where $N_C$, $N_H$, and $N_N$, are the number of carbon, hydrogen, and nitrogen atoms respectively. $N_O$ is the number of oxygen atoms minus $3N_N$ to account for -$ONO_2$ groups, $N_{C0}$ is 25 (the carbon number of a 1 µg m⁻³ alkane), $b_C$, $b_O$, $b_{CO}$, and $b_N$ are 0.475, 0.2, 0.9 and 2.5 respectively, and represent interaction and nonideality terms. The final term of equation (4) represents the -$ONO_2$ groups, each reducing the saturation vapour pressure by 2.5 orders of magnitude. $C^*$ values are calculated at 300 K and not corrected, as 300 K is within 1 K of the campaign average temperature"*

We include the first of the below figures as a new Figure 8, and the growth-rate calculations as a supplementary figure. In light of these calculations we state the following in the text

*"Calculated growth rates according to the method of Nieminen et al. (2010) are presented in Figure S6 for both $GR_{1.9-5}$ and $GR_{5-20}$. Best agreement for $GR_{5-20}$ is when condensation of SVOC, LVOC, ELVOC, MSA, $HIO_3$ and $H_2SO_4$ is considered, and best agreement for $GR_{1.9-5}$ is seen for condensation of all these apart from SVOC. The uncertainties in this method are large, and assumptions of irreversible condensation of SVOC onto particles of 5 nm likely lead to overestimations; however, these results confirm the essential role of the condensation of organic compounds to produce high growth rates observed in urban environments."*

[Figure]

**Comment 5:** *The authors conclude: "We show new particle formation rates in Barcelona are linearly dependent upon the sulphuric acid concentrations, and this mechanism plausibly proceeds*

*by the formation of clusters involving sulphuric acid and highly oxygenated organic molecules, with likely involvement of bases". Where do the authors see the clusters between organics and sulfuric acid in Figure 8? Figure 5 shows the H2SO4/DMA nucleation rates from CLOUD by Almeida et al. This data has been revised by Kürten et al. 2018. The new values would be at least an order of magnitude higher than Barcelona, which could eventually be explained by the higher ambient temperature in Barcelona.*

**Response 5:** We unfortunately do not measure such clusters. This conclusion was based upon the observations of high formation rates matching those of Almeida et al. (2013), while simultaneously measuring a low sulphuric acid dimer:monomer ratio. The root cause of this, we hypothesised, could be the simultaneous involvement of sulphuric acid, bases and HOMs, however, in light of the updated Figure 6 utilising the data of Kürten et al. (2018), we revise our conclusions a little. We state the following in section 3.2

*"Nucleation rates measured in Barcelona ($J_{1.9}$ 178 ± 190 $cm^{-3}$ $s^{-1}$ at [$H_2SO_4$] 7.1·$10^6$ ± 2.7·$10^6$ $cm^{-3}$) are around an order of magnitude lower than that seen for the $H_2SO_4$-DMA-$H_2SO_4$ system , but exceed that of the $H_2SO_4$-BioOxOrg-$H_2O$ system by ~1 order of magnitude, and that of the $H_2SO_4$-$NH_3$-$H_2O$ and $H_2SO_4$-$H_2O$ system by multiple orders of magnitude. No dissimilarity is seen between the data points corresponding to full or burst type nucleation, indicating similar mechanisms of formation, despite lower HOM concentrations on burst-event days"*

*"Particle formation plausibly operates by sulphuric acid-amine nucleation involving the measured C2 and C4 amines in our data, with nucleation rates hindered relative to those measured in the CLOUD experiments by elevated temperatures, and a decline to the sulphuric acid dimer:monomer ratio indicates that base concentrations may be limited. We cannot rule out an involvement of HOMs in particle formation processes, and, as no higher-order clusters were observed, we cannot establish sulphuric acid-amine nucleation with certainty."*

*Comment 6: From Figure 7a the authors claim a temperature dependence of HOM formation. However, the higher HOM concentrations at high temperature are also accompanied by higher global radiation. Thus, this dependence could just represent day-night time chemistry and the dependence on OH concentration.*

**Response 6:** If we take just the daylight hours, or hours where insolation is >100 W $m^{-2}$, the correlation between temperature and signal gets slightly better $R^2 = 0.29$ → $R^2 = 0.30$. Taking the periods where insolation is <100 W $m^{-2}$ then there is no correlation seen. However, the correlation between HOMs and insolation is very poor ($R^2 = 0.15$). We maintain that there exists a dependence of HOM concentration on temperature within this dataset.

[revised manuscript text omitted]

**Figure S8:** Condensational growth rates between (a) 5 - 20 nm and (b) 1.9 – 5 nm, calculated from $H_2SO_4$ condensation, $H_2SO_4$, MSA, and $HIO_3$ condensation, and SVOC, LVOC, ELVOC, $H_2SO_4$, MSA and $HIO_3$ in (a), and LVOC, ELVOC, $H_2SO_4$, MSA and $HIO_3$ in (b). Also presented are growth rates from particle count data. Error bars represent uncertainties on the concentration of species measured by CI-APi-ToF, and the uncertainties from GR calculations. Systematic uncertainties from the methods of Nieminen et al. (2010) are not included.

[Figure]

**Figure S9:** Average diurnals of particle composition as measured by ACSM on (a) non-nucleation, (b) full-nucleation and (c) burst-nucleation days.

[Figure]

**Figure S10:** Location of sampling site.

**Table 1: Ions identified by CI-APi-ToF**

| Ion | m/Q |
| --- | --- |
| $Cl^-$ | 34.97 |
| $NO_2^-$ | 45.99 |
| $C_3H_3O^-$ | 55.02 |
| $(NO_3)^-$ | 61.99 |
| $C_3H_3O_2^-$ | 71.01 |
| $C_3H_5O_2^-$ | 73.03 |
| $Br^-$ | 78.92 |
| $H_2O(NO_3)^-$ | 80.00 |
| $C_4H_5O_2^-$ | 85.03 |
| $C_3H_3O_3^-$ | 87.01 |
| $CH_3SO_3^-$ | 94.98 |
| $CFH_3(NO3)^-$ | 96.01 |
| $HSO_4^-$ | 96.96 |
| $HCl(NO3)^-$ | 97.97 |
| $H_4O_2(NO3)^-$ | 98.01 |
| $C_4H_5O_3^-$ | 101.02 |
| $C_3H_3O_4^-$ | 103.00 |
| $SO_5^-$ | 111.95 |
| $C_4H_3O_4^-$ | 115.00 |
| $C_4H_5O_4^-$ | 117.02 |
| $C_3H_4O(NO_3)^-$ | 118.01 |
| $C_3H_3O_5^-$ | 119.00 |
| $HNO_3NO_3^-$ | 124.98 |
| $I^-$ | 126.91 |
| $C_4H_3O_5^-$ | 131.00 |
| $C_5H_7O_4^-$ | 131.03 |
| $C_4H_5O_5^-$ | 133.01 |
| $NH_3(HNO_3)(NO_3)^-$ | 142.01 |
| $C_5H_5O_5^-$ | 145.01 |
| $C_5H_7O_5^-$ | 147.03 |
| $C_3H_6O_3(NO_3)^-$ | 152.02 |
| $C_6H_7O_5^-$ | 159.03 |
| $C_4H_6O_3(NO_3)^-$ | 164.02 |

| | |
|---|---|
| $C_3H_5NO_3(NO_3)^-$ | 165.02 |
| $C_3H_4O_4(NO_3)^-$ | 166.00 |
| $C_6H_5NO(NO_3)^-$ | 169.03 |
| $C_2H_7N(HNO_3)(NO_3)^-$ | 170.04 |
| $C_7H_7O_5^-$ | 171.03 |
| $C_7H_9O_5^-$ | 173.05 |
| $IO_3^-$ | 174.89 |
| $C_4H_5NO_3(NO_3)^-$ | 177.02 |
| $C_4H_4O_4(NO_3)^-$ | 178.00 |
| $C_5H_7O_7^-$ | 179.02 |
| $C_4H_6O_4(NO_3)^-$ | 180.01 |
| $C_5H_9O_7^-$ | 181.04 |
| $C_4H_8O_4(NO_3)^-$ | 182.03 |
| $C_8H_{11}O_5^-$ | 187.06 |
| $(HNO_3)_2(NO_3)^-$ | 187.98 |
| $C_7H_9O_6^-$ | 189.04 |
| $C_5H_7NO_3(NO_3)^-$ | 191.03 |
| $C_4H_6N_2O_3(NO_3)^-$ | 192.03 |
| $C_4H_5NO_4(NO_3)^-$ | 193.01 |
| $C_5H_8O_4(NO_3)^-$ | 194.03 |
| $H_2SO_4HSO_4^-$ | 194.93 |
| $C_5H_7O_8^-$ | 195.01 |
| $C_3H_6N_2O_4(NO_3)^-$ | 196.02 |
| $C_4H_8O_5(NO_3)^-$ | 198.03 |
| $C_4H_{11}N(HNO_3)(NO_3)^-$ | 198.07 |
| $C_7H_7NO_2(NO_3)^-$ | 199.04 |
| $C_6H_5NO_3(NO_3)^-$ | 201.02 |
| $C_8H_{11}O_6^-$ | 203.06 |
| $C_6H_6O_4(NO_3)^-$ | 204.01 |
| $NH_3(HNO3)_2(NO_3)^-$ | 205.01 |
| $C_6H_8O_4(NO_3)^-$ | 206.03 |
| $C_5H_7NO_4(NO_3)^-$ | 207.03 |
| $C_4H_6N_2O_4(NO_3)^-$ | 208.02 |
| $C_5H_8O_5(NO_3)^-$ | 210.03 |
| $C_4H_7NO_5(NO_3)^-$ | 211.02 |
| $C_8H_6O_3(NO_3)^-$ | 212.02 |

| | |
|---|---|
| $C_3H_5NO_6(NO_3)^-$ | 213.00 |
| $C_{10}H_{13}O_5^-$ | 213.08 |
| $C_4H_8O_6(NO_3)^-$ | 214.02 |
| $C_7H_7NO_3(NO_3)^-$ | 215.03 |
| $C_7H_6O_4(NO_3)^-$ | 216.01 |
| $C_7H_9NO_3(NO_3)^-$ | 217.05 |
| $C_7H_8O_4(NO_3)^-$ | 218.03 |
| $C_7H_{10}O_4(NO_3)^-$ | 220.05 |
| $C_6H_9NO_4(NO_3)^-$ | 221.04 |
| $C_5H_8N_2O_4(NO_3)^-$ | 222.04 |
| $C_{10}H_7O_6^-$ | 223.02 |
| $C_5H_8O_6(NO_3)^-$ | 226.02 |
| $C_4H_7NO_6(NO_3)^-$ | 227.02 |
| $C_4H_6O_7(NO_3)^-$ | 228.00 |
| $C_8H_9NO_3(NO_3)^-$ | 229.05 |
| $C_7H_7NO_4(NO_3)^-$ | 231.03 |
| $C_2H_7N(HNO3)_2(NO_3)^-$ | 233.04 |
| $C_7H_{10}O_5(NO_3)^-$ | 236.04 |
| $C_7H_{12}O_5(NO_3)^-$ | 238.06 |
| $C_{10}H_7O_7^-$ | 239.02 |
| $C_4H_6N_2O_6(NO_3)^-$ | 240.01 |
| $C_5H_8O_7(NO_3)^-$ | 242.02 |
| $C_5H_{11}NO_6(NO_3)^-$ | 243.05 |
| $C_5H_{10}O_7(NO_3)^-$ | 244.03 |
| $C_9H_{12}O_4(NO_3)^-$ | 246.06 |
| $C_7H_7NO_5(NO_3)^-$ | 247.02 |
| $C_8H_{10}O_5(NO_3)^-$ | 248.04 |
| $C_7H_9NO_5(NO_3)^-$ | 249.04 |
| $C_8H_{12}O_5(NO_3)^-$ | 250.06 |
| $C_7H_{11}NO_5(NO_3)^-$ | 251.05 |
| $C_7H_{10}O_6(NO_3)^-$ | 252.04 |
| $C_7H_{12}O_6(NO_3)^-$ | 254.05 |
| $C_6H_{11}NO_6(NO_3)^-$ | 255.05 |
| $C_6H_{10}O_7(NO_3)^-$ | 256.03 |
| $C_5H_8O_8(NO_3)^-$ | 258.01 |
| $C_8H_7NO_5(NO_3)^-$ | 259.02 |

| | |
|---|---|
| $C_9H_{10}O_5(NO_3)^-$ | 260.04 |
| $C_4H_{11}N(HNO3)_2(NO3)^-$ | 261.07 |
| $C_9H_{12}O_5(NO_3)^-$ | 262.06 |
| $C_7H_7NO_6(NO_3)^-$ | 263.02 |
| $C_8H_{11}NO_5(NO_3)^-$ | 263.05 |
| $C_8H_{10}O_6(NO_3)^-$ | 264.04 |
| $C_9H_{14}O_5(NO_3)^-$ | 264.07 |
| $C_7H_9NO_6^-(NO_3)^-$ | 265.03 |
| $C_7H_8O_7(NO_3)^-$ | 266.01 |
| $C_8H_{12}O_6(NO_3)^-$ | 266.05 |
| $C_7H_{11}NO_6(NO_3)^-$ | 267.05 |
| $C_8H_{15}NO_5(NO_3)^-$ | 267.08 |
| $C_7H_{10}O_7(NO_3)^-$ | 268.03 |
| $C_6H_{11}NO_7(NO_3)^-$ | 271.04 |
| $C_6H_{10}O_8(NO_3)^-$ | 272.03 |
| $C_5H_9NO_8(NO_3)^-$ | 273.02 |
| $C_5H_8O_9(NO_3)^-$ | 274.01 |
| $C_{10}H_{12}O_5(NO_3)^-$ | 274.06 |
| $C_{10}H_{14}O_5(NO_3)^-$ | 276.07 |
| $C_9H_{13}NO_5(NO_3)^-$ | 277.07 |
| $C_9H_{12}O_6(NO_3)^-$ | 278.05 |
| $C_{10}H_{16}O_5(NO_3)^-$ | 278.09 |
| $C_8H_{11}NO_6^-(NO_3)^-$ | 279.05 |
| $C_9H_{14}O_6(NO_3)^-$ | 280.07 |
| $C_7H_9NO_7(NO_3)^-$ | 281.03 |
| $C_8H_{12}O_7(NO_3)^-$ | 282.05 |
| $C_7H_{11}NO_7(NO_3)^-$ | 283.04 |
| $C_8H_{14}O_7(NO_3)^-$ | 284.06 |
| $C_{10}H_9NO_5(NO_3)^-$ | 285.04 |
| $C_5H_8N_2O_8(NO_3)^-$ | 286.02 |
| $C_5H_7NO_9(NO_3)^-$ | 287.00 |
| $C_7H_{15}NO_7(NO_3)^-$ | 287.07 |
| $C_5H_{10}N_2O_8(NO_3)^-$ | 288.03 |
| $C_5H_9NO_9(NO_3)^-$ | 289.02 |
| $C_{10}H_{12}O_6(NO_3)^-$ | 290.05 |
| $C_{10}H_{15}NO_5(NO_3)^-$ | 291.08 |

| | |
|---|---|
| $C_{10}H_{14}O_6(NO_3)^-$ | 292.07 |
| $C_9H_{13}NO_6(NO_3)^-$ | 293.06 |
| $C_{10}H_{16}O_6(NO_3)^-$ | 294.08 |
| $C_8H_{11}NO_7(NO_3)^-$ | 295.04 |
| $C_9H_{14}O_7(NO_3)^-$ | 296.06 |
| $C_8H_{13}NO_7(NO_3)^-$ | 297.06 |
| $C_8H_{12}O_8(NO_3)^-$ | 298.04 |
| $C_7H_{10}O_9(NO_3)^-$ | 300.02 |
| $C_{10}H_9NO_6(NO_3)^-$ | 301.03 |
| $C_7H_{12}O_9(NO_3)^-$ | 302.04 |
| $C_{10}H_{11}NO_6(NO_3)^-$ | 303.05 |
| $C_6H_{10}O_{10}(NO_3)^-$ | 304.02 |
| $C_{12}H_{18}O_5(NO_3)^-$ | 304.10 |
| $C_9H_9NO_7(NO_3)^-$ | 305.03 |
| $C_9H_8O_8(NO_3)^-$ | 306.01 |
| $C_{11}H_{16}O_6(NO_3)^-$ | 306.08 |
| $C_{10}H_{15}NO_6(NO_3)^-$ | 307.08 |
| $C_{10}H_{14}O_7(NO_3)^-$ | 308.06 |
| $C_9H_{13}NO_7(NO_3)^-$ | 309.06 |
| $C_{10}H_{16}O_7(NO_3)^-$ | 310.08 |
| $C_9H_{15}NO_7(NO_3)^-$ | 311.07 |
| $C_9H_{14}O_8(NO_3)^-$ | 312.06 |
| $C_8H_{13}NO_8(NO_3)^-$ | 313.05 |
| $C_8H_{12}O_9(NO_3)^-$ | 314.04 |
| $C_7H_{11}NO_9(NO_3)^-$ | 315.03 |
| $C_{10}H_{10}N_2O_6(NO_3)^-$ | 316.04 |
| $C_{10}H_9NO_7(NO_3)^-$ | 317.03 |
| $C_{12}H_{16}O_6(NO_3)^-$ | 318.08 |
| $C_{10}H_{11}NO_7(NO_3)^-$ | 319.04 |
| $C_{10}H_{10}O_8(NO_3)^-$ | 320.03 |
| $C_{12}H_{18}O_6(NO_3)^-$ | 320.10 |
| $C_{10}H_{13}NO_7(NO_3)^-$ | 321.06 |
| $C_{11}H_{16}O_7(NO_3)^-$ | 322.08 |
| $C_{10}H_{15}NO_7(NO_3)^-$ | 323.07 |
| $C_{10}H_{14}O_8(NO_3)^-$ | 324.06 |
| $C_{10}H_{16}O_8(NO_3)^-$ | 326.07 |

| | |
|---|---|
| $C_9H_{15}NO_8(NO_3)^-$ | 327.07 |
| $C_9H_{14}O_9(NO_3)^-$ | 328.05 |
| $C_8H_{13}NO_9(NO_3)^-$ | 329.05 |
| $C_9H_{16}O_9(NO_3)^-$ | 330.07 |
| $C_{11}H_{11}NO_7(NO_3)^-$ | 331.04 |
| $C_{10}H_9NO_8(NO_3)^-$ | 333.02 |
| $C_{12}H_{16}O_7(NO_3)^-$ | 334.08 |
| $C_{12}H_{18}O_7(NO_3)^-$ | 336.09 |
| $C_{10}H_{13}NO_8(NO_3)^-$ | 337.05 |
| $C_{11}H_{16}O_8(NO_3)^-$ | 338.07 |
| $C_{10}H_{15}NO_8(NO_3)^-$ | 339.07 |
| $C_{10}H_{14}O_9(NO_3)^-$ | 340.05 |
| $C_{13}H_{13}NO_6(NO_3)^-$ | 341.06 |
| $C_{10}H_{16}O_9(NO_3)^-$ | 342.07 |
| $C_9H_{15}NO_9(NO_3)^-$ | 343.06 |
| $C_9H_{14}O_{10}(NO_3)^-$ | 344.05 |
| $C_{12}H_{13}NO_7(NO_3)^-$ | 345.06 |
| $C_{13}H_{16}O_7(NO_3)^-$ | 346.08 |
| $C_{13}H_{18}O_7(NO_3)^-$ | 348.09 |
| $C_{12}H_{17}NO_7(NO_3)^-$ | 349.09 |
| $C_{13}H_{20}O_7(NO_3)^-$ | 350.11 |
| $C_{11}H_{15}NO_8(NO_3)^-$ | 351.07 |
| $C_{11}H_{17}NO_8(NO_3)^-$ | 353.08 |
| $C_{10}H_{15}NO_9(NO_3)^-$ | 355.06 |
| $C_{11}H_{18}O_9(NO_3)^-$ | 356.08 |
| $C_{10}H_{17}NO_9(NO_3)^-$ | 357.08 |
| $C_{13}H_{14}O_8(NO_3)^-$ | 360.06 |
| $C_{12}H_{13}NO_8(NO_3)^-$ | 361.05 |
| $C_{13}H_{19}NO_7(NO_3)^-$ | 363.10 |
| $C_{13}H_{18}O_8(NO_3)^-$ | 364.09 |
| $C_{12}H_{17}NO_8(NO_3)^-$ | 365.08 |
| $C_{13}H_{20}O_8(NO_3)^-$ | 366.10 |
| $C_{12}H_{19}NO_8(NO_3)^-$ | 367.10 |
| $C_{12}H_{18}O_9(NO_3)^-$ | 368.08 |
| $C_{11}H_{17}NO_9(NO_3)^-$ | 369.08 |
| $C_{11}H_{16}O_{10}(NO_3)^-$ | 370.06 |

| | |
|---|---|
| $C_{10}H_{15}NO_{10}(NO_3)^-$ | 371.06 |
| $C_{14}H_{14}O_8(NO_3)^-$ | 372.06 |
| $C_{13}H_{13}NO_8(NO_3)^-$ | 373.05 |
| $C_{14}H_{16}O_8(NO_3)^-$ | 374.07 |
| $C_{14}H_{20}O_8(NO_3)^-$ | 378.10 |
| $C_{13}H_{19}NO_8(NO_3)^-$ | 379.10 |
| $C_{14}H_{22}O_8(NO_3)^-$ | 380.12 |
| $C_{13}H_{21}NO_8(NO_3)^-$ | 381.12 |
| $C_{13}H_{20}O_9(NO_3)^-$ | 382.10 |
| $C_{16}H_{19}NO_6(NO_3)^-$ | 383.11 |
| $C_{16}H_{18}O_7(NO_3)^-$ | 384.09 |
| $C_{15}H_{17}NO_7(NO_3)^-$ | 385.09 |
| $C_{15}H_{16}O_8(NO_3)^-$ | 386.07 |
| $C_{10}H_{15}NO_{11}(NO_3)^-$ | 387.05 |
| $C_{11}H_{18}O_{11}(NO_3)^-$ | 388.07 |
| $C_{18}H_{21}N(NO_3)O_5^-$ | 393.13 |
| $C_{15}H_{24}(NO_3)O_8^-$ | 394.14 |
| $C_{18}H_{23}N(NO_3)O_5^-$ | 395.15 |
| $C_{18}H_{22}(NO_3)O_6^-$ | 396.13 |
| $C_{17}H_{21}N(NO_3)O_6^-$ | 397.13 |
| $C_{17}H_{20}(NO_3)O_7^-$ | 398.11 |
| $C_{16}H_{19}N(NO_3)O_7^-$ | 399.10 |
| $C_{16}H_{18}(NO_3)O_8^-$ | 400.09 |
| $C_{15}H_{17}N(NO_3)O_8^-$ | 401.08 |
| $C_{12}H_{20}(NO_3)O_{11}^-$ | 402.09 |
| $C_{10}H_{15}N(NO_3)O_{12}^-$ | 403.05 |
| $C_{15}H_{18}(NO_3)O_9^-$ | 404.08 |
| $C_{18}H_{17}N(NO_3)O_6^-$ | 405.09 |
| $C_{19}H_{23}N(NO_3)O_5^-$ | 407.15 |
| $C_{19}H_{22}(NO_3)O_6^-$ | 408.13 |

---

## Author Response (AR2)

Journal: ACP - MS No.: acp-2020-84

Title: Molecular Insights into New Particle Formation in Barcelona, Spain

Author(s): Brean et al.

**RESPONSE TO THE EDITOR**

Referee comments in this text will appear in blue, responses in black, and where new text from the manuscript is quoted, it will appear in red, and **bold** where text has been inserted into a larger section

*Comment 1: Referring to the first comment by referee 1, the multiplication factor in calculating CS should indeed be $2\pi D$, not $4\pi D$ as in equation 1 or incorrectly given in Kulmala et al. 2012. Note that in the original paper by Kulmala et al. 2001 which introduced CS, there is $4\pi D$ but that is multiplied by the particle radius, not diameter as currently in practice. Unfortunately, this has caused plenty of confusion during recent years. I would encourage the authors to correct this point: at the end it has little influence on the actual results, affecting mainly the absolute values of reported CS.*

**Response 1:** This has been amended in the methods section, further, we have fixed these values both in our figures (Figures 3, and S2) and in the text (line 280)

*Comment 2: In general, the paper is very clearly written and with a proper language. However, the following should be checked out:*
*1) in many places it reads "between M-N" although it should be either "between M and N" or "in the range of M-N",*
*2) in some of the new text, articles seems to be missing,*
*3) the text in section 2.4 requires some improvements, please check out.*

**Response 2:**
1) These have been fixed through the text and supplement
2) These have been added throughout the text, alongside an error in one of the references in-line
3) We have amended the final paragraph of this section to now read

*"…where $N_C$, $N_H$, and $N_N$, are the number of carbon, hydrogen, and nitrogen atoms respectively. $N_O$ is the number of oxygen atoms minus $3N_N$ to account for -$ONO_2$ groups, $N_{C0}$ is 25 (the carbon number of **an alkane with a saturation mass concentration of** 1 $\mu g\ m^{-3}$), $b_C$, $b_O$, $b_{CO}$, and $b_N$ are 0.475, 0.2, 0.9 and 2.5 respectively, and represent interaction and nonideality terms. The final term of equation (4) **account for** -$ONO_2$ groups, each reducing the saturation vapour pressure by 2.5 orders of magnitude. $C^*$ values are calculated at 300 K and not corrected for temperature, as 300 K is within 1 K of the campaign average temperature."*

*Comment 3: The term "high ozone" does sound scientifically correct, should it rather be "high ozone concentration"?*
**Response 3:** This has been added to the relevant lines (264 & 480)

*Comment 4: On lines 391-392 in the file marked by track changes, should one write "...with an increasing temperature"?*

**Response 4:** This has been changed and now reads as follows

*"Model studies of sulphuric acid-amine nucleation show a decline in nucleation rate **with an increasing** temperature"*

*Comment 5: The information in Yan et al (2018) in the reference list seems to be incomplete. Please correct.*
**Response 5:** This has been amended. Alongside this, we tidy up minor errors in some other references.

[revised manuscript text omitted]

**Figure S8:** Condensational growth rates  in the ranges (a) 5 - 20 nm and (b) 1.9 – 5 nm,
calculated from $H_2SO_4$ condensation, $H_2SO_4$, MSA, and $HIO_3$ condensation, and SVOC, LVOC,
ELVOC, $H_2SO_4$, MSA and $HIO_3$ in (a), and LVOC, ELVOC, $H_2SO_4$, MSA and $HIO_3$ in (b). Also
presented are growth rates from particle count data. Error bars represent uncertainties on the
concentration of species measured by CI-APi-ToF, and the uncertainties from GR calculations.
Systematic uncertainties from the methods of Nieminen et al. (2010) are not included.

[Figure]

**Figure S9:** Average diurnals of particle composition as measured by ACSM on (a) non-
nucleation, (b) full-nucleation and (c) burst-nucleation days.

[Figure]

**Figure S10:** Location of sampling site.

**Table 1:** Ions identified by CI-APi-ToF

| Ion | m/Q |
|---|---|
| Cl⁻ | 34.97 |
| $NO_2^-$ | 45.99 |
| $C_3H_3O^-$ | 55.02 |
| $(NO_3)^-$ | 61.99 |
| $C_3H_3O_2^-$ | 71.01 |
| $C_3H_5O_2^-$ | 73.03 |
| Br⁻ | 78.92 |
| $H_2O(NO_3)^-$ | 80.00 |
| $C_4H_5O_2^-$ | 85.03 |
| $C_3H_3O_3^-$ | 87.01 |
| $CH_3SO_3^-$ | 94.98 |
| $CFH_3(NO3)^-$ | 96.01 |
| $HSO_4^-$ | 96.96 |
| $HCl(NO3)^-$ | 97.97 |
| $H_4O_2(NO3)^-$ | 98.01 |
| $C_4H_5O_3^-$ | 101.02 |
| $C_3H_3O_4^-$ | 103.00 |
| $SO_5^-$ | 111.95 |
| $C_4H_3O_4^-$ | 115.00 |
| $C_4H_5O_4^-$ | 117.02 |
| $C_3H_4O(NO_3)^-$ | 118.01 |
| $C_3H_3O_5^-$ | 119.00 |
| $HNO_3NO_3^-$ | 124.98 |
| I⁻ | 126.91 |
| $C_4H_3O_5^-$ | 131.00 |
| $C_5H_7O_4^-$ | 131.03 |
| $C_4H_5O_5^-$ | 133.01 |
| $NH_3(HNO_3)(NO_3)^-$ | 142.01 |
| $C_5H_5O_5^-$ | 145.01 |
| $C_5H_7O_5^-$ | 147.03 |
| $C_3H_6O_3(NO_3)^-$ | 152.02 |
| $C_6H_7O_5^-$ | 159.03 |
| $C_4H_6O_3(NO_3)^-$ | 164.02 |
| $C_3H_5NO_3(NO_3)^-$ | 165.02 |
| $C_3H_4O_4(NO_3)^-$ | 166.00 |
| $C_6H_5NO(NO_3)^-$ | 169.03 |
| $C_2H_7N(HNO_3)(NO_3)^-$ | 170.04 |
| $C_7H_7O_5^-$ | 171.03 |
| $C_7H_9O_5^-$ | 173.05 |
| $IO_3^-$ | 174.89 |
| $C_4H_5NO_3(NO_3)^-$ | 177.02 |
| $C_4H_4O_4(NO_3)^-$ | 178.00 |
| $C_5H_7O_7^-$ | 179.02 |

| | |
|---|---|
| $C_4H_6O_4(NO_3)^-$ | 180.01 |
| $C_5H_9O_7^-$ | 181.04 |
| $C_4H_8O_4(NO_3)^-$ | 182.03 |
| $C_8H_{11}O_5^-$ | 187.06 |
| $(HNO_3)_2(NO_3)^-$ | 187.98 |
| $C_7H_9O_6^-$ | 189.04 |
| $C_5H_7NO_3(NO_3)^-$ | 191.03 |
| $C_4H_6N_2O_3(NO_3)^-$ | 192.03 |
| $C_4H_5NO_4(NO_3)^-$ | 193.01 |
| $C_5H_8O_4(NO_3)^-$ | 194.03 |
| $H_2SO_4HSO_4^-$ | 194.93 |
| $C_5H_7O_8^-$ | 195.01 |
| $C_3H_6N_2O_4(NO_3)^-$ | 196.02 |
| $C_4H_8O_5(NO_3)^-$ | 198.03 |
| $C_4H_{11}N(HNO_3)(NO_3)^-$ | 198.07 |
| $C_7H_7NO_2(NO_3)^-$ | 199.04 |
| $C_6H_5NO_3(NO_3)^-$ | 201.02 |
| $C_8H_{11}O_6^-$ | 203.06 |
| $C_6H_6O_4(NO_3)^-$ | 204.01 |
| $NH_3(HNO3)_2(NO_3)^-$ | 205.01 |
| $C_6H_8O_4(NO_3)^-$ | 206.03 |
| $C_5H_7NO_4(NO_3)^-$ | 207.03 |
| $C_4H_6N_2O_4(NO_3)^-$ | 208.02 |
| $C_5H_8O_5(NO_3)^-$ | 210.03 |
| $C_4H_7NO_5(NO_3)^-$ | 211.02 |
| $C_8H_6O_3(NO_3)^-$ | 212.02 |
| $C_3H_5NO_6(NO_3)^-$ | 213.00 |
| $C_{10}H_{13}O_5^-$ | 213.08 |
| $C_4H_8O_6(NO_3)^-$ | 214.02 |
| $C_7H_7NO_3(NO_3)^-$ | 215.03 |
| $C_7H_6O_4(NO_3)^-$ | 216.01 |
| $C_7H_9NO_3(NO_3)^-$ | 217.05 |
| $C_7H_8O_4(NO_3)^-$ | 218.03 |
| $C_7H_{10}O_4(NO_3)^-$ | 220.05 |
| $C_6H_9NO_4(NO_3)^-$ | 221.04 |
| $C_5H_8N_2O_4(NO_3)^-$ | 222.04 |
| $C_{10}H_7O_6^-$ | 223.02 |
| $C_5H_8O_6(NO_3)^-$ | 226.02 |
| $C_4H_7NO_6(NO_3)^-$ | 227.02 |
| $C_4H_6O_7(NO_3)^-$ | 228.00 |
| $C_8H_9NO_3(NO_3)^-$ | 229.05 |
| $C_7H_7NO_4(NO_3)^-$ | 231.03 |
| $C_2H_7N(HNO3)_2(NO_3)^-$ | 233.04 |
| $C_7H_{10}O_5(NO_3)^-$ | 236.04 |
| $C_7H_{12}O_5(NO_3)^-$ | 238.06 |
| $C_{10}H_7O_7^-$ | 239.02 |

| | |
|---|---|
| $C_4H_6N_2O_6(NO_3)^-$ | 240.01 |
| $C_5H_8O_7(NO_3)^-$ | 242.02 |
| $C_5H_{11}NO_6(NO_3)^-$ | 243.05 |
| $C_5H_{10}O_7(NO_3)^-$ | 244.03 |
| $C_9H_{12}O_4(NO_3)^-$ | 246.06 |
| $C_7H_7NO_5(NO_3)^-$ | 247.02 |
| $C_8H_{10}O_5(NO_3)^-$ | 248.04 |
| $C_7H_9NO_5(NO_3)^-$ | 249.04 |
| $C_8H_{12}O_5(NO_3)^-$ | 250.06 |
| $C_7H_{11}NO_5(NO_3)^-$ | 251.05 |
| $C_7H_{10}O_6(NO_3)^-$ | 252.04 |
| $C_7H_{12}O_6(NO_3)^-$ | 254.05 |
| $C_6H_{11}NO_6(NO_3)^-$ | 255.05 |
| $C_6H_{10}O_7(NO_3)^-$ | 256.03 |
| $C_5H_8O_8(NO_3)^-$ | 258.01 |
| $C_8H_7NO_5(NO_3)^-$ | 259.02 |
| $C_9H_{10}O_5(NO_3)^-$ | 260.04 |
| $C_4H_{11}N(HNO3)_2(NO3)^-$ | 261.07 |
| $C_9H_{12}O_5(NO_3)^-$ | 262.06 |
| $C_7H_7NO_6(NO_3)^-$ | 263.02 |
| $C_8H_{11}NO_5(NO_3)^-$ | 263.05 |
| $C_8H_{10}O_6(NO_3)^-$ | 264.04 |
| $C_9H_{14}O_5(NO_3)^-$ | 264.07 |
| $C_7H_9NO_6^-(NO_3)^-$ | 265.03 |
| $C_7H_8O_7(NO_3)^-$ | 266.01 |
| $C_8H_{12}O_6(NO_3)^-$ | 266.05 |
| $C_7H_{11}NO_6(NO_3)^-$ | 267.05 |
| $C_8H_{15}NO_5(NO_3)^-$ | 267.08 |
| $C_7H_{10}O_7(NO_3)^-$ | 268.03 |
| $C_6H_{11}NO_7(NO_3)^-$ | 271.04 |
| $C_6H_{10}O_8(NO_3)^-$ | 272.03 |
| $C_5H_9NO_8(NO_3)^-$ | 273.02 |
| $C_5H_8O_9(NO_3)^-$ | 274.01 |
| $C_{10}H_{12}O_5(NO_3)^-$ | 274.06 |
| $C_{10}H_{14}O_5(NO_3)^-$ | 276.07 |
| $C_9H_{13}NO_5(NO_3)^-$ | 277.07 |
| $C_9H_{12}O_6(NO_3)^-$ | 278.05 |
| $C_{10}H_{16}O_5(NO_3)^-$ | 278.09 |
| $C_8H_{11}NO_6^-(NO_3)^-$ | 279.05 |
| $C_9H_{14}O_6(NO_3)^-$ | 280.07 |
| $C_7H_9NO_7(NO_3)^-$ | 281.03 |
| $C_8H_{12}O_7(NO_3)^-$ | 282.05 |
| $C_7H_{11}NO_7(NO_3)^-$ | 283.04 |
| $C_8H_{14}O_7(NO_3)^-$ | 284.06 |
| $C_{10}H_9NO_5(NO_3)^-$ | 285.04 |
| $C_5H_8N_2O_8(NO_3)^-$ | 286.02 |

| | |
|---|---|
| $C_5H_7NO_9(NO_3)^-$ | 287.00 |
| $C_7H_{15}NO_7(NO_3)^-$ | 287.07 |
| $C_5H_{10}N_2O_8(NO_3)^-$ | 288.03 |
| $C_5H_9NO_9(NO_3)^-$ | 289.02 |
| $C_{10}H_{12}O_6(NO_3)^-$ | 290.05 |
| $C_{10}H_{15}NO_5(NO_3)^-$ | 291.08 |
| $C_{10}H_{14}O_6(NO_3)^-$ | 292.07 |
| $C_9H_{13}NO_6(NO_3)^-$ | 293.06 |
| $C_{10}H_{16}O_6(NO_3)^-$ | 294.08 |
| $C_8H_{11}NO_7(NO_3)^-$ | 295.04 |
| $C_9H_{14}O_7(NO_3)^-$ | 296.06 |
| $C_8H_{13}NO_7(NO_3)^-$ | 297.06 |
| $C_8H_{12}O_8(NO_3)^-$ | 298.04 |
| $C_7H_{10}O_9(NO_3)^-$ | 300.02 |
| $C_{10}H_9NO_6(NO_3)^-$ | 301.03 |
| $C_7H_{12}O_9(NO_3)^-$ | 302.04 |
| $C_{10}H_{11}NO_6(NO_3)^-$ | 303.05 |
| $C_6H_{10}O_{10}(NO_3)^-$ | 304.02 |
| $C_{12}H_{18}O_5(NO_3)^-$ | 304.10 |
| $C_9H_9NO_7(NO_3)^-$ | 305.03 |
| $C_9H_8O_8(NO_3)^-$ | 306.01 |
| $C_{11}H_{16}O_6(NO_3)^-$ | 306.08 |
| $C_{10}H_{15}NO_6(NO_3)^-$ | 307.08 |
| $C_{10}H_{14}O_7(NO_3)^-$ | 308.06 |
| $C_9H_{13}NO_7(NO_3)^-$ | 309.06 |
| $C_{10}H_{16}O_7(NO_3)^-$ | 310.08 |
| $C_9H_{15}NO_7(NO_3)^-$ | 311.07 |
| $C_9H_{14}O_8(NO_3)^-$ | 312.06 |
| $C_8H_{13}NO_8(NO_3)^-$ | 313.05 |
| $C_8H_{12}O_9(NO_3)^-$ | 314.04 |
| $C_7H_{11}NO_9(NO_3)^-$ | 315.03 |
| $C_{10}H_{10}N_2O_6(NO_3)^-$ | 316.04 |
| $C_{10}H_9NO_7(NO_3)^-$ | 317.03 |
| $C_{12}H_{16}O_6(NO_3)^-$ | 318.08 |
| $C_{10}H_{11}NO_7(NO_3)^-$ | 319.04 |
| $C_{10}H_{10}O_8(NO_3)^-$ | 320.03 |
| $C_{12}H_{18}O_6(NO_3)^-$ | 320.10 |
| $C_{10}H_{13}NO_7(NO_3)^-$ | 321.06 |
| $C_{11}H_{16}O_7(NO_3)^-$ | 322.08 |
| $C_{10}H_{15}NO_7(NO_3)^-$ | 323.07 |
| $C_{10}H_{14}O_8(NO_3)^-$ | 324.06 |
| $C_{10}H_{16}O_8(NO_3)^-$ | 326.07 |
| $C_9H_{15}NO_8(NO_3)^-$ | 327.07 |
| $C_9H_{14}O_9(NO_3)^-$ | 328.05 |
| $C_8H_{13}NO_9(NO_3)^-$ | 329.05 |
| $C_9H_{16}O_9(NO_3)^-$ | 330.07 |

| | |
|---|---|
| $C_{11}H_{11}NO_7(NO_3)^-$ | 331.04 |
| $C_{10}H_9NO_8(NO_3)^-$ | 333.02 |
| $C_{12}H_{16}O_7(NO_3)^-$ | 334.08 |
| $C_{12}H_{18}O_7(NO_3)^-$ | 336.09 |
| $C_{10}H_{13}NO_8(NO_3)^-$ | 337.05 |
| $C_{11}H_{16}O_8(NO_3)^-$ | 338.07 |
| $C_{10}H_{15}NO_8(NO_3)^-$ | 339.07 |
| $C_{10}H_{14}O_9(NO_3)^-$ | 340.05 |
| $C_{13}H_{13}NO_6(NO_3)^-$ | 341.06 |
| $C_{10}H_{16}O_9(NO_3)^-$ | 342.07 |
| $C_9H_{15}NO_9(NO_3)^-$ | 343.06 |
| $C_9H_{14}O_{10}(NO_3)^-$ | 344.05 |
| $C_{12}H_{13}NO_7(NO_3)^-$ | 345.06 |
| $C_{13}H_{16}O_7(NO_3)^-$ | 346.08 |
| $C_{13}H_{18}O_7(NO_3)^-$ | 348.09 |
| $C_{12}H_{17}NO_7(NO_3)^-$ | 349.09 |
| $C_{13}H_{20}O_7(NO_3)^-$ | 350.11 |
| $C_{11}H_{15}NO_8(NO_3)^-$ | 351.07 |
| $C_{11}H_{17}NO_8(NO_3)^-$ | 353.08 |
| $C_{10}H_{15}NO_9(NO_3)^-$ | 355.06 |
| $C_{11}H_{18}O_9(NO_3)^-$ | 356.08 |
| $C_{10}H_{17}NO_9(NO_3)^-$ | 357.08 |
| $C_{13}H_{14}O_8(NO_3)^-$ | 360.06 |
| $C_{12}H_{13}NO_8(NO_3)^-$ | 361.05 |
| $C_{13}H_{19}NO_7(NO_3)^-$ | 363.10 |
| $C_{13}H_{18}O_8(NO_3)^-$ | 364.09 |
| $C_{12}H_{17}NO_8(NO_3)^-$ | 365.08 |
| $C_{13}H_{20}O_8(NO_3)^-$ | 366.10 |
| $C_{12}H_{19}NO_8(NO_3)^-$ | 367.10 |
| $C_{12}H_{18}O_9(NO_3)^-$ | 368.08 |
| $C_{11}H_{17}NO_9(NO_3)^-$ | 369.08 |
| $C_{11}H_{16}O_{10}(NO_3)^-$ | 370.06 |
| $C_{10}H_{15}NO_{10}(NO_3)^-$ | 371.06 |
| $C_{14}H_{14}O_8(NO_3)^-$ | 372.06 |
| $C_{13}H_{13}NO_8(NO_3)^-$ | 373.05 |
| $C_{14}H_{16}O_8(NO_3)^-$ | 374.07 |
| $C_{14}H_{20}O_8(NO_3)^-$ | 378.10 |
| $C_{13}H_{19}NO_8(NO_3)^-$ | 379.10 |
| $C_{14}H_{22}O_8(NO_3)^-$ | 380.12 |
| $C_{13}H_{21}NO_8(NO_3)^-$ | 381.12 |
| $C_{13}H_{20}O_9(NO_3)^-$ | 382.10 |
| $C_{16}H_{19}NO_6(NO_3)^-$ | 383.11 |
| $C_{16}H_{18}O_7(NO_3)^-$ | 384.09 |
| $C_{15}H_{17}NO_7(NO_3)^-$ | 385.09 |
| $C_{15}H_{16}O_8(NO_3)^-$ | 386.07 |
| $C_{10}H_{15}NO_{11}(NO_3)^-$ | 387.05 |

| | |
|---|---|
| $C_{11}H_{18}O_{11}(NO_3)^-$ | 388.07 |
| $C_{18}H_{21}N(NO_3)O_5^-$ | 393.13 |
| $C_{15}H_{24}(NO_3)O_8^-$ | 394.14 |
| $C_{18}H_{23}N(NO_3)O_5^-$ | 395.15 |
| $C_{18}H_{22}(NO_3)O_6^-$ | 396.13 |
| $C_{17}H_{21}N(NO_3)O_6^-$ | 397.13 |
| $C_{17}H_{20}(NO_3)O_7^-$ | 398.11 |
| $C_{16}H_{19}N(NO_3)O_7^-$ | 399.10 |
| $C_{16}H_{18}(NO_3)O_8^-$ | 400.09 |
| $C_{15}H_{17}N(NO_3)O_8^-$ | 401.08 |
| $C_{12}H_{20}(NO_3)O_{11}^-$ | 402.09 |
| $C_{10}H_{15}N(NO_3)O_{12}^-$ | 403.05 |
| $C_{15}H_{18}(NO_3)O_9^-$ | 404.08 |
| $C_{18}H_{17}N(NO_3)O_6^-$ | 405.09 |
| $C_{19}H_{23}N(NO_3)O_5^-$ | 407.15 |
| $C_{19}H_{22}(NO_3)O_6^-$ | 408.13 |